# Policy Finetuning in Reinforcement Learning via Design of Experiments using Offline Data

**Ruiqi Zhang**

Department of Statistics
University of California, Berkeley
`rqzhang@berkeley.edu`

**Andrea Zanette**

Department of EECS
University of California, Berkeley
`zanette@berkeley.edu`

## Abstract

In some applications of reinforcement learning, a dataset of pre-collected experience is already available but it is also possible to acquire some additional online data to help improve the quality of the policy. However, it may be preferable to gather additional data with a single, non-reactive exploration policy and avoid the engineering costs associated with switching policies.

In this paper we propose an algorithm with provable guarantees that can leverage an offline dataset to design a single non-reactive policy for exploration. We theoretically analyze the algorithm and measure the quality of the final policy as a function of the local coverage of the original dataset and the amount of additional data collected.

## 1 Introduction

Reinforcement learning (RL) is a general framework for data-driven, sequential decision making [Puterman, 1994, Sutton and Barto, 2018]. In RL, a common goal is to identify a near-optimal policy, and there exist two main paradigms: *online* and *offline* RL.

Online RL is effective when the practical cost of a bad decision is low, such as in simulated environments (e.g., [Mnih et al., 2015, Silver et al., 2016]). In online RL, a well designed learning algorithm starts from tabula rasa and implements a sequence of policies with a value that should approach that of an optimal policy. When the cost of making a mistake is high, such as in healthcare [Gottesman et al., 2018] and in self-driving [Kiran et al., 2021], an offline approach is preferred. In offline RL, the agent uses a dataset of pre-collected experience to extract a policy that is as good as possible. In this latter case, the quality of the policy that can be extracted from the dataset is limited by the quality of the dataset.

Many applications, however, fall between these two opposite settings: for example, a company that sells products online has most likely recorded the feedback that it has received from its customers, but can also collect a small amount of additional strategic data in order to improve its recommendation engine. While in principle an online exploration algorithm can be used to collect fresh data, in practice there are a number of practical engineering considerations that require the policy to be

37th Conference on Neural Information Processing Systems (NeurIPS 2023).

deployed to be **non-reactive**. We say that a policy is non-reactive, (or passive, memoryless) if it chooses actions only according to the current state of the system. Most online algorithms are, by design, reactive to the data being acquired.

An example of a situation where non-reactive policies may be preferred are those where a human in the loop is required to validate each exploratory policy before they are deployed, to ensure they are of high quality [Dann et al., 2019] and safe [Yang et al., 2021], as well as free of discriminatory content [Koenecke et al., 2020]. Other situations that may warrant non-reactive exploration are those where the interaction with the user occurs through a distributed system with delayed feedback. In recommendation systems, data collection may only take minutes, but policy deployment and updates can span weeks [Afsar et al., 2022]. Similar considerations apply across various RL application domains, including healthcare [Yu et al., 2021], computer networks [Xu et al., 2018], and new material design [Raccuglia et al., 2016]. In all such cases, the engineering effort required to implement a system that handles real-time policy switches may be prohibitive: deploying a single, non-reactive policy is much preferred.

**Non-reactive exploration from offline data** Most exploration algorithms that we are aware of incorporate policy switches when they interact with the environment [Dann and Brunskill, 2015, Dann et al., 2017, Azar et al., 2017, Jin et al., 2018, Dann et al., 2019, Zanette and Brunskill, 2019, Zhang et al., 2020b]. Implementing a sequence of non-reactive policies is necessary in order to achieve near-optimal regret: the number of policy switches must be at least $\widetilde{O}\left(H\left|\mathcal{S}\right|\left|\mathcal{A}\right|\log\log K\right)$ where $\mathcal{S}, \mathcal{A}, H, K$ are the state space, action space, horizon and the total number of episodes, respectively [Qiao et al., 2022]. With no switches, i.e., when a fully non-reactive data collection strategy is implemented, it is information theoretically impossible [Xiao et al., 2022] to identify a good policy using a number of samples polynomial in the size of the state and action space.

However, these fundamental limits apply to the case where the agent learns from tabula rasa. In the more common case where offline data is available, we demonstrate that it is possible to leverage the dataset to design an effective non-reactive exploratory policy. More precisely, an available offline dataset contains information (e.g., transitions) about a certain area of the state-action space, a concept known as *partial coverage*. A dataset with partial coverage naturally identifies a 'sub-region' of the original MDP—more precisely, a sub-graph—that is relatively well explored. We demonstrate that it is possible to use the dataset to design a non-reactive policy that further explores such sub-region. The additional data collected can be used to learn a near-optimal policy in such sub-region.

In other words, exploration with no policy switches can collect additional information and compete with the best policy that is restricted to an area where the original dataset has sufficient information. The value of such policy can be much higher than the one that can be computed using only the offline dataset, and does not directly depend on a concentrability coefficient [Munos and Szepesvári, 2008, Chen and Jiang, 2019].

Perhaps surprisingly, addressing the problem of reactive exploration in reinforcement learning requires an approach that *combines both optimism and pessimism* in the face of uncertainty to explore efficiently. While optimism drives exploration, pessimism ensures that the agent explores conservatively, in a way that restricts its exploration effort to a region that it knows how to navigate, and so our paper makes a technical contribution which can be of independent interest.

**Contributions** To the best of our knowledge, this is the first paper with theoretical rigor that considers the problem of designing an experiment in reinforcement learning for online, passive exploration, using a dataset of pre-collected experience. More precisely, our contributions are as follows:

- We introduce an algorithm that takes as input a dataset, uses it to design and deploy a non-reactive exploratory policy, and then outputs a locally near-optimal policy.

- We introduce the concept of sparsified MDP, which is actively used by our algorithm to design the exploratory policy, as well as to theoretically analyze the quality of the final policy that it finds.

- We rigorously establish a nearly minimax-optimal upper bound for the sample complexity needed to learn a local $\varepsilon$-optimal policy using our algorithm. [1]

---

[1] More rigorously, our sample complexity matches the minimax lower bound when we have some degree of knowledge for the full MDP, see the discussion in Section section 5. The lower bound for the samples needed in reward-free case is proved in [Jin et al., 2020b], but their result applies to the non-honogeneous MDP. The sample complexity on homogeneous MDP should be shaved off by an $H$ factor.

## 2 Related Work

In this section we discuss some related literature. Our work is related to low-switching algorithms, but unlike those, we focus on the limit case where *no-switiches* are allowed. For more related work about low-switching algorithms, offline RL, task-agnostic RL, and reward-free RL we refer to Appendix F.

**Low-switching RL** In reinforcement learning, [Bai et al., 2019] first proposed Q-learning with UCB2 exploration, proving an $O(H^3 |\mathcal{S}| |\mathcal{A}| \log K)$ switching cost. This was later improved by a factor of $H$ by the UCBadvantage algorithm in [Zhang et al., 2020b]. Recently, [Qiao et al., 2022] generalized the policy elimination algorithm from [Cesa-Bianchi et al., 2013] and introduced APEVE, which attains an optimal $O(H |\mathcal{S}| |\mathcal{A}| \log \log K)$ switching cost. The reward-free version of their algorithm (which is not regret minimizing) has an $O(H |\mathcal{S}| |\mathcal{A}|)$ switching cost.

Similar ideas were soon applied in RL with linear function approximation [Gao et al., 2021, Wang et al., 2021, Qiao and Wang, 2022] and general function approximation [Qiao et al., 2023]. Additionally, numerous research efforts have focused on low-adaptivity in other learning domains, such as batched dueling bandits [Agarwal et al., 2022], batched convex optimization [Duchi et al., 2018], linear contextual bandits [Ruan et al., 2021], and deployment-efficient RL [Huang et al., 2022].

Our work was inspired by the problem of non-reactive policy design in linear contextual bandits. Given access to an offline dataset, [Zanette et al., 2021a] proposed an algorithm to output a single exploratory policy, which generates a dataset from which a near-optimal policy can be extracted. However, there are a number of additional challenges which arise in reinforcement learning, including the fact that the state space is only partially explored in the offline dataset. In fact, in reinforcement learning, [Xiao et al., 2022] established an exponential lower bound for any non-adaptive policy learning algorithm starting from tabula rasa.

## 3 Setup

Throughout this paper, we let $[n] = \{1, 2, ..., n\}$. We adopt the big-O notation, where $\widetilde{O}(\cdot)$ suppresses poly-log factors of the input parameters. We indicate the cardinality of a set $\mathcal{X}$ with $|\mathcal{X}|$.

**Markov decision process** We consider time-homogeneous episodic Markov decision processes (MDPs). They are defined by a finite state space $\mathcal{S}$, a finite action space $\mathcal{A}$, a trasition kernel $\mathbb{P}$, a reward function $r$ and the episodic length $H$. The transition probability $\mathbb{P}(s' \mid s, a)$, which does not depend on the current time-step $h \in [H]$, denotes the probability of transitioning to state $s' \in \mathcal{S}$ when taking action $a \in \mathcal{A}$ in the current state $s \in \mathcal{S}$. Typically we denote with $s_1$ the initial state. For simplicity, we consider deterministic reward functions $r : \mathcal{S} \times \mathcal{A} \to [0, 1]$. A deterministic non-reactive (or memoryless, or passive) policy $\pi = \{\pi_h\}_{h \in [H]}$ maps a given state to an action.

The value function is defined as the expected cumulated reward. It depends on the state $s$ under consideration, the transition $\mathbb{P}$ and reward $r$ that define the MDP as well as on the policy $\pi$ being implemented. It is defined as $V_h(s; \mathbb{P}, r, \pi) = \mathbb{E}_{\mathbb{P}, \pi}[\sum_{i=h}^{H} r(s_i, a_i) \mid s_h = s]$, where $\mathbb{E}_{\mathbb{P}, \pi}$ denotes the expectation generated by $\mathbb{P}$ and policy $\pi$. A closely related quantity is the state-action value function, or Q-function, defined as $Q_h(s, a; \mathbb{P}, r, \pi) = \mathbb{E}_{\mathbb{P}, \pi}[\sum_{i=h}^{H} r(s_i, a_i) \mid s_h = s, a_h = a]$. When it is clear from the context, we sometimes omit $(\mathbb{P}, r)$ and simply write them as $V_h^{\pi}(s)$ and $Q_h^{\pi}(s, a)$. We denote an MDP defined by $\mathcal{S}, \mathcal{A}$ and the transition matrix $\mathbb{P}$ as $\mathcal{M} = (\mathcal{S}, \mathcal{A}, \mathbb{P})$.

### 3.1 Interaction protocol

---
**Algorithm 1** Design of experiments in reinforcement learning
---
**Input:** Offline dataset $\mathcal{D}$
  1: *Offline phase:* use $\mathcal{D}$ to compute the exploratory policy $\pi_{ex}$
  2: *Online phase:* deploy $\pi_{ex}$ to collect the online dataset $\mathcal{D}'$
  3: *Planning phase:* receive the reward function $r$ and use $\mathcal{D} \cup \mathcal{D}'$ to extract $\pi_{final}$
**Output:** Return $\pi_{final}$

---

In this paper we assume access to an *offline dataset* $\mathcal{D} = \{(s, a, s')\}$ where every state-action $(s, a)$ is sampled in an i.i.d. fashion from some distribution $\mu$ and $s' \sim \mathbb{P}(\cdot \mid s, a)$, which is common in the offline RL literature [Xie et al., 2021a, Zhan et al., 2022, Rashidinejad et al., 2021, Uehara and Sun, 2021]. We denote $N(s, a)$ and $N(s, a, s')$ as the number of $(s, a)$ and $(s, a, s')$ samples in the offline dataset $\mathcal{D}$, respectively. The interaction protocol considered in this paper consists of three distinct phases, which are displayed in algorithm 1. They are:

- the **offline phase**, where the learner uses an *offline dataset* $\mathcal{D}$ of pre-collected experience to design the non-reactive exploratory policy $\pi_{ex}$;
- the **online phase** where $\pi_{ex}$ is deployed to generate the *online dataset* $\mathcal{D}'$;
- the **planning phase** where the learner receives a reward function and uses all the data collected to extract a good policy $\pi_{final}$ with respect to that reward function.

The objective is to minimize the number of online episodic interactions needed to find a policy $\pi_{final}$ whose value is as high as possible. Moreover, we focus on the reward-free RL setting [Jin et al., 2020a, Kaufmann et al., 2021, Li et al., 2023b], which is more general than reward-aware RL. In the offline and online phase, the data are generated without specific reward signals, and the entire reward information is then given in the planning phase. One of the primary advantages of using reward-free offline data is that it allows for the collection of data without the need for explicit reward signals. This can be particularly beneficial in environments where obtaining reward signals is costly, risky, ethically challenging, or where the reward functions are human-designed.

## 4 Algorithm: balancing optimism and pessimism for experimental design

In this section we outline our algorithm *Reward-Free Non-reactive Policy Design* (RF-NPD), which follows the high-level protocol described in algorithm 1. The technical novelty lies almost entirely in the design of the exploratory policy $\pi_{ex}$. In order to prepare the reader for the discussion of the algorithm, we first give some intuition in section 4.1 followed by the definition of sparsified MDP in section 4.2, a central concept of this paper, and then describe the implementation of line 1 in the protocol in algorithm 1 in section 4.3. We conclude by presenting the implementation of lines 2 and 3 in the protocol in algorithm 1.

### 4.1 Intuition

In order to present the main intuition for this paper, in this section we assume that enough transitions are available in the dataset for every edge $(s, a) \to s'$, namely that the *critical condition*

$$N(s, a, s') \geq \Phi = \widetilde{\Theta}(H^2) \tag{4.1}$$

holds for all tuples $(s, a, s') \in \mathcal{S} \times \mathcal{A} \times \mathcal{S}$ (the precise value for $\Phi$ will be given later in eq. (5.1)). Such condition is hardly satisfied everywhere in the state-action-state space, but assuming it in this section simplifies the presentation of one of the key ideas of this paper.

The key observation is that when eq. (4.1) holds for all $(s, a, s')$, we can use the empirical transition kernel to design an exploration policy $\pi_{ex}$ to eventually extract a near-optimal policy $\pi_{final}$ for any desired level of sub-optimality $\varepsilon$, despite eq. (4.1) being independent of $\varepsilon$. More precisely, let $\widehat{\mathbb{P}}$ be the empirical transition kernel defined in the usual way $\widehat{\mathbb{P}}(s' \mid s, a) = N(s, a, s')/N(s, a)$ for any tuple $(s, a, s')$. The intuition—which will be verified rigorously in the analysis of the algorithm—is the following:

*If eq. (4.1) holds for every $(s, a, s')$ then $\widehat{\mathbb{P}}$ can be used to design a non-reactive exploration policy $\pi_{ex}$ which can be deployed on $\mathcal{M}$ to find an $\varepsilon$-optimal policy $\pi_{final}$ using $\asymp \frac{1}{\varepsilon^2}$ samples.*

We remark that even if the condition 4.1 holds for all tuples $(s, a, s')$, the empirical kernel $\widehat{\mathbb{P}}$ is not accurate enough to extract an $\varepsilon$-optimal policy from the dataset $\mathcal{D}$ without collecting further data. Indeed, the threshold $\Phi = \widetilde{\Theta}(H^2)$ on the number of samples is independent of the desired sub-optimality $\varepsilon > 0$, while it is well known that at least $\sim \frac{1}{\varepsilon^2}$ offline samples are needed to find an $\varepsilon$-optimal policy. Therefore, directly implementing an offline RL algorithm to use the available

offline dataset $\mathcal{D}$ does not yield an $\varepsilon$-optimal policy. However, the threshold $\Phi = \widetilde{\Theta}(H^2)$ is sufficient to *design* a non-reactive exploratory policy $\pi_{ex}$ that can discover an $\varepsilon$-optimal policy $\pi_{final}$ after collecting $\sim \frac{1}{\varepsilon^2}$ online data.

## 4.2 Sparsified MDP

The intuition in the prior section must be modified to work with heterogeneous datasets and dynamics where $N(s, a, s') \geq \Phi$ may fail to hold everywhere. For example, if $\mathbb{P}(s' \mid s, a)$ is very small for a certain tuple $(s, a, s')$, it is unlikely that the dataset contains $N(s, a, s') \geq \Phi$ samples for that particular tuple. In a more extreme setting, if the dataset is empty, the critical condition in eq. (4.1) is violated for all tuples $(s, a, s')$, and in fact the lower bound of Xiao et al. [2022] states that finding $\varepsilon$-optimal policies by exploring with a non-reactive policy is not feasible with $\sim \frac{1}{\varepsilon^2}$ sample complexity. This suggests that in general it is not possible to output an $\varepsilon$-optimal policy using the protocol in algorithm 1.

However, a real-world dataset generally covers at least a portion of the state-action space, and so we expect the condition $N(s, a, s') \geq \Phi$ to hold somewhere; the sub-region of the MDP where it holds represents the connectivity graph of the *sparsified MDP*. This is the region that the agent knows how to navigate using the offline dataset $\mathcal{D}$, and so it is the one that the agent can explore further using $\pi_{ex}$. More precisely, the sparsified MDP is defined to have identical dynamics as the original MDP on the edges $(s, a) \longrightarrow s'$ that satisfy the critical condition 4.1. When instead the edge $(s, a) \longrightarrow s'$ fails to satisfy the critical condition 4.1, it is replaced with a transition $(s, a) \longrightarrow s^\dagger$ to an absorbing state $s^\dagger$.

**Definition 4.1** (Sparsified MDP). *Let $s^\dagger$ be an absorbing state, i.e., such that $\mathbb{P}\left(s^\dagger \mid s^\dagger, a\right) = 1$ and $r(s^\dagger, a) = 0$ for all $a \in \mathcal{A}$. The state space in the sparsified MDP $\mathcal{M}^\dagger$ is defined as that of the original MDP with the addition of $s^\dagger$. The dynamics $\mathbb{P}^\dagger$ of the sparsified MDP are defined as*

$$\mathbb{P}^\dagger(s' \mid s, a) = \begin{cases} \mathbb{P}(s' \mid s, a) & \text{if } N(s, a, s') \geq \Phi \\ 0 & \text{if } N(s, a, s') < \Phi, \end{cases} \qquad \mathbb{P}^\dagger(s^\dagger \mid s, a) = \sum_{\substack{s' \neq s^\dagger \\ N(s, a, s') < \Phi}} \mathbb{P}(s' \mid s, a).$$

(4.2)

*For any deterministic reward function $r : \mathcal{S} \times \mathcal{A} \to [0, 1]$, the reward function on the sparsified MDP is defined as $r^\dagger(s, a) = r(s, a)$; for simplicity we only consider deterministic reward functions.*

The *empirical sparsified MDP* $\widehat{\mathcal{M}}^\dagger = (\mathcal{S} \cup \{s^\dagger\}, \mathcal{A}, \widehat{\mathbb{P}}^\dagger)$ is defined in the same way but by using the empirical transition kernel in eq. (4.2). The empirical sparsified MDP is used by our algorithm to design the exploratory policy, while the (population) sparsified MDP is used for its theoretical analysis. They are two fundamental concepts in this paper. By formulating the sparsified MDP, we restrict the transitions and rewards within the area where we know how to navigate, embodying the principle of pessimism. Various forms of pessimistic regularization have been introduced to address the challenges of partially covered offline data. Examples include a pessimistic MDP [Kidambi et al., 2020] and limiting policies to those covered by offline data [Liu et al., 2020].

## 4.3 Offline design of experiments

In this section we describe the main sub-component of the algorithm, namely the sub-routine that uses the offline dataset $\mathcal{D}$ to compute the exploratory policy $\pi_{ex}$. The exploratory policy $\pi_{ex}$ is a mixture of the policies $\pi^1, \pi^2, \ldots$ produced by a variant of the reward-free exploration algorithm of [Kaufmann et al., 2021, Ménard et al., 2021]. Unlike prior literature, the reward-free algorithm is not interfaced with the real MDP $\mathcal{M}$, but rather *simulated* on *the empirical sparsified MDP* $\widehat{\mathcal{M}}^\dagger$. This avoids interacting with $\mathcal{M}$ with a reactive policy, but it introduces some bias that must be controlled. The overall procedure is detailed in algorithm 2. To be clear, no real-world samples are collected by algorithm 2; instead we use the word 'virtual samples' to refer to those generated from $\widehat{\mathcal{M}}^\dagger$.

At a high level, algorithm 2 implements value iteration using the empirical transition kernel $\widehat{\mathbb{P}}^\dagger$, with the exploration bonus defined in eq. (4.3) that replaces the reward function. The exploration bonus can be seen as implementing the principle of optimism in the face of uncertainty; however, the possibility of transitioning to an absorbing state with zero reward (due to the use of the absorbing state in the definition of $\widehat{\mathbb{P}}^\dagger$) implements the principle of pessimism.

**Algorithm 2** RF-UCB $(\widehat{\mathcal{M}}^{\dagger}, K_{ucb}, \varepsilon, \delta)$

---

**Input:** $\delta \in (0,1), \varepsilon > 0$, number of episode $K_{ucb}$, MDP $\widehat{\mathcal{M}}^{\dagger}$.
 1: **Initialize** Counter $n^1(s,a) = n^1(s,a,s') = 0$ for any $(s,a,s') \in \mathcal{S} \times \mathcal{A} \times \mathcal{S}$.
 2: **for** $k = 1, 2, ..., K_{ucb}$ **do**
 3:     **for** $h = H, H-1, \ldots, 1$ **do**
 4:         Set $U_h^k(s,a) = 0$ for any $(h,s,a) \in [H] \times \mathcal{S} \times \mathcal{A}$.
 5:         **for** $(s,a) \in \mathcal{S} \times \mathcal{A}$ **do**
 6:             Calculate the empirical uncertainty $U_h^k(s,a)$ using eq. (4.4) where $\phi$ is from eq. (4.3)
 7:         **end for**
 8:         $\pi_h^k(s) := \arg\max_{a \in \mathcal{A}} U_h^k(s,a), \forall s \in \mathcal{S}$ and $\pi_h^k(s^{\dagger}) :=$ any action.
 9:     **end for**
10:     Set initial state $s_1^k = s_1$.
11:     **for** $h = 1, 2, ..., H$ **do** Sample $a_h^k \sim \pi_h^k(s_h^k), s_{h+1}^k \sim \widehat{\mathbb{P}}^{\dagger}(s_h^k, a_h^k)$.
12:     **end for**
13:     $n^{k+1}(s,a) = n^k(s,a) + \sum_{h \in [H]} \mathbb{I}[(s,a) = (s_h^k, a_h^k)]$.
14:     $n^{k+1}(s,a,s') = n^k(s,a,s') + \sum_{h \in [H]} \mathbb{I}[(s,a,s') = (s_h^k, a_h^k, s_{h+1}^k)]$.
15: **end for**
**Output:** $\pi_{ex} = \text{Uniform}\{\pi^k\}_{k \in [K_{ucb}]}$.

---

This delicate *interplay between optimism and pessimism is critical* to the success of the overall procedure: while optimism encourages exploration, pessimism ensures that the exploration efforts are directed to the region of the state space that the agent actually knows how to navigate, and prevents the agent from getting 'trapped' in unknown regions. In fact, these latter regions could have combinatorial structures [Xiao et al., 2022] which cannot be explored with non-reactive policies.

More precisely, at the beginning of the $k$-th virtual episode in algorithm 2, $n^k(s,a)$ and $n^k(s,a,s')$ denote the counters for the number of virtual samples simulated from $\widehat{\mathcal{M}}^{\dagger}$ at each $(s,a)$ and $(s,a,s')$ tuple. We define the bonus function

$$\phi(x,\delta) = \frac{H}{x}[\log(6H|\mathcal{S}||\mathcal{A}|/\delta) + |\mathcal{S}|\log(e(1 + x/|\mathcal{S}|))], \tag{4.3}$$

which is used to construct the *empirical uncertainty function* $U_h^k$, a quantity that serves as a proxy for the uncertainty of the value of any policy $\pi$ on the spasified MDP. Specifically, for the $k$-th virtual episode, we set $U_{H+1}^k(s,a) = 0$ and $s \in \mathcal{S}, a \in \mathcal{A}$. For $h \in [H]$, we further define:

$$U_h^k(s,a) = H \min\{1, \phi(n^k(s,a))\} + \widehat{\mathbb{P}}^{\dagger}(s,a)^{\top}(\max_{a'} U_{h+1}^k(\cdot, a')). \tag{4.4}$$

Note that, the above bonus function takes a similar form of the bonus function in [Ménard et al., 2021]. This order of $O(1/x)$ is set to achieve the optimal sample complexity, and other works have also investigated into other forms of bonus function [Kaufmann et al., 2021]. Finally, in line 10 through line 12 the current policy $\pi^k$—which is the greedy policy with respect to $U^k$—is simulated on the empirical reference MDP $\widehat{\mathcal{M}}^{\dagger}$, and the virtual counters are updated. It is crucial to note that the simulation takes place entirely offline, by generating virtual transitions from $\widehat{\mathcal{M}}^{\dagger}$. Upon termination of algorithm 2, the uniform mixture of policies $\pi^1, \pi^2, \ldots$ form the non-reactive exploration policy $\pi_{ex}$, ensuring that the latter has wide 'coverage' over $\mathcal{M}^{\dagger}$.

## 4.4 Online and planning phase

Algorithm 2 implements line 1 of the procedure in algorithm 1 by finding the exploratory policy $\pi_{ex}$. After that, in line 2 of the interaction protocol the online dataset $\mathcal{D}'$ is generated by deploying $\pi_{ex}$ on the real MDP $\mathcal{M}$ to generate $K_{de}$ trajectories. Conceptually, the online dataset $\mathcal{D}'$ and the offline dataset $\mathcal{D}$ identify an updated empirical transition kernel $\widetilde{\mathbb{P}}$ and its sparsified version[2] $\widetilde{\mathbb{P}}^{\dagger}$. Finally, in

---

[2]For any $(s,a,s') \in \mathcal{S} \times \mathcal{A} \times \mathcal{S}$, we define $\widetilde{\mathbb{P}}^{\dagger}(s' \mid s,a) = \frac{m(s,a,s')}{m(s,a)}$ if $N(s,a,s') \geq \Phi$ and $\widetilde{\mathbb{P}}^{\dagger}(s' \mid s,a) = 0$ otherwise. Finally, for any $(s,a) \in \mathcal{S} \times \mathcal{A}$, we have $\widetilde{\mathbb{P}}^{\dagger}(s^{\dagger} \mid s,a) = \frac{1}{m(s,a)}\sum_{s' \in \mathcal{S}, N(s,a,s') < \Phi} m(s,a,s')$ and for any $a \in \mathcal{A}$, we have $\widetilde{\mathbb{P}}^{\dagger}(s^{\dagger} \mid s^{\dagger}, a) = 1$. Here $N(s,a,s')$ is the counter of initial offline data and $m(\cdot, \cdot)$ is the counter of online data.

line 3 a reward function $r$ is received, and the value iteration algorithm (See Appendix E) is invoked with $r$ as reward function and $\widetilde{\mathbb{P}}^\dagger$ as dynamics, and the near-optimal policy $\pi_{final}$ is produced. The use of the (updated) empirical sparsified dynamics $\widetilde{\mathbb{P}}^\dagger$ can be seen as incorporating the principle of pessimism under uncertainty due to the presence of the absorbing state.

Our complete algorithm is reported in algorithm 3, and it can be seen as implementing the interaction protocol described in algorithm 1.

---

**Algorithm 3** Reward-Free Non-reactive Policy Design (RF-NPD)

---

**Input:** Offline dataset $\mathcal{D}$, target suboptimality $\varepsilon > 0$, failure tolerance $\delta \in (0, 1]$.
 1: Construct the empirical sparsified MDP $\widehat{\mathcal{M}}^\dagger$.
 2: *Offline phase:* run RF-UCB$(\widehat{\mathcal{M}}^\dagger, K_{ucb}, \varepsilon, \delta)$ to obtain the exploratory policy $\pi_{ex}$.
 3: *Online phase:* deploy $\pi_{ex}$ on the MDP $\mathcal{M}$ for $K_{de}$ episodes to get the online dataset $\mathcal{D}'$.
 4: *Planning phase:* receive the reward function $r$, construct $\widetilde{\mathcal{M}}^\dagger$ from the online dataset $\mathcal{D}'$, compute $\pi_{final}$ (which is the optimal policy on $\widetilde{\mathcal{M}}^\dagger$) using value iteration (Appendix E).
**Output:** $\pi_{final}$.

---

## 5 Main Result

In this section, we present a performance bound on our algorithm, namely a bound on the suboptimality of the value of the final policy $\pi_{final}$ when measured on the sparsified MDP $\mathcal{M}^\dagger$. The sparsified MDP arises because it is generally not possible to directly compete with the optimal policy using a non-reactive data collection strategy and a polynomial number of samples due to the lower bound of Xiao et al. [2022]; more details are given in Appendix C.

In order to state the main result, we let $K = K_{ucb} = K_{de}$, where $K_{ucb}$ and $K_{de}$ are the number of episodes for the offline simulation and online interaction, respectively. Let $C$ be some universal constant, and choose the threshold in the definition of sparsified MDP as

$$\Phi = 6H^2 \log(12H |\mathcal{S}|^2 |\mathcal{A}| /\delta). \tag{5.1}$$

**Theorem 5.1.** *For any $\varepsilon > 0$ and $0 < \delta < 1$, if we let the number of online episodes be*

$$K = \frac{CH^2 |\mathcal{S}|^2 |\mathcal{A}|}{\varepsilon^2} \text{polylog}\left(|\mathcal{S}|, |\mathcal{A}|, H, \frac{1}{\varepsilon}, \frac{1}{\delta}\right),$$

*then with probability at least $1 - \delta$, for any reward function $r$, the final policy $\pi_{final}$ returned by Algorithm 3 satisfies the bound*

$$\max_{\pi \in \Pi} V_1\left(s_1; \mathbb{P}^\dagger, r^\dagger, \pi\right) - V_1\left(s_1; \mathbb{P}^\dagger, r^\dagger, \pi_{final}\right) \leq \varepsilon. \tag{5.2}$$

The theorem gives a performance guarantee on the value of the policy $\pi_{final}$, which depends both on the initial coverage of the offline dataset $\mathcal{D}$ as well as on the number of samples collected in the online phase. The dependence on the coverage of the offline dataset is implicit through the definition of the (population) sparsified $\mathcal{M}^\dagger$, which is determined by the counts $N(\cdot, \cdot)$.

In order to gain some intuition, we examine some special cases as a function of the coverage of the offline dataset.

**Empty dataset** Suppose that the offline dataset $\mathcal{D}$ is empty. Then the sparsified MDP identifies a *multi-armed bandit* at the initial state $s_1$, where any action $a$ taken from such state gives back the reward $r(s_1, a)$ and leads to the absorbing state $s^\dagger$. In this case, our algorithm essentially designs an allocation strategy $\pi_{ex}$ that is uniform across all actions at the starting state $s_1$. Given enough online samples, $\pi_{final}$ converges to the *action* with the highest instantaneous reward on the multi-armed bandit induced by the start state. With no coverage from the offline dataset, the lower bound of Xiao et al. [2022] for non-reactive policies precludes finding an $\varepsilon$-optimal policy on the original MDP $\mathcal{M}$ unless exponentially many samples are collected.

**Known connectivity graph** On the other extreme, assume that the offline dataset contains enough information everywhere in the state-action space such that the critical condition 4.1 is satisfied for

all $(s, a, s')$ tuples. Then the sparsified MDP and the real MDP coincide, i.e., $\mathcal{M} = \mathcal{M}^{\dagger}$, and so the final policy $\pi_{final}$ directly competes with the optimal policy $\pi^*$ for any given reward function in eq. (5.2). More precisely, the policy $\pi_{final}$ is $\varepsilon$-suboptimal on $\mathcal{M}$ if $\widetilde{O}(H^2 |\mathcal{S}|^2 |\mathcal{A}| / \varepsilon^2)$ trajectories are collected in the online phase, a result that matches the lower bound for reward-free exploration of Jin et al. [2020b] up to log factors. However, we achieve such result with a data collection strategy that is completely passive, one that is computed with the help of an initial offline dataset whose size $|\mathcal{D}| \approx \Phi \times |\mathcal{S}|^2 |\mathcal{A}| = \widetilde{O}(H^2 |\mathcal{S}|^2 |\mathcal{A}|)$ need *not depend on final accuracy* $\varepsilon$.

**Partial coverage** In more typical cases, the offline dataset has only *partial* coverage over the state-action space and the critical condition 4.1 may be violated in certain state-action-successor states. In this case, the connectivity graph of the sparsified MDP $\mathcal{M}^{\dagger}$ is a sub-graph of the original MDP $\mathcal{M}$ augmented with edges towards the absorbing state. The lack of coverage of the original dataset arises through the sparsified MDP in the guarantees that we present in theorem 5.1. In this section, we 'translate' such guarantees into guarantees on $\mathcal{M}$, in which case the 'lack of coverage' is naturally represented by the concentrability coefficient

$$C^* = \sup_{s,a} d_\pi(s,a)/\mu(s,a),$$

see for examples the papers [Munos and Szepesvári, 2008, Chen and Jiang, 2019] for background material on the concentrability factor. More precisely, we compute the sample complexity—in terms of online as well as offline samples—required for $\pi_{final}$ to be $\varepsilon$-suboptimal with respect to any comparator policy $\pi$, and so in particular with respect to the optimal policy $\pi_*$ on the "real" MDP $\mathcal{M}$. The next corollary is proved in appendix B.3.

**Corollary 5.2.** *Suppose that the offline dataset contains*

$$\widetilde{O}\Big(\frac{H^4 |\mathcal{S}|^2 |\mathcal{A}| C^*}{\varepsilon}\Big),$$

*samples and that additional*

$$\widetilde{O}\Big(\frac{H^3 |\mathcal{S}|^2 |\mathcal{A}|}{\varepsilon^2}\Big)$$

*online samples are collected during the online phase. Then with probability at least $1 - \delta$, for any reward function $r$, the policy $\pi_{final}$ is $\epsilon$-suboptimal with respect to any comparator policy $\pi$*

$$V_1\left(s_1; \mathbb{P}, r, \pi\right) - V_1\left(s_1; \mathbb{P}, r, \pi_{final}\right) \leq \varepsilon. \tag{5.3}$$

The online sample size is equivalent to the one that arises in the statement of theorem 5.1 (expressed as number of online trajectories), and does not depend on the concentrability coefficient. The dependence on the offline dataset in theorem 5.1 is implicit in the definition of sparsified MDP; here we have made it explicit using the notion of concentrability.

Corollary 5.2 can be used to compare the achievable guarantees of our procedure with that of an offline algorithm, such as the minimax-optimal procedure detailed in [Xie et al., 2021b]. The proceedure described in [Xie et al., 2021b] achieves (5.3) with probability at least $1 - \delta$ by using

$$\widetilde{O}\left(\frac{H^3 |\mathcal{S}| C^*}{\varepsilon^2} + \frac{H^{5.5} |\mathcal{S}| C^*}{\varepsilon}\right) \tag{5.4}$$

offline samples[3]. In terms of offline data, our procedure has a similar dependence on various factors, but it depends on the desired accuracy $\varepsilon$ through $\widetilde{O}(1/\varepsilon)$ as opposed to $\widetilde{O}(1/\varepsilon^2)$ which is typical for an offline algorithm. This implies that in the small-$\varepsilon$ regime, if sufficient online samples are collected, one can improve upon a fully offline procedure by collecting a number of additional online samples in a non-reactive way.

Finally, notice that one may improve upon an offline dataset by collecting more data from the distribution $\mu$, i.e., without performing experimental design. Compared to this latter case, notice that our *online sample complexity does not depend on the concentrability coefficient*. Further discussion can be found in appendix B.

---

[3]Technically, [Zhan et al., 2022] considers the non-homogeneous setting, and expresses their result in terms of number of trajectories. In obtaining eq. (5.4), we 'removed' an $H$ factor due to our dynamics being homogeneous, and add it back to express the result in terms of number of samples. However, notice that [Zhan et al., 2022] consider the reward-aware setting, which is simpler than reward-free RL setting that we consider. This should add an additional $|\mathcal{S}|$ factor that is not accounted for in eq. (5.4), see the paper Jin et al. [2020b] for more details.

# 6 Proof

In this section we prove theorem 5.1, and defer the proofs of the supporting statements to the Appendix A.

Let us define the comparator policy $\pi_*^\dagger$ used for the comparison in eq. (5.2) to be the (deterministic) policy with the highest value function on the sparsified MDP:

$$\pi_*^\dagger := \arg\max_{\pi \in \Pi} V_1(s_1; \mathbb{P}^\dagger, r^\dagger, \pi).$$

We can bound the suboptimality using the triangle inequality as

$$V_1\left(s_1; \mathbb{P}^\dagger, r^\dagger, \pi_*^\dagger\right) - V_1\left(s_1; \mathbb{P}^\dagger, r^\dagger, \pi_{final}\right) \leq \left|V_1\left(s_1; \mathbb{P}^\dagger, r^\dagger, \pi_*^\dagger\right) - V_1\left(s_1; \widetilde{\mathbb{P}}^\dagger, r^\dagger, \pi_*^\dagger\right)\right|$$

$$+ \underbrace{V_1\left(s_1; \widetilde{\mathbb{P}}^\dagger, r^\dagger, \pi_*^\dagger\right) - V_1\left(s_1; \widetilde{\mathbb{P}}^\dagger, r^\dagger, \pi_{final}\right)}_{\leq 0} + \left|V_1\left(s_1; \widetilde{\mathbb{P}}^\dagger, r^\dagger, \pi_{final} - V_1\left(s_1; \mathbb{P}^\dagger, r^\dagger, \pi_{final}\right)\right)\right|$$

$$\leq 2 \sup_{\pi \in \Pi, r} \left|V_1\left(s_1; \mathbb{P}^\dagger, r^\dagger, \pi\right) - V_1\left(s_1; \widetilde{\mathbb{P}}^\dagger, r^\dagger, \pi\right)\right|.$$

The middle term after the first inequality is negative due to the optimality of $\pi_{final}$ on $\widetilde{\mathbb{P}}^\dagger$ and $r^\dagger$. It suffices to prove that for any arbitrary policy $\pi$ and reward function $r$ the following statement holds with probability at least $1 - \delta$

$$\underbrace{\left|V_1\left(s_1; \mathbb{P}^\dagger, r^\dagger, \pi\right) - V_1\left(s_1; \widetilde{\mathbb{P}}^\dagger, r^\dagger, \pi\right)\right|}_{\text{Estimation error}} \leq \frac{\varepsilon}{2}. \tag{6.1}$$

**Bounding the estimation error using the population uncertainty function** In order to prove eq. (6.1), we first define the population *uncertainty function* $X$, which is a scalar function over the state-action space. It represents the maximum estimation error on the value of any policy when it is evaluated on $\widetilde{\mathcal{M}}^\dagger$ instead of $\mathcal{M}$. For any $(s, a) \in \mathcal{S} \times \mathcal{A}$, the uncertainty function is defined as $X_{H+1}(s, a) := 0$ and for $h \in [H]$,

$$X_h(s, a) := \min \left\{ H - h + 1; \ 9H\phi(m(s, a)) + \left(1 + \frac{1}{H}\right) \sum_{s'} \widetilde{\mathbb{P}}^\dagger(s' \mid s, a) \left(\max_{a'} \{X_{h+1}(s', a')\}\right) \right\}.$$

We extend the definition to the absorbing state by letting $X_h(s^\dagger, a) = 0$ for any $h \in [H], a \in \mathcal{A}$. The summation $\sum_{s'}$ used above is over $s' \in \mathcal{S} \cup \{s^\dagger\}$, but since $X_h(s^\dagger, a) = 0$ for any $h \in [H], a \in \mathcal{A}$, it is equivalent to that over $s' \in \mathcal{S}$. Intuitively, $X_h(s, a)$ takes a similar form as Bellman optimality equation. The additional $(1 + 1/H)$ factor and additional term $9H\phi(m(s, a))$ quantify the uncertainty of the true Q function on the sparsifed MDP and $9H\phi(m(s, a))$ will converge to zero when the sample size goes to infinity. This definition of uncertainty function and the following lemma follow closely from the uncertainty function defined in [Ménard et al., 2021].

The next lemma highlights the key property of the uncertainty function $X$, namely that for any reward function and any policy $\pi$, we can upper bound the estimation error via the uncertainty function at the initial times-step; it is proved in appendix A.2.1.

**Lemma 6.1.** *With probability $1 - \delta$, for any reward function $r$ and any deterministic policy $\pi$, it holds that*

$$|V_1(s_1; \widetilde{\mathbb{P}}^\dagger, r^\dagger, \pi) - V_1(s_1; \mathbb{P}^\dagger, r^\dagger, \pi)| \leq \max_a X_1(s_1, a) + C\sqrt{\max_a X_1(s_1, a)}. \tag{6.2}$$

The uncertainty function $X$ contains the inverse number of *online* samples $1/m(s, a)$ through $\phi(m(s, a))$, and so lemma 6.1 expresses the estimation error in eq. (6.1) as the maximum expected size of the confidence intervals $\sup_\pi \mathbb{E}_{\widetilde{\mathbb{P}}^\dagger, (s,a) \sim \pi} \sqrt{1/m(s, a)}$, a quantity that directly depends on the number $m(\cdot, \cdot)$ of samples collected during the online phase.

**Leveraging the exploration mechanics**   Throughout this section, $C$ denotes some universal constant and may vary from line to line. Recall that the agent greedily minimizes the *empirical uncertainty function $U$* to compute the exploratory policy $\pi_{ex}$. The empirical uncertainty is defined as $U_{H+1}^k(s,a) = 0$ for any $k, s \in \mathcal{S}, a \in \mathcal{A}$ and

$$U_h^k(s,a) = H \min\{1, \phi(n^k(s,a))\} + \widehat{\mathbb{P}}^\dagger(s,a)^\top (\max_{a'} U_{h+1}^k(\cdot, a')), \tag{6.3}$$

where $n^k(s,a)$ is the counter of the times we encounter $(s,a)$ until the beginning of the $k$-the virtual episode in the simulation phase. Note that, $U_h^k(s,a)$ takes a similar form as $X_h(s,a)$, except that $U_h^k(s,a)$ depends on the empirical transition probability $\widehat{\mathbb{P}}^\dagger$ while $X_h(s,a)$ depends on the true transition probability on the sparsified MDP. For the exploration scheme to be effective, $X$ and $U$ should be close in value, a concept which is at the core of this work and which we formally state below and prove in appendix A.2.2.

**Lemma 6.2** (Bounding uncertainty function with empirical uncertainty functions)**.** *With probability at least $1 - \delta$, we have for any $(h, s, a) \in [H] \times \mathcal{S} \times \mathcal{A}$,*

$$X_h(s,a) \leq \frac{C}{K} \sum_{k=1}^K U_h^k(s,a).$$

Notice that $X_h$ is the population uncertainty after the online samples have been collected, while $U_h^k$ is the corresponding empirical uncertainty which varies during the planning phase.

**Rate of decrease of the estimation error**   Combining lemmas 6.1 and 6.2 shows that (a function of) the agent's uncertainty estimate $U$ upper bounds the estimation error in eq. (6.1). In order to conclude, we need to show that $U$ decreases on average at the rate $1/K$, a statement that we present below and prove in appendix A.2.3.

**Lemma 6.3.** *With probability at least $1 - \delta$, we have*

$$\frac{1}{K} \sum_{k=1}^K U_1^k(s,a) \leq \frac{H^2 |\mathcal{S}|^2 |\mathcal{A}|}{K} \text{polylog}\left(K, |\mathcal{S}|, |\mathcal{A}|, H, \frac{1}{\varepsilon}, \frac{1}{\delta}\right). \tag{6.4}$$

*Then, for any $\varepsilon > 0$, if we take*

$$K := \frac{CH^2 |\mathcal{S}| |\mathcal{A}|}{\varepsilon^2} \left(\iota + |\mathcal{S}|\right) \text{polylog}\left(|\mathcal{S}|, |\mathcal{A}|, H, \frac{1}{\varepsilon}, \frac{1}{\delta}\right),$$

*then with probability at least $1 - \delta$, it holds that*

$$\frac{1}{K} \sum_{k=1}^K U_1^k(s_1, a) \leq \varepsilon^2.$$

After combining lemmas 6.1 to 6.3, we see that the estimation error can be bounded as

$$\left| V_1\left(s_1; \mathbb{P}^\dagger, r^\dagger, \pi\right) - V_1\left(s_1; \widetilde{\mathbb{P}}^\dagger, r^\dagger, \pi\right) \right| \leq \max_a X_1(s_1, a) + C\sqrt{\max_a X_1(s_1, a)}$$

$$\leq C \max_a \left[ \sqrt{\frac{1}{K} \sum_{k=1}^K U_1^k(s_1, a)} + \frac{1}{K} \sum_{k=1}^K U_1^k(s_1, a) \right]$$

$$\leq C \left(\varepsilon + \varepsilon^2\right)$$

$$\leq C\varepsilon \qquad \qquad \text{(for } 0 < \varepsilon < const)$$

Here, the constant $C$ may vary between lines. Rescaling the universal constant $C$ and the failure probability $\delta$, we complete the upper bound in equation (6.1) and hence the proof for the main result.

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

# Contents

# A Proof of the main result

## A.1 Definitions

In this section, we define some crucial concepts that will be used in the proof of the main result.

### A.1.1 Sparsified MDP

First, we restate the definition 4.1 in the main text.

**Definition A.1** (Sparsified MDP). *Let $s^\dagger$ be an absorbing state, i.e., such that $\mathbb{P}\left(s^\dagger \mid s^\dagger, a\right) = 1$ and $r(s^\dagger, a) = 0$ for all $a \in \mathcal{A}$. The state space in the sparsified MDP $\mathcal{M}^\dagger$ is defined as that of the original MDP with the addition of $s^\dagger$. The dynamics $\mathbb{P}^\dagger$ of the sparsified MDP are defined as*

$$\mathbb{P}^\dagger(s' \mid s, a) = \begin{cases} \mathbb{P}(s' \mid s, a) & \text{if } N(s, a, s') \geq \Phi \\ 0 & \text{if } N(s, a, s') < \Phi, \end{cases} \qquad \mathbb{P}^\dagger(s^\dagger \mid s, a) = \sum_{\substack{s' \neq s^\dagger \\ N(s,a,s')<\Phi}} \mathbb{P}(s' \mid s, a). \tag{A.1}$$

*For any deterministic reward function $r : \mathcal{S} \times \mathcal{A} \to [0, 1]$, the reward function on the sparsified MDP is defined as $r^\dagger(s, a) = r(s, a)$; for simplicity we only consider deterministic reward functions.*

In the offline phase of our algorithm, we simulate the virtual episodes from the empirical version of sparsified MDP (See Algorithm 2). Now we formally this MDP.

**Definition A.2** (Emprical sparsified MDP). *Let $s^\dagger$ be the absorbing state defined in the sparsified MDP. The state space in the empirical sparsified MDP $\widehat{\mathcal{M}}^\dagger$ is defined as that of the original MDP with the addition of $s^\dagger$. The dynamics $\widehat{\mathbb{P}}^\dagger$ of the sparsified MDP are defined as*

$$\widehat{\mathbb{P}}^\dagger(s' \mid s, a) = \begin{cases} \frac{N(s,a,s')}{N(s,a)} & \text{if } N(s, a, s') \geq \Phi \\ 0 & \text{if } N(s, a, s') < \Phi, \end{cases} \qquad \widehat{\mathbb{P}}^\dagger(s^\dagger \mid s, a) = \sum_{\substack{s' \neq s^\dagger \\ N(s,a,s')<\Phi}} \frac{N(s,a,s')}{N(s,a)}. \tag{A.2}$$

*For any deterministic reward function $r : \mathcal{S} \times \mathcal{A} \to [0, 1]$, the reward function on the empirical sparsified MDP is defined as $r^\dagger(s, a) = r(s, a)$; for simplicity we only consider deterministic reward functions. Here, the counters $N(s, a)$ and $N(s, a, s')$ are the number of $(s, a)$ and $(s, a, s')$ in the offline data, respectively.*

Finally, in the planning phase, after we interact with the true environment for many online episodes, construct a fine-estimated sparsified MDP, which is used to extract the optmal policy of given reward functions. We formally define it below.

**Definition A.3** (Fine-estimated sparsified MDP). *Let $s^\dagger$ be the absorbing state defined in the sparsified MDP. The state space in the fine-estimated sparsified MDP $\widetilde{\mathcal{M}}^\dagger$ is defined as that of the original MDP with the addition of $s^\dagger$. The dynamics $\widetilde{\mathbb{P}}^\dagger$ of the sparsified MDP are defined as*

$$\widetilde{\mathbb{P}}^\dagger(s' \mid s, a) = \begin{cases} \frac{m(s,a,s')}{m(s,a)} & \text{if } N(s, a, s') \geq \Phi \\ 0 & \text{if } N(s, a, s') < \Phi, \end{cases} \qquad \widetilde{\mathbb{P}}^\dagger(s^\dagger \mid s, a) = \sum_{\substack{s' \neq s^\dagger \\ N(s,a,s')<\Phi}} \frac{m(s,a,s')}{m(s,a)}. \tag{A.3}$$

*For any deterministic reward function $r : \mathcal{S} \times \mathcal{A} \to [0, 1]$, the reward function on the fine-estimated sparsified MDP is defined as $r^\dagger(s, a) = r(s, a)$; for simplicity we only consider deterministic reward functions. Here, the counters $N(s, a)$ and $N(s, a, s')$ are the number of $(s, a)$ and $(s, a, s')$ in the offline data, respectively, while $m(s, a)$ and $m(s, a, s')$ are the counters of $(s, a)$ and $(s, a, s')$ in online episodes.*

### A.1.2 High probability events

In this section, we define all high-probability events we need in order to make our theorem to hold. Specifically, we define

$$\mathcal{E}^P := \left\{ \forall (s,a,s') \text{ s.t. } N(s,a,s') \geq \Phi, \left| \widehat{\mathbb{P}}^\dagger(s' \mid s,a) - \mathbb{P}^\dagger(s' \mid s,a) \right| \right.$$

$$\left. \leq \sqrt{\frac{2\widehat{\mathbb{P}}^\dagger(s' \mid s,a)}{N(s,a)} \log\left(\frac{12 |\mathcal{S}|^2 |\mathcal{A}|}{\delta}\right)} + \frac{14}{3N(s,a)} \log\left(\frac{12 |\mathcal{S}|^2 |\mathcal{A}|}{\delta}\right) \right\} ;$$

$$\mathcal{E}^2 := \left\{ \forall k \in \mathbb{N}, (s,a) \in \mathcal{S} \times \mathcal{A}, n^k(s,a) \geq \frac{1}{2} \sum_{i<k} \sum_{h \in [H]} \widehat{d}^\dagger_{\pi^i,h}(s,a) - H \ln\left(\frac{6H |\mathcal{S}| |\mathcal{A}|}{\delta}\right) \right\} ;$$

$$\mathcal{E}^3 := \left\{ \forall k \in \mathbb{N}, (s,a) \in \mathcal{S} \times \mathcal{A}, n^k(s,a) \leq 2 \sum_{i<k} \sum_{h \in [H]} \widehat{d}^\dagger_{\pi^i,h}(s,a) + H \ln\left(\frac{6H |\mathcal{S}| |\mathcal{A}|}{\delta}\right) \right\} ;$$

$$\mathcal{E}^4 := \left\{ \forall (s,a) \in \mathcal{S} \times \mathcal{A}, \text{KL}\left(\widetilde{\mathbb{P}}^\dagger(s,a); \mathbb{P}^\dagger(s,a)\right) \right.$$

$$\left. \leq \frac{1}{m(s,a)} \left[ \log\left(\frac{6 |\mathcal{S}| |\mathcal{A}|}{\delta}\right) + |\mathcal{S}| \log\left(e\left(1 + \frac{m(s,a)}{|\mathcal{S}|}\right)\right) \right] \right\} ;$$

$$\mathcal{E}^5 := \left\{ \forall (s,a) \in [H] \times \mathcal{S} \times \mathcal{A}, m(s,a) \geq \frac{1}{2} K_{de} \sum_{h \in [H]} w_h^{mix}(s,a) - H \ln\left(\frac{6H |\mathcal{S}| |\mathcal{A}|}{\delta}\right) \right\} ,$$

where

$$w_h^{mix}(s,a) := \frac{1}{K_{ucb}} \sum_{k=1}^{K_{ucb}} d^\dagger_{\pi^k,h}(s,a).$$

Here, $\text{KL}(\cdot ; \cdot)$ denotes the Kullback–Leibler divergence between two distributions. $K_{ucb}$ is the number of episodes of the sub-routine RF-UCB and $\pi^i$ is the policy executed at the i-th episode of RF-UCB (See algorithm 2). $K_{de}$ is the number of episodes executed in the online phase. $n^k(s,a)$ is the counter of $(s,a)$ before the beginning of the $k$-th episode in the offline phase and $m(s,a)$ is the number of $(s,a)$ samples in the online data (See definitions in section A.1). We denote $d_{\pi,h}(s,a)$, $d^\dagger_{\pi,h}(s,a)$ and $\widehat{d}^\dagger_{\pi,h}(s,a)$ as the occupancy measure of $(s,a)$ at stage $h$ under policy $\pi$, on $\mathbb{P}$ (the true transition dynamics), $\mathbb{P}^\dagger$ (the transition dynamics in the sparsfied MDP) and $\widehat{\mathbb{P}}^\dagger$(the transition dynamics in the empirical sparsified MDP) respectively.

For the event defined above, we have the following guarantee, which is proved in appendix A.3.1.

**Lemma A.4.** *For $\delta > 0$, we have*

$$\mathcal{E} := \left\{ \mathcal{E}^P \cup \mathcal{E}^2 \cup \mathcal{E}^3 \cup \mathcal{E}^4 \cup \mathcal{E}^5 \right\}$$

*happens with probability at least $1 - \delta$,*

### A.1.3 Uncertainty function and empirical uncertainty function

In this section, we restate the definition of uncertainty function and empirical uncertainty functions, as well as some intermediate uncertainty functions which will be often used in the proof. First, we restate the definition of bonus function.

**Definition A.5** (Bonus Function). *We define*

$$\phi(x) = \frac{H}{x} \left[ \log\left( \frac{6H\,|\mathcal{S}|\,|\mathcal{A}|}{\delta} \right) + |\mathcal{S}| \log\left( e\left( 1 + \frac{x}{|\mathcal{S}|} \right) \right) \right]. \tag{A.4}$$

*Further, we define*

$$\overline{\phi}(x) = \min\{1, H\phi(x)\}. \tag{A.5}$$

Next, we restate the definition of uncertainty function and empirical uncertainty function. Notice that the uncertainty function does not depend on reward function or specific policy $\pi$.

**Definition A.6** (Uncertainty function). *We define for any $(h, s, a) \in [H] \times \mathcal{S} \times \mathcal{A}$ and any deterministic policy $\pi$,*

$$X_{H+1}(s, a) := 0,$$

$$X_h(s, a) := \min\left\{ H - h + 1, 9H\phi(m(s, a)) + \left( 1 + \frac{1}{H} \right) \widetilde{\mathbb{P}}^\dagger(\cdot \mid s, a)^\top \left( \max_{a'} X_{h+1}(\cdot, a') \right) \right\},$$

*where we specify*

$$\widetilde{\mathbb{P}}^\dagger(\cdot \mid s, a)^\top \left( \max_{a'} \{ X_{h+1}(\cdot, a') \} \right) := \sum_{s'} \widetilde{\mathbb{P}}^\dagger(s' \mid s, a) \left( \max_{a'} \{ X_{h+1}(s', a') \} \right).$$

*In addition, we define $X_h(s^\dagger, a) = 0$ for any $h \in [H], a \in \mathcal{A}$. Here, the notation $\sum_{s'}$ above means summation over $s' \in \mathcal{S} \cup \{s^\dagger\}$. But since $X_h(s^\dagger, a) = 0$ for any $h \in [H], a \in \mathcal{A}$, this is equivalent to the summation over $s' \in \mathcal{S}$. $\widetilde{\mathbb{P}}^\dagger$ is the transition probability of fine-estimated sparsified MDP defined in section A.1.1.*

Then, for the reader to better understand the proof, we define some intermediate quantity, which also measure the uncertainty of value estimation, but they may be reward or policy dependent.

**Definition A.7** (Intermediate uncertainty function). *We define for any $(h, s, a) \in [H] \times \mathcal{S} \times \mathcal{A}$ and any deterministc policy $\pi$,*

$$W_{H+1}^\pi(s, a, r) := 0,$$

$$W_h^\pi(s, a, r) := \min\left\{ H - h + 1, \sqrt{\frac{8}{H^2} \overline{\phi}(m(s, a)) \cdot \mathrm{Var}_{s' \sim \widetilde{\mathbb{P}}^\dagger(s, a)} \left( V_{h+1}\left( s'; \widetilde{\mathbb{P}}^\dagger, r^\dagger, \pi \right) \right)} \right.$$

$$\left. + 9H\phi(m(s, a)) + \left( 1 + \frac{1}{H} \right) \left( \widetilde{\mathbb{P}}^\dagger \pi_{h+1} \right) W_{h+1}^\pi(s, a, r) \right\},$$

*where*

$$\left( \widetilde{\mathbb{P}}^\dagger \pi_{h+1} \right) W_{h+1}^\pi(s, a, r) := \sum_{s'} \sum_{a'} \widetilde{\mathbb{P}}^\dagger(s' \mid s, a) \pi_{h+1}(a' \mid s') W_{h+1}^\pi(s', a', r).$$

*In addition, we define $W_h^\pi(s^\dagger, a, r) = 0$ for any $h \in [H], a \in \mathcal{A}$ and any reward function $r$. Here, $\widetilde{\mathbb{P}}^\dagger$ is the transition probability of fine-estimated sparsified MDP defined in section A.1.1.*

The intermediate function $W$ is reward and policy dependent and can be used to upper bound the value estimation error. Further, we need to define some policy-dependent version of the uncertainty function, denoted as $X_h^\pi(s, a)$, as well as another quantity $Y_h^\pi(s, a, r)$.

**Definition A.8.** *We define $X_{H+1}^\pi(s, a) = Y_{H+1}^\pi(s, a, r) = 0$ for any $\pi, r, s, a$ and*

$$X_h^\pi(s, a) := \min\left\{ H - h + 1, 9H\phi(m(s, a)) + \left( 1 + \frac{1}{H} \right) \left( \widetilde{\mathbb{P}}^\dagger \pi_{h+1} \right) X_{h+1}^\pi(s, a) \right\};$$

$$Y_h^\pi(s, a, r) := \sqrt{\frac{8}{H^2} \overline{\phi}(m(s,a)) \operatorname{Var}_{s' \sim \widetilde{\mathbb{P}}^\dagger(s,a)} \left(V_{h+1}\left(s'; \widetilde{\mathbb{P}}^\dagger, r^\dagger, \pi\right)\right)} + \left(1 + \frac{1}{H}\right)\left(\widetilde{\mathbb{P}}^\dagger \pi_{h+1}\right) Y_{h+1}^\pi(s, a, r)$$

*where*

$$\left(\widetilde{\mathbb{P}}^\dagger \pi_{h+1}\right) X_{h+1}^\pi(s, a) = \sum_{s'} \sum_{a'} \widetilde{\mathbb{P}}^\dagger\left(s' \mid s, a\right) \pi_{h+1}\left(a' \mid s'\right) X_{h+1}^\pi\left(s', a'\right);$$

$$\left(\widetilde{\mathbb{P}}^\dagger \pi_{h+1}\right) Y_{h+1}^\pi(s, a, r) = \sum_{s'} \sum_{a'} \widetilde{\mathbb{P}}^\dagger\left(s' \mid s, a\right) \pi_{h+1}\left(a' \mid s'\right) Y_{h+1}^\pi\left(s', a', r\right).$$

*In addition, we define $X_h^\pi(s^\dagger, a) = Y_h^\pi(s^\dagger, a, r) = 0$ for any $a \in \mathcal{A}, h \in [H]$ and any reward $r$, any policy $\pi$. Here, $\widetilde{\mathbb{P}}^\dagger$ is the transition probability of fine-estimated sparsified MDP defined in section A.1.1.*

Finally, we define the empirical uncertainty functions.

**Definition A.9.** *We define $U_{H+1}^k(s, a) = 0$ for any $k \in [K_{ucb}]$ and $s \in \mathcal{S}, a \in \mathcal{A}$. Further, we define*

$$U_h^k(s, a) = H \min\left\{1, \phi\left(n^k(s, a)\right)\right\} + \widehat{\mathbb{P}}^\dagger(s, a)^\top \left(\max_{a'} U_{h+1}^k(\cdot, a')\right), \tag{A.6}$$

*where $\phi$ is the bonus function defined in eq. (A.4) and $n^k(s, a)$ is the counter of the times we encounter $(s, a)$ until the beginning of the $k$-the episode when running the sub-routine RF-UCB in the offline phase.*

## A.2 Proof of the key theorems and lemmas

### A.2.1 Uncertainty Functions upper bounds the estimation error

*Proof.* This proof follows closely from the techniques in [Ménard et al., 2021], but here we consider the homogeneous MDP. We let $\widetilde{d}_{\pi,h}^\dagger(s, a)$ be the probability of encountering $(s, a)$ at stage $h$ when running policy $\pi$ on $\widetilde{\mathbb{P}}^\dagger$, starting from $s_1$. Now we assume $\mathcal{E}$ happens and fix a reward function $r$ and policy $\pi$. From lemma A.14, we know

$$\left|V_1\left(s_1; \widetilde{\mathbb{P}}^\dagger, r^\dagger, \pi\right) - V_1\left(s_1; \mathbb{P}^\dagger, r^\dagger, \pi\right)\right| \le \left|Q_h\left(s_1, \pi_1(a_1); \widetilde{\mathbb{P}}^\dagger, r^\dagger, \pi\right) - Q_h\left(s_1, \pi_1(s_1); \mathbb{P}^\dagger, r^\dagger, \pi\right)\right|$$
$$\le W_h^\pi(s_1, \pi(s_1), r),$$

where $W_h^\pi(s, a, r)$ is the intermediate uncertainty function defined in definition A.7. Here, we use the policy-dependent version of uncertainty function $W^\pi(s, a, r)$, which will then upper bounded using another two policy-dependent quantities $X_h^\pi(s, a)$ and $Y_h^\pi(s, a, r)$. We define these policy-dependent uncertainty functions to upper bound the estimation error of specific policy, but since we are considering a reward-free setting, which entails a low estimation error for any policy and reward, these quantities cannot be directly used in the algorithm to update the policy. Next, we claim

$$W_h^\pi(s, a, r) \le X_h^\pi(s, a) + Y_h^\pi(s, a, r),$$

where $X_h^\pi(s, a)$ and $Y_h^\pi(s, a, r)$ are defined in definition A.8. Actually, this is easy from the definition of $X_h^\pi(s, a), Y_h^\pi(s, a, r)$ and $W_h^\pi(s, a, r)$. For $h = H + 1$, this is trivial. Assume this is true for $h + 1$, then the case of $h$ is given by the definition and the fact that $\min\{x, y + z\} \le \min\{x, y\} + z$ for any $x, y, z \ge 0$. Therefore, we have

$$\left|V_1\left(s_1; \widetilde{\mathbb{P}}^\dagger, r^\dagger, \pi\right) - V_1\left(s_1; \mathbb{P}^\dagger, r^\dagger, \pi\right)\right| \le X_1^\pi(s_1, \pi_1(s_1)) + Y_1^\pi(s_1, \pi_1(s_1), r). \tag{A.7}$$

Next, we eliminate the dependency to policy $\pi$ and obtain an upper bound of estimation error using the policy-independent uncertainty function $X_h(s, a)$. This is done by bounding $Y_1^\pi(s_1, \pi_1(s_1), r)$ by $X_h^\pi(s, a)$ and then upper bounding $X_h^\pi(s, a)$ by $X_h(s, a)$. From definition A.8, the Cauchy-Schwarz inequality and the fact that $r \in [0, 1]$, we have for any reward function and any deterministic policy $\pi$,

$$Y_1^\pi(s_1, \pi_1(s_1), r)$$

$$\leq \sum_{s,a} \sum_{h=1}^{H} \widetilde{d}_{\pi,h}^{\dagger}(s,a) \left(1 + \frac{1}{H}\right)^{h-1} \sqrt{\frac{8}{H^2}\overline{\phi}\left(m(s,a)\right) \cdot \mathrm{Var}_{s'\sim \widetilde{\mathbb{P}}^{\dagger}(s,a)}\left(V_{h+1}\left(s'; \widetilde{\mathbb{P}}^{\dagger}, r^{\dagger}, \pi\right)\right)}$$

(Induction by successively using the definition of $Y$)

$$\leq \frac{e}{H} \sqrt{8 \sum_{s,a}\sum_{h=1}^{H} \widetilde{d}_{\pi,h}^{\dagger}(s,a)\,\mathrm{Var}_{s'\sim\widetilde{\mathbb{P}}^{\dagger}(s,a)}\left(V_{h+1}\left(s';\widetilde{\mathbb{P}}^{\dagger}, r^{\dagger}, \pi\right)\right)} \cdot \sqrt{\sum_{s,a}\sum_{h=1}^{H} \widetilde{d}_{\pi,h}^{\dagger}(s,a)\overline{\phi}\left(m(s,a)\right)}$$

(Cauchy-Schwarz Inequality)

$$\leq \frac{e}{H}\sqrt{8H^2 \sum_{s,a}\sum_{h=1}^{H}\widetilde{d}_{\pi,h}^{\dagger}(s,a)} \cdot \sqrt{\sum_{s,a}\sum_{h=1}^{H}\widetilde{d}_{\pi,h}^{\dagger}(s,a)\overline{\phi}\left(m(s,a)\right)}$$

$$\leq e\sqrt{8\sum_{s,a}\sum_{h=1}^{H}\widetilde{d}_{\pi,h}^{\dagger}(s,a)\overline{\phi}\left(m(s,a)\right)}.$$

We notice that the right hand side of the last inequality above is the policy value of some specific reward function when running $\pi$ on $\widetilde{\mathbb{P}}$. Concretely, if the transition probability is $\widetilde{\mathbb{P}}$ and the reward function at $(s,a)$ is $\overline{\phi}\left(m(s,a)\right)$, then the state value function at the initial state $s_1$ is $\sum_{s,a}\sum_{h=1}^{H}\widetilde{d}_{\pi,h}^{\dagger}(s,a)\overline{\phi}\left(m(s,a)\right)$. This specific reward function is non-negative and uniformly bounded by one, so it holds that $\sum_{s,a}\sum_{i=h}^{H}\widetilde{d}_{\pi,i}^{\dagger}(s,a)\overline{\phi}\left(m(s,a)\right) \leq H - h + 1$. Moreover, from the definition of $\phi$ and $\overline{\phi}$ (eq. (A.4)), we know $\overline{\phi}\left(m(s,a)\right) \leq H\phi\left(m(s,a)\right)$. Then, from the definition of $X_h(s,a)$ in definition A.6, we apply an inductive argument to obtain

$$\sum_{s,a}\sum_{h=1}^{H}\widetilde{d}_{\pi,h}^{\dagger}(s,a)\overline{\phi}\left(m(s,a)\right) \leq X_1^{\pi}(s_1, \pi(s_1))$$

for any deterministic policy $\pi$. So we have

$$Y_1^{\pi}(s_1, \pi_1(s_1), r) \leq 2\sqrt{2e}\sqrt{X_1^{\pi}(s_1, \pi(s_1))}.$$

Therefore, combining the last inequality with (A.7), we have

$$\left|V_1\left(s_1; \widetilde{\mathbb{P}}^{\dagger}, r^{\dagger}, \pi\right) - V_1\left(s_1; \mathbb{P}^{\dagger}, r^{\dagger}, \pi\right)\right| \leq X_1^{\pi}(s_1, \pi_1(s_1)) + 2\sqrt{2e}\sqrt{X_1^{\pi}(s_1, \pi_1(s_1))}.$$

From the definition of $X_h(s,a)$ (definition A.6) and $X_h^{\pi}(s,a)$ (definition A.8), we can see

$$X_h^{\pi}(s,a) \leq X_h(s,a)$$

for any $(h,s,a)$ and any deterministic policy $\pi$. Therefore, we conclude that

$$\left|V_1\left(s_1; \widetilde{\mathbb{P}}^{\dagger}, r^{\dagger}, \pi\right) - V_1\left(s_1; \mathbb{P}^{\dagger}, r^{\dagger}, \pi\right)\right| \leq X_1(s_1, \pi_1(s_1)) + 2\sqrt{2e}\sqrt{X_1(s_1, \pi_1(s_1))}$$

$$\leq \max_a X_1(s_1, a) + 2\sqrt{2e}\sqrt{\max_a X_1(s_1, a)}.$$

□

### A.2.2 Proof of lemma 6.2

*Proof.* It suffices to prove that for any $k \in [K], h \in [H], s \in \mathcal{S}, a \in \mathcal{A}$, it holds that

$$X_h(s,a) \leq C\left(1 + \frac{1}{H}\right)^{3(H-h)} U_h^k(s,a)$$

since the theorem comes from an average of the inequalities above and the fact that $(1 + 1/H)^H \leq e$. For any fixed $k$, we prove it by induction on $h$. When $h = H + 1$, both sides are zero by definition. Suppose the claim holds for $h + 1$; we will prove that it also holds for $h$. We denote $K_{ucb}$ as the

number of virtual episodes in the offline phase. From lemma A.16, the decreasing property of $\phi(\cdot)$(lemma A.17) and lemma A.21, we have

$$X_h(s,a) \leq CH\phi\left(n^{K_{ucb}}(s,a)\right) + \left(1 + \frac{2}{H}\right)\mathbb{P}^\dagger(\cdot \mid s,a)^\top \left(\max_{a'}\{X_{h+1}(\cdot,a')\}\right) \quad \text{(Lemma A.16)}$$

$$\leq CH\phi\left(n^k(s,a)\right) + \left(1 + \frac{2}{H}\right)\mathbb{P}^\dagger(\cdot \mid s,a)^\top \left(\max_{a'}\{X_{h+1}(\cdot,a')\}\right) \quad \text{(Lemma A.17)}$$

$$\leq CH\phi\left(n^k(s,a)\right) + \left(1 + \frac{1}{H}\right)^3 \widehat{\mathbb{P}}^\dagger(\cdot \mid s,a)^\top \left(\max_{a'}\{X_{h+1}(\cdot,a')\}\right)$$
$$\text{(Lemma A.21 and } 1 + 2/H \leq (1+1/H)^2)$$

The inductive hypothesis gives us for any $s'$,

$$\left(\max_{a'}\{X_{h+1}(s',a')\}\right) \leq C\left(1 + \frac{1}{H}\right)^{3(H-h-1)}\left(\max_{a'}U_{h+1}^k(s',a')\right),$$

which implies

$$X_h(s,a) \leq CH\phi\left(n^k(s,a)\right) + \left(1 + \frac{1}{H}\right)^{3(H-h)}\widehat{\mathbb{P}}^\dagger(\cdot \mid s,a)^\top \left(\max_{a'}\{U_{h+1}^k(\cdot,a')\}\right)$$

$$\leq C\left(1 + \frac{1}{H}\right)^{3(H-h)}\left[H\phi\left(n^k(s,a)\right) + \widehat{\mathbb{P}}^\dagger(\cdot \mid s,a)^\top \left(\max_{a'}\{U_{h+1}^k(\cdot,a')\}\right)\right]. \quad \text{(A.8)}$$

Again, by definition of $X_h(s,a)$, we have

$$X_h(s,a) \leq H - h + 1 \leq C\left(1 + \frac{1}{H}\right)^{3(H-h)}H. \quad \text{(A.9)}$$

Combining (A.8) and (A.9), as well as the definition of $U_h^k$ (definition A.9), we prove the case of stage $h$ and we conclude the theorem by induction. $\qquad\square$

### A.2.3 Upper bounding the empirical uncertainty function (lemma 6.3)

*Proof.* First, we are going to prove

$$\frac{1}{K}\sum_{k=1}^K U_1^k(s,a) \leq \frac{H^2|\mathcal{S}||\mathcal{A}|}{K}\left[\log\left(\frac{6H|\mathcal{S}||\mathcal{A}|}{\delta}\right) + |\mathcal{S}|\log\left(e\left(1 + \frac{KH}{|\mathcal{S}|}\right)\right)\right]\log\left(1 + \frac{HK}{|\mathcal{S}||\mathcal{A}|}\right).$$
$$\text{(A.10)}$$

From the definition (see algorithm 2), we know an important property of uncertainty function is that $\pi_h^k$ is greedy with respect to $U_h^k$. So we have

$$U_h^k(s,a) = H\min\left\{1, \phi\left(n^k(s,a)\right)\right\} + \widehat{\mathbb{P}}^\dagger(s,a)^\top \left(\max_{a'}U_{h+1}^k(\cdot,a')\right)$$

$$= H\min\left\{1, \phi\left(n^k(s,a)\right)\right\} + \sum_{s',a'}\widehat{\mathbb{P}}^\dagger(s' \mid s,a)\pi_{h+1}^k(a' \mid s')U_{h+1}^k(s',a').$$

Therefore, we have

$$\frac{1}{K_{ucb}}\sum_{k=1}^{K_{ucb}}U_1^k(s_1,a)$$

$$\leq \frac{1}{K_{ucb}}\sum_{k=1}^{K_{ucb}}\max_a U_1^k(s_1,a)$$

$$\leq \frac{H}{K_{ucb}}\sum_{k=1}^{K_{ucb}}\sum_{h=1}^H\sum_{(s,a)}\widehat{d}_{\pi^k,h}^\dagger(s,a)\min\left\{1, \phi\left(n^k(s,a)\right)\right\} \qquad \text{(definition A.9)}$$

$$\leq \frac{4H^2}{K_{ucb}} \left[ \log\left(\frac{6H|\mathcal{S}||\mathcal{A}|}{\delta}\right) + |\mathcal{S}| \log\left(e\left(1 + \frac{K_{ucb}H}{|\mathcal{S}|}\right)\right) \right] \sum_{h=1}^{H} \sum_{k=1}^{K_{ucb}} \sum_{(s,a)} \frac{\widehat{d}^{\dagger}_{\pi^k,h}(s,a)}{\max\left\{1, \sum_{i=1}^{k-1} \sum_{h\in[H]} \widehat{d}^{\dagger}_{\pi^i,h}(s,a)\right\}}$$

(lemma A.20)

We apply Lemma D.7 and Jensen's Inequality to obtain

$$\sum_{h\in[H]} \sum_{k=1}^{K_{ucb}} \sum_{(s,a)} \widehat{d}^{\dagger}_{\pi^k,h}(s,a) \frac{1}{\max\left\{1, \sum_{i=1}^{k-1} \sum_{h\in[H]} \widehat{d}^{\dagger}_{\pi^i,h}(s,a)\right\}}$$

$$\leq 4 \sum_{(s,a)} \log\left(1 + \sum_{i=1}^{K_{ucb}} \sum_{h\in[H]} \widehat{d}^{\dagger}_{\pi^i,h}(s,a)\right) \qquad \text{(lemma D.7)}$$

$$\leq 4|\mathcal{S}||\mathcal{A}| \log\left(1 + \frac{1}{|\mathcal{S}||\mathcal{A}|} \sum_{i=1}^{K_{ucb}} \sum_{h\in[H]} \sum_{(s,a)} \widehat{d}^{\dagger}_{\pi^i,h}(s,a)\right)$$

$$\leq 4|\mathcal{S}||\mathcal{A}| \log\left(1 + \frac{HK_{ucb}}{|\mathcal{S}||\mathcal{A}|}\right)$$

Inserting the last display to the upper bound, we conclude the proof of the upper bound on $U$.

Then, to prove the sample complexity, we use lemma D.8. We take

$$B = \frac{64H^2|\mathcal{A}|}{\varepsilon^2} \left[\log\left(\frac{6H|\mathcal{S}||\mathcal{A}|}{\delta}\right) + |\mathcal{S}|\right] \text{ and } x = \frac{K_{ucb}}{|\mathcal{S}|}$$

in Lemma D.8. From lemma D.8, we know there exists a universal constant $C_1$ such that when $x \geq C_1 B \log(B)^2$, it holds that $B \log(1+x)(1 + \log(1+x)) \leq x$, which implies whenever

$$K_{ucb} \geq \frac{64H^2|\mathcal{S}||\mathcal{A}|}{\varepsilon^2} \left[\log\left(\frac{6H|\mathcal{S}||\mathcal{A}|}{\delta}\right) + |\mathcal{S}|\right],$$

it holds that

$$\frac{64H^2|\mathcal{S}||\mathcal{A}|}{K_{ucb}} \left[\log\left(\frac{6H|\mathcal{S}||\mathcal{A}|}{\delta}\right) + |\mathcal{S}|\right] \log\left(1 + \frac{K_{ucb}}{|\mathcal{S}|}\right) \left[1 + \log\left(1 + \frac{K_{ucb}}{|\mathcal{S}|}\right)\right] \leq \varepsilon^2. \quad \text{(A.11)}$$

For notation simplicity, we define

$$L = 16H^2|\mathcal{S}||\mathcal{A}| \left[\log\left(\frac{6H|\mathcal{S}||\mathcal{A}|}{\delta}\right) + |\mathcal{S}|\right].$$

Comparing the left hand side of (A.11) and the right hand side of (A.10), we know

Right hand side of (A.10)

$$= \frac{16H^2|\mathcal{S}||\mathcal{A}|}{K_{ucb}} \log\left(1 + \frac{HK_{ucb}}{|\mathcal{S}||\mathcal{A}|}\right) \left[\log\left(\frac{6H|\mathcal{S}||\mathcal{A}|}{\delta}\right) + |\mathcal{S}| + |\mathcal{S}| \log\left(1 + \frac{K_{ucb}H}{|\mathcal{S}|}\right)\right]$$

$$\leq \frac{16H^2|\mathcal{S}||\mathcal{A}|}{K_{ucb}} \left[\log(H) + \log\left(1 + \frac{K_{ucb}}{|\mathcal{S}|}\right)\right] \left[\log\left(\frac{6H|\mathcal{S}||\mathcal{A}|}{\delta}\right) + |\mathcal{S}| + |\mathcal{S}| \log\left(1 + \frac{K_{ucb}H}{|\mathcal{S}|}\right)\right]$$

$$\leq \frac{16H^2|\mathcal{S}||\mathcal{A}|}{K_{ucb}} \left[\log(H) + \log\left(1 + \frac{K_{ucb}}{|\mathcal{S}|}\right)\right] \left[\log\left(\frac{6H|\mathcal{S}||\mathcal{A}|}{\delta}\right) + |\mathcal{S}|\right] \left[1 + \log(H) + \log\left(1 + \frac{K_{ucb}}{|\mathcal{S}|}\right)\right]$$

$$= \frac{L}{K_{ucb}} \left[\log(H) + \log\left(1 + \frac{K_{ucb}}{|\mathcal{S}|}\right)\right] \left[1 + \log(H) + \log\left(1 + \frac{K_{ucb}}{|\mathcal{S}|}\right)\right].$$

It is easy to see that $\log(H) \leq \log(1 + K_{ucb}/|\mathcal{S}|)$, so the right hand side of last inequality is upper bounded by

$$\frac{4L}{K_{ucb}} \log\left(1 + \frac{K_{ucb}}{|\mathcal{S}|}\right) \left[1 + \log\left(1 + \frac{K_{ucb}}{|\mathcal{S}|}\right)\right].$$

From (A.11), we know this is upper bounded by $\varepsilon^2$ as long as $K_{ucb} \geq 4L/\varepsilon^2$. From the definition of L and equations (A.10) (A.11), we have when $K_{ucb} \geq 4L/\varepsilon^2$, it holds that

$$\frac{1}{K_{ucb}} \sum_{k=1}^{K_{ucb}} U_1^k(s_1, a) \leq \text{ right hand side of (A.10)} \leq \text{ left hand side of (A.11)} \leq \varepsilon^2.$$

In conclusion, this admits a sample complexity of

$$K_{ucb} = \frac{CH^2 |\mathcal{S}| |\mathcal{A}|}{\varepsilon^2} \left(\iota + |\mathcal{S}|\right) \text{polylog}\left(|\mathcal{S}|, |\mathcal{A}|, H, \frac{1}{\varepsilon}, \frac{1}{\delta}\right).$$

$\square$

## A.3 Omitted proofs

### A.3.1 Proof for lemma A.4 (high probability event)

The lemma A.4 is proved by combining all the lemmas below via a union bound. Below, we always denote $N(s, a, s')$ as the number of $(s, a, s')$ in the offline dataset.

**Lemma A.10.** $\mathcal{E}^P$ holds with probability at least $1 - \delta/6$.

*Proof.* For any fixed $(s, a, s')$ such that $N(s, a, s') \geq \Phi$, we denote $n_i(s, a)$ as the index of the i-th time when we visit $(s, a)$. For notation simplicity, we fix the state-action pair $(s, a)$ here and write $n_i(s, a)$ as $n_i$ simply for $i = 1, 2, ..., N(s, a)$. Notice that the total visiting time $N(s, a)$ is random, so our argument is based on conditional probability. We denote $X_i = \mathbb{I}\left(s'_{n_i} = s'\right)$ as the indicator of whether the next state is $s'$ when we visited $(s, a)$ at the i-th time. From the data generating mechanism and the definition for the reference MDP, we know conditional on the total number of visiting $N(s, a)$ and all the indexes $n_1 \leq ... \leq n_{N(s,a)}$, it holds that $X_1, X_2, ..., X_{N(s,a)}$ are independent Bernoulli random variable with successful probability being $\mathbb{P}^\dagger(s' \mid s, a)$. We denote $\overline{X}$ as their arithmetic average. Using Empirical Bernstein Inequality (Lemma D.3), and the fact that $\frac{1}{N(s,a)} \sum_{i=1}^{N(s,a)} \left(X_i - \overline{X}\right)^2 \leq \frac{1}{N(s,a)} \sum_{i=1}^{N(s,a)} X_i^2 = \frac{1}{N(s,a)} \sum_{i=1}^{N(s,a)} X_i = \widehat{\mathbb{P}}^\dagger(s' \mid s, a)$, we have

$$\mathbb{P}\left(\mathcal{E}^P \mid n_i \left(1 \leq i \leq N(s, a)\right), N(s, a) = n\right) \geq 1 - \delta/6.$$

Taking integral of the conditional probability, we conclude that the unconditional probability of the event $\mathcal{E}^P$ is at least $1 - \delta/6$. Therefore, we conclude. $\square$

**Lemma A.11.** For $\delta > 0$, it holds that $\mathbb{P}\left(\mathcal{E}^4\right) \geq 1 - \delta/6$.

*Proof.* Consider for a fixed triple $(s, a) \in \mathcal{S} \times \mathcal{A}$. Let $m(s, a)$ denote the number of times the tuple $(s, a)$ was encountered in total during the online interaction phase. We define $X_i$ as follows. For $i \leq m(s, a)$, we let $X_i$ be the subsequent state $s'$ when $(s, a)$ was encountered the i-th time in the whole run of the algorithm. Otherwise, we let $X_i$ be an independent sample from $\mathbb{P}^\dagger(s, a)$. By construction, $\{X_i\}$ is a sequence of i.i.d. categorical random variables from $\mathcal{S}$ with distribution $\mathbb{P}^\dagger(\cdot \mid s, a)$. We denote $\widetilde{\mathbb{P}}_X^{\dagger,i} = \frac{1}{i} \sum_{j=1}^i \delta_{X_j}$ as the empirical probability mass and $\mathbb{P}_X^\dagger$ as the probability mass of $X_i$. Then, from Lemma D.4, we have with probability at least $1 - \delta/6$, it holds that for any $i \in \mathbb{N}$,

$$\text{KL}\left(\widetilde{\mathbb{P}}_X^{\dagger,i}; \mathbb{P}_X^\dagger\right) \leq \frac{1}{i} \left[\log\left(\frac{6}{\delta}\right) + |\mathcal{S}| \log\left(e\left(1 + \frac{i}{|\mathcal{S}|}\right)\right)\right],$$

which implies

$$\text{KL}\left(\widetilde{\mathbb{P}}_X^\dagger; \mathbb{P}^\dagger(s, a)\right) \leq \frac{1}{m(s, a)} \left[\log\left(\frac{6}{\delta}\right) + |\mathcal{S}| \log\left(e\left(1 + \frac{m(s, a)}{|\mathcal{S}|}\right)\right)\right].$$

Using a union bound for $(s, a) \in \mathcal{S} \times \mathcal{A}$, we conclude. $\square$

**Lemma A.12** (Lower Bound on Counters). *For $\delta > 0$, it holds that $\mathbb{P}\left(\mathcal{E}^2\right) \geq 1 - \delta/6$ and $\mathbb{P}\left(\mathcal{E}^5\right) \geq 1 - \delta/6$.*

*Proof.* We denote $n_h^k$ as the number of times we encounter $(s,a)$ at stage $h$ before the beginning of episode $k$. Concretely speaking, we define $n_h^1(s,a) = 0$ for any $(h,s,a)$. Then, we define

$$n_h^k(s,a) = n_h^k(s,a) + \mathbb{I}\left[(s,a) = (s_h^k, a_h^k)\right].$$

We fixed an $(s,a,h) \in [H] \times \mathcal{S} \times \mathcal{A}$ and denote $\mathcal{F}_k$ as the sigma field generated by the first $k-1$ episodes when running RF-UCB and $X_k = \mathbb{I}\left[(s_h^k, a_h^k) = (s,a)\right]$. Then, we know $X_k$ is $\mathcal{F}_{k+1}$-measurable and $\mathbb{E}\left[X_k \mid \mathcal{F}_k\right] = \widehat{d}^\dagger_{\pi^k,h}(s,a)$ is $\mathcal{F}_k$-measurable. Taking $W = \ln(6/\delta)$ in Lemma D.1 and applying a union bound, we know with probability $1 - \delta/6$, the following event happens:

$$\forall k \in \mathbb{N}, (h,s,a) \in [H] \times \mathcal{S} \times \mathcal{A}, \quad n_h^k(s,a) \geq \frac{1}{2} \sum_{i<k} \widehat{d}^\dagger_{\pi^i,h}(s,a) - \ln\left(\frac{6H\,|\mathcal{S}|\,|\mathcal{A}|}{\delta}\right).$$

To finish the proof, it remains to note that the event above implies the event we want by summing over $h \in [H]$ for each $k \in \mathbb{N}$ and each $(s,a) \in \mathcal{S} \times \mathcal{A}$. For the $\mathcal{E}^5$, the proof is almost the same. $\square$

**Lemma A.13** (Upper Bound on Counters). *For $\delta > 0$, it holds that $\mathbb{P}\left(\mathcal{E}^3\right) \geq 1 - \delta/6$.*

*Proof.* We denote $n_h^k$ as the number of times we encounter $(s,a)$ at stage $h$ before the beginning of episode $k$. Concretely speaking, we define $n_h^1(s,a) = 0$ for any $(h,s,a)$. Then, we define

$$n_h^k(s,a) = n_h^k(s,a) + \mathbb{I}\left[(s,a) = (s_h^k, a_h^k)\right].$$

We fixed an $(s,a,h) \in [H] \times \mathcal{S} \times \mathcal{A}$ and denote $\mathcal{F}_k$ as the sigma field generated by the first $k-1$ episodes when running RF-UCB and $X_k = \mathbb{I}\left[(s_h^k, a_h^k) = (s,a)\right]$. Then, we know $X_k$ is $\mathcal{F}_{k+1}$-measurable and $\mathbb{E}\left[X_k \mid \mathcal{F}_k\right] = \widehat{d}^\dagger_{\pi^k,h}(s,a)$ is $\mathcal{F}_k$-measurable. Taking $W = \ln(6/\delta)$ in Lemma D.2 and applying a union bound, we know with probability $1 - \delta/6$, the following event happens:

$$\forall k \in \mathbb{N}, (h,s,a) \in [H] \times \mathcal{S} \times \mathcal{A}, \quad n_h^k(s,a) \leq 2 \sum_{i<k} \widehat{d}^\dagger_{\pi^i,h}(s,a) + \ln\left(\frac{6H\,|\mathcal{S}|\,|\mathcal{A}|}{\delta}\right).$$

To finish the proof, it remains to note that the event above implies the event we want by summing over $h \in [H]$ for each $k \in \mathbb{N}$ and each $(s,a) \in \mathcal{S} \times \mathcal{A}$. $\square$

### A.3.2 Proof for lemma A.14 (property of intermediate uncertainty function)

**Lemma A.14** (Intermediate Uncertainty Function). *We define for any $(h,s,a) \in [H] \times \mathcal{S} \times \mathcal{A}$ and any deterministc policy $\pi$,*

$$W_{H+1}^\pi(s,a,r) := 0,$$

$$W_h^\pi(s,a,r) := \min\left\{ H - h + 1, \sqrt{\frac{8}{H^2} \overline{\phi}\left(m(s,a)\right) \cdot \mathrm{Var}_{s' \sim \widetilde{\mathbb{P}}^\dagger(s,a)}\left(V_{h+1}\left(s'; \widetilde{\mathbb{P}}^\dagger, r^\dagger, \pi\right)\right)} \right.$$

$$\left. + 9H\phi\left(m(s,a)\right) + \left(1 + \frac{1}{H}\right)\left(\widetilde{\mathbb{P}}^\dagger \pi_{h+1}\right) W_{h+1}^\pi(s,a,r) \right\},$$

*where*

$$\left(\widetilde{\mathbb{P}}^\dagger \pi_{h+1}\right) W_{h+1}^\pi(s,a,r) := \sum_{s'} \sum_{a'} \widetilde{\mathbb{P}}^\dagger\left(s' \mid s,a\right) \pi_{h+1}\left(a' \mid s'\right) W_{h+1}^\pi\left(s',a',r\right).$$

*In addition, we define $W_h^\pi(s^\dagger, a, r) = 0$ for any $h \in [H], a \in \mathcal{A}$ and any reward function $r$. Then, under $\mathcal{E}$, for any $(h,s,a) \in [H] \times \mathcal{S} \times \mathcal{A}$, any deterministic policy $\pi$, and any deterministic reward function $r$ (with its augmentation $r^\dagger$), it holds that*

$$\left| Q_h\left(s,a; \widetilde{\mathbb{P}}^\dagger, r^\dagger, \pi\right) - Q_h\left(s,a; \mathbb{P}^\dagger, r^\dagger, \pi\right) \right| \leq W_h^\pi(s,a,r).$$

*Proof.* We prove by induction. For $h = H + 1$, we know $W_{H+1}(s, a, r) = 0$ by definition and the left hand side of the inequality we want also vanishes. Suppose the claim holds for $h + 1$ and for any $s, a$. The Bellman equation gives us

$$\Delta_h := Q_h\left(s, a; \widetilde{\mathbb{P}}^\dagger, r^\dagger, \pi\right) - Q_h\left(s, a; \mathbb{P}^\dagger, r^\dagger, \pi\right) \leq \underbrace{\sum_{s'}\left(\widetilde{\mathbb{P}}^\dagger\left(s' \mid s, a\right) - \mathbb{P}^\dagger\left(s' \mid s, a\right)\right) V_{h+1}\left(s'; \mathbb{P}^\dagger, r^\dagger, \pi\right)}_{I}$$

$$+ \underbrace{\sum_{s'} \widetilde{\mathbb{P}}^\dagger\left(s' \mid s, a\right)\left|V_{h+1}\left(s'; \widetilde{\mathbb{P}}^\dagger, r^\dagger, \pi\right) - V_{h+1}\left(s'; \mathbb{P}^\dagger, r^\dagger, \pi\right)\right|}_{II}$$

Under $\mathcal{E}$, we know that $\mathrm{KL}\left(\widetilde{\mathbb{P}}^\dagger; \mathbb{P}^\dagger\right) \leq \frac{1}{H}\phi\left(m(s, a)\right)$ for any $(s, a)$, so from Lemma D.5 we have

$$|I| \leq \sqrt{\frac{2}{H}\phi\left(m(s, a)\right) \cdot \mathrm{Var}_{s' \sim \mathbb{P}^\dagger(s,a)}\left(V_{h+1}\left(s'; \mathbb{P}^\dagger, r^\dagger, \pi\right)\right)} + \frac{2}{3}(H - h)\frac{\phi\left(m(s, a)\right)}{H}$$

$$\leq \sqrt{\frac{2}{H}\phi\left(m(s, a)\right) \cdot \mathrm{Var}_{s' \sim \mathbb{P}^\dagger(s,a)}\left(V_{h+1}\left(s'; \mathbb{P}^\dagger, r^\dagger, \pi\right)\right)} + \frac{2}{3}\phi\left(m(s, a)\right),$$

where $\phi$ is the bonus defined as (A.4). Here, we let $f$ in Lemma D.5 be $V_{h+1}\left(s'; \mathbb{P}^\dagger, r^\dagger, \pi\right)$. So the range will be $H - h$ and the upper bound for KL divergence is given by high-probability event $\mathcal{E}$. We further apply Lemma D.6 to get

$$\mathrm{Var}_{s' \sim \mathbb{P}^\dagger(s,a)}\left(V_{h+1}\left(s'; \mathbb{P}^\dagger, r^\dagger, \pi\right)\right) \leq 2\,\mathrm{Var}_{s' \sim \widetilde{\mathbb{P}}^\dagger(s,a)}\left(V_{h+1}\left(s'; \mathbb{P}^\dagger, r^\dagger, \pi\right)\right) + 4H\phi\left(m(s, a)\right)$$

and

$$\mathrm{Var}_{s' \sim \widetilde{\mathbb{P}}^\dagger(s,a)}\left(V_{h+1}\left(s'; \mathbb{P}^\dagger, r^\dagger, \pi\right)\right) \leq 2\,\mathrm{Var}_{s' \sim \widetilde{\mathbb{P}}^\dagger(s,a)}\left(V_{h+1}\left(s'; \widetilde{\mathbb{P}}^\dagger, r^\dagger, \pi\right)\right)$$

$$+ 2H\sum_{s'}\widetilde{\mathbb{P}}^\dagger\left(s' \mid s, a\right)\left|V_{h+1}\left(s'; \widetilde{\mathbb{P}}^\dagger, r^\dagger, \pi\right) - V_{h+1}\left(s'; \mathbb{P}^\dagger, r^\dagger, \pi\right)\right|.$$

Therefore, we have

$$|I| \leq \frac{2}{3}\phi\left(m(s, a)\right) + \left[\frac{8}{H}\phi\left(m(s, a)\right)\mathrm{Var}_{s' \sim \widetilde{\mathbb{P}}^\dagger(s,a)}\left(V_{h+1}\left(s'; \widetilde{\mathbb{P}}^\dagger, r^\dagger, \pi\right)\right) + 8\phi\left(m(s, a)\right)^2\right.$$

$$\left. + 8\phi\left(m(s, a)\right)\sum_{s'}\widetilde{\mathbb{P}}^\dagger\left(s' \mid s, a\right)\left|V_{h+1}\left(s'; \widetilde{\mathbb{P}}^\dagger, r^\dagger, \pi\right) - V_{h+1}\left(s'; \mathbb{P}^\dagger, r^\dagger, \pi\right)\right|\right]^{1/2}.$$

Using the fact that $\sqrt{x + y} \leq \sqrt{x} + \sqrt{y}$ and $\sqrt{xy} \leq \frac{x+y}{2}$ for positive $x, y$, and the definition of $II$, we obtain

$$|I| \leq \sqrt{\frac{8}{H}\phi\left(m(s, a)\right) \cdot \mathrm{Var}_{s' \sim \widetilde{\mathbb{P}}^\dagger(s,a)}\left(V_{h+1}\left(s'; \widetilde{\mathbb{P}}^\dagger, r^\dagger, \pi\right)\right)} + 6H\phi\left(m(s, a)\right) + \frac{1}{H}|II|,$$

which implies

$$|\Delta_h| \leq \sqrt{\frac{8}{H}\phi\left(m(s, a)\right) \cdot \mathrm{Var}_{s' \sim \widetilde{\mathbb{P}}^\dagger(s,a)}\left(V_{h+1}\left(s'; \widetilde{\mathbb{P}}^\dagger, r^\dagger, \pi\right)\right)} + 6H\phi\left(m(s, a)\right) + \left(1 + \frac{1}{H}\right)|II|.$$

If $H\phi\left(m(s, a)\right) \leq 1$, then we have by definition

$$|\Delta_h| \leq \sqrt{\frac{8}{H^2} \cdot H\phi\left(m(s, a)\right) \cdot \mathrm{Var}_{s' \sim \widetilde{\mathbb{P}}^\dagger(s,a)}\left(V_{h+1}\left(s'; \widetilde{\mathbb{P}}^\dagger, r^\dagger, \pi\right)\right)} + 6H\phi\left(m(s, a)\right) + \left(1 + \frac{1}{H}\right)|II|$$

$$\leq \sqrt{\frac{8}{H^2}\overline{\phi}\left(m(s, a)\right) \cdot \mathrm{Var}_{s' \sim \widetilde{\mathbb{P}}^\dagger(s,a)}\left(V_{h+1}\left(s'; \widetilde{\mathbb{P}}^\dagger, r^\dagger, \pi\right)\right)} + 6H\phi\left(m(s, a)\right) + \left(1 + \frac{1}{H}\right)|II|$$

by definition. Otherwise, if $H\phi\left(m(s, a)\right) \geq 1$, we have

$$\sqrt{\frac{8}{H}\phi\left(m(s, a)\right) \cdot \mathrm{Var}_{s' \sim \widetilde{\mathbb{P}}^\dagger(s,a)}\left(V_{h+1}\left(s'; \widetilde{\mathbb{P}}^\dagger, r^\dagger, \pi\right)\right)} \leq \sqrt{8H\phi\left(m(s, a)\right)} \leq 3H\phi\left(m(s, a)\right).$$

Therefore, we have in either case

$$|\Delta_h| \leq \sqrt{\frac{8}{H^2}\overline{\phi}\left(m(s,a)\right)\cdot \mathrm{Var}_{s'\sim\widetilde{\mathbb{P}}^\dagger(s,a)}\left(V_{h+1}\left(s';\widetilde{\mathbb{P}}^\dagger,r^\dagger,\pi\right)\right)}+9H\phi\left(m(s,a)\right)+\left(1+\frac{1}{H}\right)|II|.$$

The induction hypothesis gives us that

$$|II| \leq \sum_{s'}\sum_{a'}\widetilde{\mathbb{P}}^\dagger\left(s'\mid s,a\right)\pi_{h+1}\left(a'\mid s'\right)\left|Q_{h+1}\left(s',a';\widetilde{\mathbb{P}}^\dagger,r^\dagger,\pi\right)-Q_{h+1}\left(s',a';\mathbb{P}^\dagger,r^\dagger,\pi\right)\right|$$

$$\leq \left(\widetilde{\mathbb{P}}^\dagger\pi_{h+1}\right)W_{h+1}^\pi(s,a,r).$$

Inserting this into the upper bound of $\Delta_h$, we prove the case of $h$ by the definition of $W_h^\pi(s,a,r)$ and the simple fact that $|\Delta_h| \leq H-h+1$. $\qquad\square$

### A.3.3 Proof for lemma A.15 and lemma A.16 (properties of uncertainty function)

In this section, we prove some lemmas for upper bounding the uncertainty functions $X_h(s,a)$. We first provide a basic upper bound for $X_h(s,a)$. The uncertainty function is defined in definition A.6.

**Lemma A.15.** *Under the high probability event $\mathcal{E}$ (defined in appendix A.1.2) for all $(h,s,a) \in [H] \times \mathcal{S} \times \mathcal{A}$, it holds that*

$$X_h(s,a) \leq 11H\min\left\{1,\phi\left(m(s,a)\right)\right\} + \left(1+\frac{2}{H}\right)\mathbb{P}^\dagger\left(\cdot\mid s,a\right)^\top\left(\max_{a'}X_{h+1}(\cdot,a')\right).$$

*In addition, for $s=s^\dagger$, from definition, we know the above upper bound naturally holds.*

*Proof.* For any fixed $(h,s,a) \in [H] \times \mathcal{S} \times \mathcal{A}$, from the definition of the uncertainty function, we know

$$X_h(s,a) \leq 9H\phi\left(m(s,a)\right) + \left(1+\frac{1}{H}\right)\widetilde{\mathbb{P}}^\dagger\left(\cdot\mid s,a\right)^\top\left(\max_{a'}X_{h+1}(\cdot,a')\right).$$

To prove the lemma, it suffices to bound the difference between $\widetilde{\mathbb{P}}^\dagger\left(\cdot\mid s,a\right)^\top\left(\max_{a'}X_{h+1}(\cdot,a')\right)$ and $\mathbb{P}^\dagger\left(\cdot\mid s,a\right)^\top\left(\max_{a'}X_{h+1}(\cdot,a')\right)$. Under $\mathcal{E}$(which happens with probability at least $1-\delta$), it holds that $\mathrm{KL}\left(\widetilde{\mathbb{P}}_h^\dagger\left(\cdot\mid s,a\right);\mathbb{P}^\dagger\left(\cdot\mid s,a\right)\right) \leq \frac{1}{H}\phi\left(m(s,a)\right)$. Applying Lemma D.5 and a simple bound for variance gives us

$$\left|\widetilde{\mathbb{P}}^\dagger\left(\cdot\mid s,a\right)^\top\left(\max_{a'}X_{h+1}(\cdot,a')\right)-\mathbb{P}^\dagger\left(\cdot\mid s,a\right)^\top\left(\max_{a'}X_{h+1}(\cdot,a')\right)\right|$$

$$\leq\sqrt{\frac{2}{H}\mathrm{Var}_{s'\sim\mathbb{P}^\dagger(s,a)}\left(\max_{a'}X_{h+1}\left(s',a'\right)\right)\phi\left(m(s,a)\right)}+\frac{2}{3}\phi\left(m(s,a)\right)$$

$$\leq\sqrt{\left[\frac{2}{H}\mathbb{P}^\dagger\left(\cdot\mid s,a\right)^\top\left(\max_{a'}X_{h+1}(\cdot,a')^2\right)\right]\phi\left(m(s,a)\right)}+\frac{2}{3}\phi\left(m(s,a)\right)$$

$$\leq\sqrt{\left[\frac{2}{H}\mathbb{P}^\dagger\left(\cdot\mid s,a\right)^\top\left(\max_{a'}X_{h+1}(\cdot,a')\right)\right]\cdot H\phi\left(m(s,a)\right)}+\frac{2}{3}\phi\left(m(s,a)\right)$$

$$\leq\frac{1}{H}\mathbb{P}^\dagger\left(\cdot\mid s,a\right)^\top\left(\max_{a'}X_{h+1}(\cdot,a')\right)+2H\phi\left(m(s,a)\right).$$

The last line comes from $\sqrt{ab} \leq \frac{a+b}{2}$ for any positive $a,b$, and the fact that $\frac{1}{2}H\phi(m(s,a)) + \frac{2}{3}\phi(m(s,a)) \leq 2H\phi(m(s,a))$. Insert this bound into the definition of $X_h(s,a)$ to obtain

$$X_h(s,a) \leq 11H\phi\left(m(s,a)\right) + \left(1+\frac{2}{H}\right)\mathbb{P}^\dagger\left(\cdot\mid s,a\right)^\top\left(\max_{a'}X_{h+1}(\cdot,a')\right).$$

Noticing that $X_h(s,a) \leq H-h+1 \leq 11H$, we conclude. $\qquad\square$

**Lemma A.16.** *There exists a universal constant $C \geq 1$ such that under the high probability event $\mathcal{E}$, when $3K_{ucb} \geq K_{de}$, we have for any $(h, s, a) \in [H] \times \mathcal{S} \times \mathcal{A}$, it holds that*

$$X_h(s, a) \leq CH \frac{K_{ucb}}{K_{de}} \phi \left(n^{K_{ucb}}(s, a)\right) + \left(1 + \frac{2}{H}\right) \mathbb{P}^\dagger(\cdot \mid s, a)^\top \left(\max_{a'} \{X_{h+1}(\cdot, a')\}\right).$$

*In addition, for $s = s^\dagger$, from definition, we know the above upper bound naturally holds.*

*Proof.* Here, the universal constant may vary from line to line. Under $\mathcal{E}$, we have

$$X_h(s, a)$$

$$\leq CH\phi\left(m(s, a)\right) + \left(1 + \frac{1}{H}\right) \widetilde{\mathbb{P}}^\dagger(\cdot \mid s, a)^\top \left(\max_{a'} X_{h+1}(\cdot, a')\right). \qquad \text{(definition)}$$

$$\leq CH \min\{1, \phi\left(m(s, a)\right)\} + \left(1 + \frac{2}{H}\right) \mathbb{P}^\dagger(\cdot \mid s, a)^\top \left(\max_{a'} \{X_{h+1}(\cdot, a')\}\right) \qquad \text{(Lemma A.15)}$$

$$\leq CH\phi\left(K_{de} \sum_{h \in [H]} w_h^{mix}(s, a)\right) + \left(1 + \frac{2}{H}\right) \mathbb{P}^\dagger(\cdot \mid s, a)^\top \left(\max_{a'} \{X_{h+1}(\cdot, a')\}\right)$$

$$\text{(Lemma A.18)}$$

$$= CH\phi\left(\frac{K_{de}}{K_{ucb}} \sum_{k=1}^{K_{ucb}} \sum_{h \in [H]} d_{\pi^k, h}^\dagger(s, a)\right) + \left(1 + \frac{2}{H}\right) \mathbb{P}^\dagger(\cdot \mid s, a)^\top \left(\max_{a'} \{X_{h+1}(\cdot, a')\}\right)$$

$$\text{(definition)}$$

$$\leq CH\phi\left(\frac{K_{de}}{3K_{ucb}} \sum_{k=1}^{K_{ucb}} \sum_{h \in [H]} \widehat{d}_{\pi^k, h}^\dagger(s, a)\right) + \left(1 + \frac{2}{H}\right) \mathbb{P}^\dagger(\cdot \mid s, a)^\top \left(\max_{a'} \{X_{h+1}(\cdot, a')\}\right)$$

$$\text{(Lemma A.22 and A.17)}$$

$$\leq CH \frac{K_{ucb}}{K_{de}} \phi\left(\sum_{k=1}^{K_{ucb}} \sum_{h \in [H]} \widehat{d}_{\pi^k, h}^\dagger(s, a)\right) + \left(1 + \frac{2}{H}\right) \mathbb{P}^\dagger(\cdot \mid s, a)^\top \left(\max_{a'} \{X_{h+1}(\cdot, a')\}\right)$$

$$\text{(Lemma A.17)}$$

Since $X_h(s, a) \leq (H - h) \leq CH \frac{K_{ucb}}{K_{de}}$, we can modify the last display as

$$X_h(s, a) \leq CH \frac{K_{ucb}}{K_{de}} \min\left\{1, \phi\left(\sum_{k=1}^{K_{ucb}} \sum_{h \in [H]} \widehat{d}_{\pi^k, h}^\dagger(s, a)\right)\right\} + \left(1 + \frac{2}{H}\right) \mathbb{P}^\dagger(\cdot \mid s, a)^\top \left(\max_{a'} \{X_{h+1}(\cdot, a')\}\right)$$

$$\leq CH \frac{K_{ucb}}{K_{de}} \phi\left(n^{K_{ucb}}(s, a)\right) + \left(1 + \frac{2}{H}\right) \mathbb{P}^\dagger(\cdot \mid s, a)^\top \left(\max_{a'} \{X_{h+1}(\cdot, a')\}\right)$$

$$\text{(Lemma A.19)}$$

Therefore, we conclude. $\qquad \square$

### A.3.4 Proof for lemma A.17, lemma A.18, lemma A.19, lemma A.20 (properties of bonus function)

For our bonus function, we have the following basic property.

**Lemma A.17.** $\phi(x)$ *is non-increasing when $x > 0$. For any $\alpha \leq 1$, we have $\phi(\alpha x) \leq \frac{1}{\alpha} \phi(x)$.*

*Proof.* We define $f(x) := \frac{1}{x} \left[C + D \log\left(e\left(1 + \frac{x}{D}\right)\right)\right]$, where $C, D \geq 1$. Then, for $x > 0$,

$$f'(x) = -\frac{C + D \log\left(e\left(1 + \frac{x}{D}\right)\right)}{x^2} + \frac{D}{x(D + x)} \leq -\frac{C + D \log\left(1 + \frac{x}{D}\right)}{x^2} \leq 0.$$

Taking $C = \log\left(6H |\mathcal{S}| |\mathcal{A}| \delta\right)$ and $D = |\mathcal{S}|$, we conclude thee first claim. For the second claim, it is trivial since the logarithm function is increasing. $\qquad \square$

**Lemma A.18.** *Under $\mathcal{E}$, we have for any $(s, a) \in \mathcal{S} \times \mathcal{A}$,*

$$\min\{1, \phi(m(s,a))\} \leq 4\phi\left(K_{de} \sum_{h=1}^{H} w_h^{mix}(s,a)\right).$$

*Proof.* For fixed $(s,a) \in \mathcal{S} \times \mathcal{A}$, when $H \ln(6H |\mathcal{S}| |\mathcal{A}| / \delta) \leq \frac{1}{4}\left(K_{de} \sum_{h \in [H]} w_h^{mix}(s,a)\right)$, we know that $\mathcal{E}^5$ implies $m(s,a) \geq \frac{1}{4} \sum_{h \in [H]} K_{de} w_h^{mix}(s,a)$, From Lemma A.17, we know

$$\phi(m(s,a)) \leq \phi\left(\frac{1}{4} \sum_{h \in [H]} K_{de} w_h^{mix}(s,a)\right) \leq 4\phi\left(\sum_{h \in [H]} K_{de} w_h^{mix}(s,a)\right).$$

When $H \ln(6H |\mathcal{S}| |\mathcal{A}| / \delta) > \frac{1}{4}\left(K_{de} \sum_{h \in [H]} w_h^{mix}(s,a)\right)$, simple algebra shows that

$$\min\{1, \phi(m(s,a))\} \leq 1 \leq \frac{4H \ln(6H |\mathcal{S}| |\mathcal{A}| / \delta)}{K_{de} \sum_{h \in [H]} w_h^{mix}(s,a)} \leq 4\phi\left(K_{de} \sum_{h \in [H]} w_h^{mix}(s,a)\right).$$

Therefore, we conclude. $\qquad\square$

**Lemma A.19.** *Under $\mathcal{E}$, we have for any $(s, a) \in \mathcal{S} \times \mathcal{A}$,*

$$\min\left\{1, \phi\left(\sum_{k=1}^{K_{ucb}} \sum_{h=1}^{H} \widehat{d}_{\pi^k,h}^{\dagger}(s,a)\right)\right\} \leq 4\phi\left(n^{K_{ucb}}(s,a)\right).$$

*Proof.* We consider under the event $\mathcal{E}^3$, it holds that for any $(s,a) \in \mathcal{S} \times \mathcal{A}$,

$$\sum_{k=1}^{K_{ucb}} \sum_{h=1}^{H} \widehat{d}_{\pi^k,h}^{\dagger}(s,a) \geq \frac{1}{2} n^{K_{ucb}}(s,a) - \frac{1}{2} H \ln\left(\frac{6H |\mathcal{S}| |\mathcal{A}|}{\delta}\right).$$

If $H \ln(6H |\mathcal{S}| |\mathcal{A}| / \delta) \leq \frac{1}{2} n^{K_{ucb}}(s,a)$, then we have $\sum_{k=1}^{K_{ucb}} \sum_{h=1}^{H} \widehat{d}_{\pi^k,h}^{\dagger}(s,a) \geq \frac{1}{4} n^{K_{ucb}}(s,a)$, which implies

$$\phi\left(\sum_{k=1}^{K_{ucb}} \sum_{h=1}^{H} \widehat{d}_{\pi^k,h}^{\dagger}(s,a)\right) \leq \phi\left(\frac{1}{4} n^{K_{ucb}}(s,a)\right) \leq 4\phi\left(n^{K_{ucb}}(s,a)\right).$$

Otherwise, if $H \ln(6H |\mathcal{S}| |\mathcal{A}| / \delta) > \frac{1}{2} n^{K_{ucb}}(s,a)$, then we have

$$\min\left\{1, \phi\left(\sum_{k=1}^{K_{ucb}} \widehat{d}_{\pi^k,h}^{\dagger}(s,a)\right)\right\} \leq 1 \leq \frac{2H \ln(6H |\mathcal{S}| |\mathcal{A}| / \delta)}{n^{K_{ucb}}(s,a)} \leq 4\phi\left(n^{K_{ucb}}(s,a)\right).$$

Combining the two cases above, we conclude. $\qquad\square$

**Lemma A.20.** *Under $\mathcal{E}$, we have for any $k \in [K_{ucb}]$ and any $(s, a) \in \mathcal{S} \times \mathcal{A}$,*

$$\min\{1, \phi(n^k(s,a))\}$$
$$\leq \frac{4H}{\max\left\{1, \sum_{i<k} \sum_{h \in [H]} \widehat{d}_{\pi^i,h}^{\dagger}(s,a)\right\}} \left[\log\left(\frac{6H|\mathcal{S}||\mathcal{A}|}{\delta}\right) + |\mathcal{S}| \log\left(e\left(1 + \frac{\sum_{i<k} \sum_{h \in [H]} \widehat{d}_{\pi^i,h}^{\dagger}(s,a)}{|\mathcal{S}|}\right)\right)\right].$$

*Proof.* Under $\mathcal{E}^2$, it holds that

$$n^k(s,a) \geq \frac{1}{2} \sum_{i<k} \sum_{h \in [H]} \widehat{d}^{\dagger}_{\pi^i,h}(s,a) - H \ln \left( \frac{6H|\mathcal{S}||\mathcal{A}|}{\delta} \right).$$

If $H \ln \left( 6H|\mathcal{S}||\mathcal{A}|/\delta \right) \leq \frac{1}{4} \sum_{i<k} \sum_{h \in [H]} \widehat{d}^{\dagger}_{\pi^i,h}(s,a)$, then $n^k(s,a) \geq \frac{1}{4} \sum_{i<k} \sum_{h \in [H]} \widehat{d}^{\dagger}_{\pi^i,h}(s,a)$ and hence,

$$\min \left\{ 1, \phi \left( n^k(s,a) \right) \right\} \leq \phi \left( n^k(s,a) \right) \leq \phi \left( \frac{1}{4} \sum_{i<k} \sum_{h \in [H]} \widehat{d}^{\dagger}_{\pi^i,h}(s,a) \right) \leq 4\phi \left( \sum_{i<k} \sum_{h \in [H]} \widehat{d}^{\dagger}_{\pi^i,h}(s,a) \right).$$

This result equals to the right hand side in the lemma, because $\sum_{i<k} \sum_{h \in [H]} \widehat{d}^{\dagger}_{\pi^i,h}(s,a) \geq 4H \ln \left( 6H|\mathcal{S}||\mathcal{A}|/\delta \right) \geq 1$ (so taking maximum does not change anything). Otherwise, if $H \ln \left( 6H|\mathcal{S}||\mathcal{A}|/\delta \right) > \frac{1}{4} \sum_{i<k} \sum_{h \in [H]} \widehat{d}^{\dagger}_{\pi^i,h}(s,a)$, then

$$\min \left\{ 1, \phi \left( n^k(s,a) \right) \right\} \leq 1 \leq \frac{4H \ln \left( H|\mathcal{S}||\mathcal{A}|/\delta' \right)}{\sum_{i<k} \sum_{h \in [H]} \widehat{d}^{\dagger}_{\pi^i,h}(s,a)}.$$

Since $1 \leq 4H \ln \left( 6H|\mathcal{S}||\mathcal{A}|/\delta \right)$, we have

$$\min \left\{ 1, \phi \left( n^k(s,a) \right) \right\} \leq 1 \leq \frac{4H \ln \left( H|\mathcal{S}||\mathcal{A}|/\delta' \right)}{\max \left\{ 1, \sum_{i<k} \sum_{h \in [H]} \widehat{d}^{\dagger}_{\pi^i,h}(s,a) \right\}} \leq \text{RHS}$$

The last inequality comes from simple algebra. Therefore, we conclude. $\square$

### A.3.5 Proof for lemma A.21 and lemma A.22 (properties of empirical sparsified MDP)

In this section, we state two important properties of the empirical sparsified MDP and prove them. We remark that we do not include $\mathbb{P} \left( s^{\dagger} \mid s,a \right)$ in these two lemmas, since by definition $s^{\dagger} \notin \mathcal{S}$.

**Lemma A.21** (Multiplicative Accuracy). *We set*

$$\Phi \geq 6H^2 \log \left( \frac{12 |\mathcal{S}|^2 |\mathcal{A}|}{\delta} \right).$$

*Then, when $H \geq 2$, under $\mathcal{E}^P$, we have for any $(s,a,s') \in \mathcal{S} \times \mathcal{A} \times \mathcal{S}$,*

$$\left( 1 - \frac{1}{H} \right) \widehat{\mathbb{P}}^{\dagger}(s' \mid s,a) \leq \mathbb{P}^{\dagger}(s' \mid s,a) \leq \left( 1 + \frac{1}{H} \right) \widehat{\mathbb{P}}^{\dagger}(s' \mid s,a).$$

*Proof.* For $N(s,a,s') < \Phi$, both sides are zero. For $N(s,a,s') \geq \Phi$, recall $\widehat{\mathbb{P}}^{\dagger}(s' \mid s,a) = \frac{N(s,a,s')}{N(s,a)}$, then from lemma A.10, with probability at least $1 - \delta$,

$$\left| \widehat{\mathbb{P}}^{\dagger}(s' \mid s,a) - \mathbb{P}^{\dagger}(s' \mid s,a) \right|$$

$$\leq \sqrt{\frac{2\widehat{\mathbb{P}}^{\dagger}(s' \mid s,a)}{N(s,a)} \log \left( \frac{12 |\mathcal{S}|^2 |\mathcal{A}|}{\delta} \right)} + \frac{14}{3N(s,a)} \log \left( \frac{12 |\mathcal{S}|^2 |\mathcal{A}|}{\delta} \right)$$

$$\text{(lemma A.10 and definition of } \mathcal{E}^P\text{)}$$

$$= \left[ \sqrt{\frac{2}{N(s,a,s')} \log \left( \frac{12 |\mathcal{S}|^2 |\mathcal{A}|}{\delta} \right)} + \frac{14}{3N(s,a,s')} \log \left( \frac{12 |\mathcal{S}|^2 |\mathcal{A}|}{\delta} \right) \right] \cdot \widehat{\mathbb{P}}^{\dagger}(s' \mid s,a)$$

$$(\widehat{\mathbb{P}}^{\dagger}(s' \mid s,a) = \frac{N(s,a,s')}{N(s,a)})$$

$$\leq \left[ \sqrt{\frac{1}{3H^2}} + \frac{7}{9H^2} \right] \cdot \widehat{\mathbb{P}}^{\dagger}(s' \mid s,a) \leq \frac{\widehat{\mathbb{P}}^{\dagger}(s' \mid s,a)}{H},$$

where the second line comes from the lower bound on $\Phi$. We conclude. $\square$

**Lemma A.22** (Bound on Ratios of Visitation Probability). *For any deterministic policy $\pi$ and any $(h, s, a) \in [H] \times \mathcal{S} \times \mathcal{A}$, it holds that*

$$\frac{1}{4} d^{\dagger}_{\pi,h}(s, a) \leq \widehat{d}^{\dagger}_{\pi,h}(s, a) \leq 3 d^{\dagger}_{\pi,h}(s, a).$$

*Here, recall that we denote $d^{\dagger}_{\pi,h}(s, a)$ and $\widehat{d}^{\dagger}_{\pi,h}(s, a)$ as the occupancy measure of $(s, a)$ at stage $h$ under policy $\pi$, on $\mathbb{P}^{\dagger}$ (the transition dynamics in the sparsfied MDP) and $\widehat{\mathbb{P}}^{\dagger}$(the transition dynamics in the empirical sparsified MDP) respectively.*

We remark that for $s^{\dagger} \notin \mathcal{S}$ the inequality does not necessarily hold.

*Proof.* We denote $T_{h,s,a}$ as all truncated trajectories $(s_1, a_1, s_2, a_2, ..., s_h, a_h)$ up to stage $h$ such that $(s_h, a_h) = (s, a)$. Notice that if $\tau_h = (s_1, a_1, s_2, a_2, ..., s_h, a_h) \in T_{h,s,a}$, then it holds that $s_i \neq s^{\dagger}$ for $1 \leq i \leq h - 1$. We denote $\mathbb{P}(\cdot; \mathbb{P}', \pi)$ as the probability under a specific transition dynamics $\mathbb{P}'$ and policy $\pi$. For any fixed $(h, s, a) \in [H] \times \mathcal{S} \times \mathcal{A}$ and any fixed $\tau \in T_{h,s,a}$, we apply Lemma A.21 to get

$$\mathbb{P}\left[\tau; \widehat{\mathbb{P}}^{\dagger}, \pi\right] = \prod_{i=1}^{h} \pi_i (a_i \mid s_i) \prod_{i=1}^{h-1} \widehat{\mathbb{P}}^{\dagger} (s_{i+1} \mid s_i, a_i)$$

$$\leq \left(1 + \frac{1}{H}\right)^{H} \prod_{i=1}^{h} \pi_i (a_i \mid s_i) \prod_{i=1}^{h-1} \mathbb{P}^{\dagger} (s_{i+1} \mid s_i, a_i) \leq 3\mathbb{P}\left[\tau; \mathbb{P}^{\dagger}, \pi\right]$$

and

$$\mathbb{P}\left[\tau; \widehat{\mathbb{P}}^{\dagger}, \pi\right] = \prod_{i=1}^{h} \pi_i (a_i \mid s_i) \prod_{i=1}^{h-1} \widehat{\mathbb{P}}^{\dagger} (s_{i+1} \mid s_i, a_i)$$

$$\geq \left(1 - \frac{1}{H}\right)^{H} \prod_{i=1}^{h} \pi_i (a_i \mid s_i) \prod_{i=1}^{h-1} \mathbb{P}^{\dagger} (s_{i+1} \mid s_i, a_i) \geq \frac{1}{4}\mathbb{P}\left[\tau; \mathbb{P}^{\dagger}, \pi\right].$$

We conclude by rewriting the visiting probability as

$$d^{\dagger}_{\pi,h}(s, a) = \sum_{\tau \in T_{h,s,a}} \mathbb{P}\left[\tau; \mathbb{P}^{\dagger}, \pi\right]; \quad \widehat{d}^{\dagger}_{\pi,h}(s, a) = \sum_{\tau \in T_{h,s,a}} \mathbb{P}\left[\tau; \widehat{\mathbb{P}}^{\dagger}, \pi\right].$$

$\square$

## B  Additional comparisons

### B.1  Comparison with other comparator policy

Our main result compares the sub-optimality of the policy $\pi_{final}$ against the optimal policy on the sparsified MDP. We can further derive the sub-optimality of our output with respect to any comparator policy on the original MDP $\mathcal{M}$. If we denote $\pi_*, \pi_*^\dagger$ and $\pi_{final}$ as the global optimal policy, the optimal policy on the sparsified MDP and the policy output by our algorithm, respectively, and denote $\pi$ as the comparator policy, we have

$$
\begin{aligned}
&V_1\left(s_1, \mathbb{P}, r, \pi\right) - V_1\left(s_1, \mathbb{P}, r, \pi_{final}\right) \\
\leq & V_1\left(s_1, \mathbb{P}, r, \pi\right) - V_1\left(s_1, \mathbb{P}^\dagger, r, \pi\right) + \underbrace{V_1\left(s_1, \mathbb{P}^\dagger, r, \pi\right) - V_1\left(s_1, \mathbb{P}^\dagger, r, \pi_*^\dagger\right)}_{\leq 0} \\
& + \underbrace{V_1\left(s_1, \mathbb{P}^\dagger, r, \pi_*^\dagger\right) - V_1\left(s_1, \mathbb{P}^\dagger, r, \pi_{final}\right)}_{\lesssim \varepsilon} + \underbrace{V_1\left(s_1, \mathbb{P}^\dagger, r, \pi_{final}\right) - V_1\left(s_1, \mathbb{P}, r, \pi_{final}\right)}_{\leq 0} \\
\lesssim & \underbrace{V_1\left(s_1, \mathbb{P}, r, \pi_*\right) - V_1\left(s_1, \mathbb{P}^\dagger, r, \pi_*\right)}_{\text{Approximation Error}} + \varepsilon.
\end{aligned}
\tag{B.1}
$$

Here, the second term is non-positive from the definition of $\pi_*^\dagger$, the third term is upper bounded by $\varepsilon$ due to our main theorem (Theorem 5.1), and the last term is non-positive from the definition of the sparsified MDP. Since the connectivity graph of the sparsified MDP is a sub-graph of the original MDP, for any policy, the policy value on the sparsified MDP must be no higher than that on the original MDP.

At a high level, the $\varepsilon$ term in the last line of (B.1) represents the error from the finite online episodes, while the approximation error term $V_1\left(s_1, \mathbb{P}, r, \pi\right) - V_1\left(s_1, \mathbb{P}^\dagger, r, \pi\right)$ measures the policy value difference of $\pi$ on the original MDP and the sparsified one, representing the *coverage quality of the offline dataset*. If the dataset covers most of what $\pi$ covers, then this gap should be small. When $\pi$ is the global optimal policy $\pi_*$, this means the data should cover the state-actions pairs where optimal policy covers. The approximation error here plays a similar role as the concentrability coefficient in the offline reinforcement learning.

### B.2  Comparison with offline reinforcement learning

Our algorithm leverages some information from the offline dataset, so it is natural to ask how we expect that offline dataset to be, compared to traditional offline reinforcement learning does. In offline RL, we typically require the *concentrablity condition*, namely good coverage for the offline dataset, in order to achieve a polynomial sample complexity. Specifically, if we assume the offline data are sampled by first sampling $(s, a)$ i.i.d. from $\mu$ and then sampling the subsequent state from the transition dynamics, then the concentrability condition says the following constant $C^*$ is well-defined and finite.

$$
C^* := \sup_{(s,a)} \frac{d^{\pi_*}(s, a)}{\mu(s, a)} < \infty.
$$

The concentrability coefficient can be defined in several alternative ways, either for a set of policies or with respect to a single policy [Chen and Jiang, 2019, Zhan et al., 2022, Xie et al., 2021b, Zanette et al., 2021b]. Here, we follow the definition in [Xie et al., 2021b]. This means, the sampling distribution must covers the region where the global optimal policy covers, which is a very similar intuition obtained from our setting.

[Xie et al., 2021b] also gave optimal sample complexity (in terms of state-action pairs) for an offline RL algorithm is

$$
N = \widetilde{O}\left(\frac{C^* H^3 |\mathcal{S}|}{\varepsilon^2} + \frac{C^* H^{5.5} |\mathcal{S}|}{\varepsilon}\right),
$$

which is minimax optimal up to logarithm terms and higher order terms. Similar sample complexity were also given in several literature [Yin and Wang, 2020, Yin et al., 2020, Xie and Jiang, 2020b].

**Uniform data distribution**  For simplicity, we first assume $\mu$ to be uniform on all state-action pairs and the reward function to be given. Consider we have $N$ state-action pairs in the offline data, which are sampled i.i.d. from the distribution $\mu$. Notice that here, the global optimal policy $\pi_*$ still differs from the optimum on the sparsified MDP $\pi_*^\dagger$, since even if we get enough samples from each $(s, a)$ pairs, we might not get enough samples for every $(s, a, s')$ and hence, not all $(s, a, s')$ will be included in the set of known tuples.

Concretely, if we consider the case when we sample each state-action pair for $N/(|\mathcal{S}||\mathcal{A}|)$ times and simply treat the transition frequency as the true probability, then for any $N(s, a, s') < \Phi$, it holds that $\mathbb{P}\left(s' \mid s, a\right) = \frac{N(s,a,s')}{N(s,a)} = \frac{N(s,a,s')|\mathcal{S}||\mathcal{A}|}{N} \leq \frac{\Phi|\mathcal{S}||\mathcal{A}|}{N}$ . So for any any $N(s, a, s') \geq \Phi$, we know $\mathbb{P}\left(s' \mid s, a\right) = \mathbb{P}^\dagger\left(s' \mid s, a\right)$; while for any $N(s, a, s') < \Phi$, we have $\mathbb{P}\left(s' \mid s, a\right) \leq \frac{\Phi|\mathcal{S}||\mathcal{A}|}{N}$ and $\mathbb{P}^\dagger\left(s' \mid s, a\right) = 0$. Therefore, we have

$$\left|\mathbb{P}(s' \mid s, a) - \mathbb{P}^\dagger(s' \mid s, a)\right| \leq \frac{\Phi\,|\mathcal{S}|\,|\mathcal{A}|}{N}$$

From the value difference lemma (lemma D.11), we can upper bound the approximation error by

$$V_1\left(s_1, \mathbb{P}, r, \pi\right) - V_1\left(s_1, \mathbb{P}^\dagger, r, \pi\right)$$

$$= \mathbb{E}_{\mathbb{P},\pi}\left[\sum_{h=1}^{H}\sum_{s_{h+1}}\left(\mathbb{P}^\dagger(s_{h+1} \mid s_h, a_h) - \mathbb{P}(s_{h+1} \mid s_h, a_h)\right) \cdot V_h\left(s_{h+1}, \mathbb{P}, r, \pi\right)\middle|s_h = s\right]$$

$$\text{(lemma D.11)}$$

$$\leq \mathbb{E}_{\mathbb{P},\pi}\left[\sum_{h=1}^{H}\sum_{s_{h+1}}\frac{\Phi\,|\mathcal{S}|\,|\mathcal{A}|}{N} \cdot H\middle|s_h = s\right] \qquad \text{(the value function is upper bounded by } H\text{)}$$

$$\leq \frac{\Phi H^2\,|\mathcal{S}|^2\,|\mathcal{A}|}{N} \qquad\qquad\qquad \text{(summation over } h \in [H] \text{ and } s_{h+1} \in \mathcal{S}\text{)}$$

$$\asymp \widetilde{O}\left(\frac{H^4\,|\mathcal{S}|^2\,|\mathcal{A}|}{N}\right). \qquad\qquad\qquad \text{(definition of } \Phi \text{ in (5.1))}$$

Therefore, to get an $\varepsilon$-optimal policy compared to the global optimal one, we need the number of state-action pairs in the initial offline dataset $\mathcal{D}$ to be

$$N = \widetilde{O}\left(\frac{H^4\,|\mathcal{S}|^2\,|\mathcal{A}|}{\varepsilon}\right).$$

From the theorem 5.1, the offline data size we need here is actually significantly smaller than what we need for an offline algorithm. As long as $\sup_{(s,a)} d^{\pi_*}(s, a)$ is not too small, for instance, larger than $H^{-1.5}$, then we shave off the whole $1/\varepsilon^2$ term. The order of offline sample complexity here is actually $O(1/\varepsilon)$ instead of $O(1/\varepsilon^2)$ typical in offline RL, and this is significantly smaller in small $\varepsilon$ regime. To compensate the smaller offline sample size, actually we need more online sample to obtain an globally $\varepsilon$-optimal policy, and we summarize the general requirement for offline and online sample size in corollary 5.2.

**Non-uniform data distribution**  Assume the data generating distribution $\mu$ is not uniform but still supported on all $(s, a)$ pairs such that $d^{\pi_*}(s, a) > 0$, so that the concentrability coefficient in offline RL is still well defined. We simply consider the case when each state-action pair $(s, a)$ is sampled by $N\mu(s, a)$ times and treat the transition frequency as the true underlying probability. Then, following a very similar argument as in the last paragraph, the number of state-action pairs needed in the initial offline dataset in order to extract an $\varepsilon$-globally optimal policy is

$$N = \widetilde{O}\left(\frac{H^4\,|\mathcal{S}|}{\varepsilon}\sum_{s,a}\frac{d^{\pi_*}(s, a)}{\mu(s, a)}\right).$$

Here, the quantity

$$C^\dagger := \sum_{s,a}\frac{d^{\pi_*}(s, a)}{\mu(s, a)}$$

plays a similar role of classical concentrability coefficient and also measures the distribution shift between two policies. In the worst case, this coefficient can be $|\mathcal{S}|\,|\mathcal{A}|\,C^*$, resulting in an extra $|\mathcal{S}|\,|\mathcal{A}|$ factor compared to the optimal offline sample complexity. However, we still shave off the entire $1/\varepsilon^2$ term and also shave off $H^{1.5}$ in the $1/\varepsilon$ term.

**Partial coverage data**  Under partial coverage, we expect the output policy $\pi_{final}$ to be competitive with the value of the best policy supported in the region covered by the offline dataset. In such case, theorem 5.1 provides guarantees with the best comparator policy on the sparsified MDP $\mathcal{M}^\dagger$. In order to gain further intuition, it is best to 'translate' such guarantees into guarantees on $\mathcal{M}$.

In the worst case, the data distribution $\mu$ at a certain $(s, a)$ pair can be zero when $d^\pi(s, a) > 0$, which implies the concentrability coefficient $C^* = \infty$. Here, $\pi$ is an arbitrary comparator policy. In this case, either classical offline RL algorithm or our policy finetuning algorithm cannot guarantee an $\varepsilon$-optimal policy compared to the global optimal policy. However, we can still output a locally $\varepsilon$-optimal policy, compared to the optimal policy on the sparsified MDP.

In order to compare $\pi_{final}$ to any policy on the original MDP, we have the corollary 5.2, which will be proved in appendix B.3.

The statement in corollary 5.2 is a quite direct consequence of theorem 5.1, and it expresses the sub-optimality gap of $\pi_{final}$ with respect to any comparator policy $\pi$ on the original MDP $\mathcal{M}$. It can also be written in terms of the sub-optimality: If we fix a comparator policy $\pi$, then with probability at least $1 - \delta$, for any reward function $r$, the policy $\pi_{final}$ returned by algorithm 2 satisfies:

$$
V_1\left(s_1; \mathbb{P}, r, \pi\right) - V_1\left(s_1; \mathbb{P}, r, \pi_{final}\right) = \widetilde{O}\Big( \underbrace{\frac{H\,|\mathcal{S}|\,\sqrt{|\mathcal{A}|}}{\sqrt{K_{de}}}}_{\text{Online error}} + \underbrace{\frac{H^4\,|\mathcal{S}|}{N} \sum_{s,a} \frac{d^\pi(s, a)}{\mu(s, a)}}_{\text{Offline error}} \Big)
$$

$$
= \widetilde{O}\left( \frac{H\,|\mathcal{S}|\,\sqrt{|\mathcal{A}|}}{\sqrt{K_{de}}} + \frac{H^4\,|\mathcal{S}|^2\,|\mathcal{A}|}{N} \sup_{s,a} \frac{d^\pi(s, a)}{\mu(s, a)}. \right),
$$

where $K_{de}$ is the number of online episodes and $N$ is the number of state-action pairs in offline data. Here, the sub-optimality depends on an *online error* as well as on an *offline error*. The online error is the one that also arises in the statement of theorem 5.1. It is an error that can be reduced by collecting more online samples, i.e., by increasing $K$, with the typical inverse square-root depedence $1/\sqrt{K}$.

However, the upper bound suggests that even in the limit of infinite online data, the value of $\pi_{final}$ will not approach that of $\pi_*$ because of a residual error due to the *offline* dataset $\mathcal{D}$. Such residual error depends on certain concentrability factor expressed as $\sum_{s,a} \frac{d^\pi(s,a)}{\mu(s,a)} \le |\mathcal{S}||\mathcal{A}| \sup_{s,a} \frac{d^\pi(s,a)}{\mu(s,a)}$, whose presence is intuitive: if a comparator policy $\pi$ is not covered well, our algorithm does not have enough information to navigate to the area that $\pi$ tends to visit, and so it is unable to refine its estimates there. However the dependence on the number of offline samples $N = |\mathcal{D}|$ is through its *inverse*, i.e., $1/N$ as opposed to the more typical $1/\sqrt{N}$: such gap represents the improvable performance when additional online data are collected non-reactively.

It is useful to compare corollary 5.2 with what is achievable by using a minimax-optimal online algorithm[Xie et al., 2021b]. In this latter case, one can bound the sub-optimality gap for any comparator policy $\pi$ with high probability as

$$
V_1\left(s_1; \mathbb{P}, r^\dagger, \pi\right) - V_1\left(s_1; \mathbb{P}, r^\dagger, \pi_{final}\right) \le \widetilde{O}\left( \sqrt{\frac{H^3\,|\mathcal{S}|}{N} \sup_{s,a} \frac{d^\pi(s, a)}{\mu(s, a)}} \right). \tag{B.2}
$$

### B.3  Proof of corollary 5.2

Let's denote $\mathcal{D} = \{(s_i, a_i, s_i')\}_{i \in [N]}$ as the offline dataset, where $N$ as the total number of tuples. We keep the notation the same as in the main text. We use $N(s, a)$ and $N(s, a, s')$ to denote the counter of $(s, a)$ and $(s, a, s')$ in the offline data $\mathcal{D}$. The state-action pairs are sampled i.i.d. from $\mu(s, a)$ and the subsequent states are sampled from the transition dynamics. We fix a comparator policy $\pi$ and assume $\mu(s, a) > 0$ for any $(s, a)$ such that $d^\pi(s, a) > 0$, which implies a finite concentrability constant $C^*$. Here, $d^\pi(s, a)$ is the occupancy probability of $(s, a)$ when executing policy $\pi$, averaged over all stages $h \in [H]$.

Similar to the Section B.1, we have

$$V_1\left(s_1, \mathbb{P}, r, \pi\right) - V_1\left(s_1, \mathbb{P}, r, \pi_{final}\right)$$

$$\leq V_1\left(s_1, \mathbb{P}, r, \pi\right) - V_1\left(s_1, \mathbb{P}^\dagger, r, \pi\right) + \underbrace{V_1\left(s_1, \mathbb{P}^\dagger, r, \pi\right) - V_1\left(s_1, \mathbb{P}^\dagger, r, \pi_*^\dagger\right)}_{\leq 0}$$

$$+ V_1\left(s_1, \mathbb{P}^\dagger, r, \pi_*^\dagger\right) - V_1\left(s_1, \mathbb{P}^\dagger, r, \pi_{final}\right) + \underbrace{V_1\left(s_1, \mathbb{P}^\dagger, r, \pi_{final}\right) - V_1\left(s_1, \mathbb{P}, r, \pi_{final}\right)}_{\leq 0}$$

$$\lesssim \underbrace{V_1\left(s_1, \mathbb{P}, r, \pi\right) - V_1\left(s_1, \mathbb{P}^\dagger, r, \pi\right)}_{\text{Approximation Error}} + \underbrace{V_1\left(s_1, \mathbb{P}^\dagger, r, \pi_*^\dagger\right) - V_1\left(s_1, \mathbb{P}^\dagger, r, \pi_{final}\right)}_{\text{Estimation Error}}. \qquad \text{(B.3)}$$

where $\pi_*^\dagger$ and $\pi_{final}$ are the optimal policy on the sparsified MDP and the policy output by our algorithm, respectively, and $\pi$ is the fixed comparator policy. Here, the second term is non-positive from the definition of $\pi_*^\dagger$, the last term is non-positive from the definition of the sparsified MDP . This is because, for any state-action pair $(s, a)$ and any fixed policy $\pi$, the probability of reaching $(s, a)$ under $\mathbb{P}^\dagger$ will not exceed that under the true transition probability $\mathbb{P}$. If we denote the visiting probability under $\pi$ and $\mathbb{P}$ (or $\mathbb{P}^\dagger$ resp.) as $d_{\pi,h}(s, a)$ ($d_{\pi,h}^\dagger(s, a)$ resp.), then we have

$$d_{\pi,h}^\dagger(s, a) \leq d_{\pi,h}(s, a)$$

for any $h \in [H], s \in \mathcal{S}, a \in \mathcal{A}$. Note that, for $s = s^\dagger$, this does not hold necessarily. Them, for any policy $\pi$, we have

$$V_1\left(s_1, \mathbb{P}^\dagger, r, \pi\right) = \sum_{h=1}^H \sum_{s,a} d_{\pi,h}^\dagger(s, a) r(s, a) \qquad \text{(definition of policy value)}$$

$$= \sum_{h=1}^H \sum_{s \neq s^\dagger, a} d_{\pi,h}^\dagger(s, a) r(s, a) \qquad (r(s^\dagger, a) = 0 \text{ for any } a)$$

$$\leq \sum_{h=1}^H \sum_{s \neq s^\dagger, a} d_{\pi,h}(s, a) r(s, a) = V_1\left(s_1, \mathbb{P}, r, \pi\right).$$

Therefore, we get $V_1\left(s_1, \mathbb{P}^\dagger, r, \pi_{final}\right) - V_1\left(s_1, \mathbb{P}, r, \pi_{final}\right) \leq 0$ when we take $\pi = \pi_{final}$.

From the main result (Theorem 5.1), we know the estimation error is bounded as

$$\underbrace{V_1\left(s_1, \mathbb{P}^\dagger, r, \pi_*^\dagger\right) - V_1\left(s_1, \mathbb{P}^\dagger, r, \pi_{final}\right)}_{\text{Estimation Error}} \lesssim \widetilde{O}\left(\frac{H|\mathcal{S}|\sqrt{|\mathcal{A}|}}{\sqrt{K_{de}}}\right), \qquad \text{(B.4)}$$

where $K_{de}$ is the number of online episodes. Therefore, to make the right hand side of (B.4) less than $\varepsilon/2$, one needs at least $\widetilde{O}\left(\frac{H^2|\mathcal{S}||\mathcal{A}|}{\varepsilon^2}\right)$ online episodes. This is exactly what the main result shows.

So it suffices to bound the approximation error term. From the value difference lemma (Lemma D.11), we have

$$\left|V_1\left(s_1, \mathbb{P}, r, \pi\right) - V_1\left(s_1, \mathbb{P}^\dagger, r, \pi\right)\right|$$

$$= \left|\mathbb{E}_{\mathbb{P},\pi}\left[\sum_{i=1}^H \left(\mathbb{P}^\dagger(\cdot \mid s_i, a_i) - \mathbb{P}(\cdot \mid s_i, a_i)\right)^\top V_{i+1}\left(\cdot; \mathbb{P}^\dagger, r, \pi\right) \middle| s_1 = s\right]\right|$$

$$= \left|\sum_{i=1}^H \sum_{s_i, a_i} d_{\pi,h}(s_i, a_i)\left(\mathbb{P}^\dagger(\cdot \mid s_i, a_i) - \mathbb{P}(\cdot \mid s_i, a_i)\right)^\top V_{i+1}\left(\cdot; \mathbb{P}^\dagger, r, \pi\right)\right|$$

$$\text{(by expanding the expectation)}$$

$$\leq H|\mathcal{S}| \cdot \sum_{i=1}^H \sum_{s_i, a_i} d_{\pi,h}(s_i, a_i)\left(\sup_{s'}\left|\mathbb{P}^\dagger(s' \mid s_i, a_i) - \mathbb{P}(s' \mid s_i, a_i)\right|\right).$$

$$(V_{i+1} \leq H \text{ and the inner product has } |\mathcal{S}| \text{ terms})$$

We define

$$d_\pi(s, a) = \frac{1}{H} \sum_{h=1}^{H} d_{\pi,h}(s, a) \tag{B.5}$$

as the average visiting probability. Then, we have

$$\left| V_1\left(s_1, \mathbb{P}, r, \pi\right) - V_1\left(s_1, \mathbb{P}^\dagger, r, \pi\right) \right| \leq |\mathcal{S}| H^2 \sum_{s,a} \left[ \sup_{s'} \left| \mathbb{P}^\dagger(s' \mid s, a) - \mathbb{P}(s' \mid s, a) \right| d_\pi(s, a) \right], \tag{B.6}$$

So it suffices to upper bound $\left| \mathbb{P}^\dagger(s' \mid s, a) - \mathbb{P}(s' \mid s, a) \right|$. Notice here we only consider $s \neq s^\dagger$ and $s' \neq s^\dagger$, since the value function starting from $s^\dagger$ is always zero.

For $(s, a, s')$, if $N(s, a, s') \geq \Phi$, it holds that $\mathbb{P}^\dagger(s' \mid s, a) = \mathbb{P}(s' \mid s, a)$. Otherwise, from the definition, we know $\mathbb{P}^\dagger(s' \mid s, a) = 0$, so it suffices to bound $\mathbb{P}(s' \mid s, a)$ in this case. From lemma B.1, we know that with probability at least $1 - \delta/2$, for any $(s, a, s')$ such that $N(s, a, s') < \Phi$, it holds that

$$\mathbb{P}(s' \mid s, a) \leq \frac{2N(s, a, s') + 2\log\left(2|\mathcal{S}|^2 |\mathcal{A}|/\delta\right)}{N(s, a)}.$$

Then we deal with two cases. When $N\mu(s, a) \geq 6\Phi$, from lemma B.2 we have

$$\mathbb{P}(s' \mid s, a) \leq \frac{4N(s, a, s') + 2\log\left(4|\mathcal{S}|^2 |\mathcal{A}|/\delta\right)}{N\mu(s, a) - 2\log\left(2|\mathcal{S}||\mathcal{A}|/\delta\right)} \leq \frac{4\Phi + 2\log\left(4|\mathcal{S}|^2 |\mathcal{A}|/\delta\right)}{N\mu(s, a) - 2\log\left(2|\mathcal{S}||\mathcal{A}|/\delta\right)}.$$

From the definition of $\Phi$, we know that $2\log\left(4|\mathcal{S}|^2 |\mathcal{A}|/\delta\right) \leq \Phi$ and $2\log\left(2|\mathcal{S}||\mathcal{A}|/\delta\right) \leq \Phi$, which implies

$$\mathbb{P}(s' \mid s, a) \leq \frac{5\Phi}{N\mu(s, a) - \Phi} \leq \frac{6\Phi}{N\mu(s, a)}.$$

The last inequality comes from our assumption for $N\mu(s, a) \geq 6\Phi$.

In the other case, when $N\mu(s, a) < 6\Phi$, it holds that

$$\mathbb{P}(s' \mid s, a) \leq 1 \leq \frac{6\Phi}{N\mu(s, a)}$$

for any $(s, a, s')$. Therefore, for any for any $(s, a, s')$ such that $N(s, a, s') < \Phi$, we have

$$\mathbb{P}(s' \mid s, a) \leq \frac{6\Phi}{N\mu(s, a)}. \tag{B.7}$$

Combining equations (B.7) and (B.6), we know for any comparator policy $\pi$, it holds that

$$\underbrace{\left| V_1\left(s_1, \mathbb{P}, r, \pi\right) - V_1\left(s_1, \mathbb{P}^\dagger, r, \pi\right) \right|}_{\text{Approximation Error}} \lesssim \frac{\Phi H^2 |\mathcal{S}|}{N} \sum_{s,a} \frac{d_\pi(s, a)}{\mu(s, a)} \lesssim \widetilde{O}\left( \frac{H^4 |\mathcal{S}|}{N} \sum_{s,a} \frac{d_\pi(s, a)}{\mu(s, a)} \right),$$

where $N$ is the total number of transitions in the offline data. In order to make the right hand side of last display less than $\varepsilon/2$, one needs at least

$$\widetilde{O}\left( \frac{H^4 |\mathcal{S}|}{\varepsilon} \sum_{s,a} \frac{d_\pi(s, a)}{\mu(s, a)} \right) \leq \widetilde{O}\left( \frac{H^4 |\mathcal{S}|^2 |\mathcal{A}| C^*}{\varepsilon} \right),$$

offline transitions. Combining the proof for estimation error and approximation error, we conclude.

### B.4 Proof for lemma B.1 and lemma B.2

**Lemma B.1.** *With probability at least $1 - \delta/2$, for any $(s, a, s') \in \mathcal{S} \times \mathcal{A} \times \mathcal{S}$, it holds that*

$$N(s, a, s') \geq \frac{1}{2} N(s, a) \mathbb{P}\left(s' \mid s, a\right) - \log\left(\frac{2\left|\mathcal{S}\right|^2 \left|\mathcal{A}\right|}{\delta}\right).$$

*Proof.* We fixed $(s, a, s')$ and denote $I$ as the index set where $(s_i, a_i) = (s, a)$ for $i \in I$. We range the indexes in $I$ as $i_1 < i_2 < ... < i_{N(s,a)}$. For $j \leq N(s, a)$, we denote $X_j = \mathbb{I}\left(s'_{i_j} = s'\right)$, which is the indicator of whether the next state is $s'$ when we encounter $(s, a)$ the $j$-th time. When $j \geq N(s, a)$, we denote $X_j$ as independent Bernoulli random variables with successful rate $\mathbb{P}(s' \mid s, a)$. Then, we know $X_j$ for all $j \in \mathbb{N}$ are i.i.d. sequence of Bernoulli random variables. From Lemma D.1, we know with probability at least $1 - \delta/2$, for any positive integer $n$, it holds that

$$\sum_{j=1}^{n} X_j \geq \frac{1}{2} \sum_{j=1}^{n} \mathbb{P}(s' \mid s, a) - \log\left(\frac{2}{\delta}\right).$$

We take $n = N(s, a)$ (although $N(s, a)$ is random, we can still take it because for any $n$ the inequality above holds) to get

$$N(s, a, s') = \sum_{j=1}^{N(s,a)} X_j \geq \frac{1}{2} N(s, a) \mathbb{P}(s' \mid s, a) - \log\left(\frac{2}{\delta}\right).$$

Applying a union bound for all $(s, a, s')$, we conclude. $\qquad\square$

**Lemma B.2.** *With probability at least $1 - \delta/2$, for any $(s, a) \in \mathcal{S} \times \mathcal{A}$, it holds that*

$$N(s, a) \geq \frac{1}{2} N\mu(s, a) - \log\left(\frac{2\left|\mathcal{S}\right|\left|\mathcal{A}\right|}{\delta}\right).$$

*Proof.* If $\mu(s, a) = 0$, this is trivial. We fixed an $(s, a)$ such that $\mu(s, a) > 0$. For $j \leq N$, we denote $X_j = \mathbb{I}\left(s_j = s, a_j = a\right)$, which is the indicator of whether the $j$-th state-action pair we encounter in the offline dataset is $(s, a)$. When $j \geq N$, we denote $X_j$ as independent Bernoulli random variables with successful rate $\mu(s, a)$. Then, we know $X_j$ for all $j \in \mathbb{N}$ are i.i.d. sequence of Bernoulli random variables. From Lemma D.1, we know with probability at least $1 - \delta/2$, for any positive integer $n$, it holds that

$$\sum_{j=1}^{n} X_j \geq \frac{1}{2} \sum_{j=1}^{n} \mu(s, a) - \log\left(\frac{2}{\delta}\right).$$

We take $n = N$ to get

$$N(s, a) = \sum_{j=1}^{N} X_j \geq \frac{1}{2} N\mu(s, a) - \log\left(\frac{2}{\delta}\right).$$

Applying a union bound for all $(s, a)$ such that $\mu(s, a) > 0$, we conclude. $\qquad\square$

# C  Lower bound

In this section we briefly discuss the optimality of the algorithm. Although the following considerations are also mentioned in the main text, here we mention how they naturally lead to a lower bound.

**Lower bound for reward-free exploration**  Consider the MDP class $\mathcal{M}$ defined in the proof of the lower bound of Theorem 4.1 in [Jin et al., 2020b]. Assume that the dataset arises from a logging policy $\pi_{log}$ which induces the condition $N(s, a, s') \geq \Phi$ for all $(s, a, s') \in \mathcal{S} \times \mathcal{A}$ for every instance of the class. In this case, every MDP instance $\mathcal{M} \in \mathcal{M}$ and its sparsified version $\mathcal{M}^{\dagger}$ coincide. Then the concatenation of the logging policy $\pi_{log}$ and of the policy $\pi_{final}$ produced by our algorithm (i.e., algorithm 3) can be interpreted as a reactive policy, which must comply with the reward free lower bound established in Theorem 4.1 of [Jin et al., 2020b]. More precisely, the reward-free sample complexity lower bound established in Theorem 4.1 in [Jin et al., 2020b] is

$$\Omega\Big(\frac{|\mathcal{S}|^2|\mathcal{A}|H^2}{\varepsilon^2}\Big) \tag{C.1}$$

trajectories. This matches the sample complexity of theorem 5.1. Notice that the number of samples originally present in the dataset can be

$$|\mathcal{S}|^2|\mathcal{A}| \times \widetilde{O}(H^2) = \widetilde{O}(H^2|\mathcal{S}|^2|\mathcal{A}|), \tag{C.2}$$

a term independent of the accuracy $\varepsilon$. Given that when $\epsilon \leq 1$ we have

$$\widetilde{O}(H^2|\mathcal{S}|^2|\mathcal{A}|) + \Omega\Big(\frac{|\mathcal{S}|^2|\mathcal{A}|H^2}{\varepsilon^2}\Big) = \Omega\Big(\frac{|\mathcal{S}|^2|\mathcal{A}|H^2}{\varepsilon^2}\Big),$$

our algorithm is unimprovable beyond constant terms and logarithmic terms in a minimax sense.

**Lower bound for non-reactive exploration**  Consider the MDP class $\mathcal{M}$ defined in the proof of the lower bound in Theorem 1 of [Xiao et al., 2022]. It establishes an exponential sample complexity for non-reactive exploration *when no prior knowledge is available*. In other words, in absence of any data about the MDP, non-reactive exploration must suffer an exponential sample complexity. In such case, our theorem 5.1 (correctly) provides vacuous guarantees, because the sparsified MDP $\mathcal{M}$ is degenerate (all edges lead to the absorbing state).

**Combining the two constructions**  It is possible to combine the MDP class $\mathcal{M}_1$ from the paper [Xiao et al., 2022] with the MDP class $\mathcal{M}_2$ from the paper [Jin et al., 2020b]. In lieu of a formal proof, here we provide only a sketch of the construction that would induce a lower bound. More precisely, consider a starting state $s_1$ where only two actions—$a = 1$ and $a = 2$—are available. Taking $a = 1$ leads to the start state of an instance of the class $\mathcal{M}_1$, while taking $a = 2$ leads to the start state of an instance of the class $\mathcal{M}_2$; in both cases the transition occurs with probability one and zero reward is collected.

Furthermore, assume that the reward function given over the MDPs in $\mathcal{M}_1$ is shifted such that the value of a policy that takes $a = 1$ in $s_1$ and then plays optimally is 1 and that the reward functions on $\mathcal{M}_2$ is shifted such that the value of a policy which takes $a = 2$ initially and then playes optimally is $2\varepsilon$.

In addition, assume that the dataset arises from a logging policy $\pi_{log}$ which takes $a = 2$ initially and then visits all $(s, a, s')$ uniformly.

Such construction and dataset identify a sparsified MDP which coincide with $\mathcal{M}_2$ with the addition of $s_1$ (and its transition to $\mathcal{M}_2$ with zero reward). Intuitively, a policy with value arbitrarily close to 1 must take action $a = 1$ which leads to $\mathcal{M}_1$, which is the portion of the MDP that is unexplored in the dataset. In this case, unless the agent collects exponentially many trajectories in the online phase, the lower bound from [Xiao et al., 2022] implies that it is not possible to discover a policy with value close to 1 (e.g., larger than $1/2$). On the other hand, our theorem 5.1 guarantees that $\pi_{final}$ has a value at least $\varepsilon$, because $\pi_{final}$ is $\varepsilon$-optimal on the sparsified MDP—i.e., $\varepsilon$-optimal when restricted to an instance on $\mathcal{M}_2$—with high probability using at most $\sim H^2|\mathcal{S}|^2|\mathcal{A}|/\varepsilon^2$ trajectories. This value is unimprovable given the lower bound of Jin et al. [2020b], which applies to the class $\mathcal{M}_2$.

For completeness, in the next sub-section we refine the lower bound of [Xiao et al., 2022] to handle mixture policies.

## D   Technical lemmas and proofs

**Lemma D.1** (Lemma F.4 in [Dann et al., 2017]). *Let $\mathcal{F}_i$ for $i = 1 \ldots$ be a filtration and $X_1, \ldots X_n$ be a sequence of Bernoulli random variables with $\mathbb{P}\left(X_i = 1 \mid \mathcal{F}_{i-1}\right) = P_i$ with $P_i$ being $\mathcal{F}_{i-1}$-measurable and $X_i$ being $\mathcal{F}_i$ measurable. It holds that*

$$\mathbb{P}\left(\exists n : \sum_{t=1}^{n} X_t < \sum_{t=1}^{n} P_t/2 - W\right) \le e^{-W}.$$

**Lemma D.2.** *Let $\mathcal{F}_i$ for $i = 1 \ldots$ be a filtration and $X_1, \ldots X_n$ be a sequence of Bernoulli random variables with $\mathbb{P}\left(X_i = 1 \mid \mathcal{F}_{i-1}\right) = P_i$ with $P_i$ being $\mathcal{F}_{i-1}$-measurable and $X_i$ being $\mathcal{F}_i$ measurable. It holds that*

$$\mathbb{P}\left(\exists n : \sum_{t=1}^{n} X_t > \sum_{t=1}^{n} 2P_t + W\right) \le e^{-W}.$$

*Proof.* Notice that $\frac{1}{u^2}\left[\exp(u) - u - 1\right]$ is non-decreasing on $\mathbb{R}$, where at zero we continuously extend this function. For any $t \in \mathbb{N}$, since $X_t - P_t \le 1$, we have $\exp\left(X_t - P_t\right) - \left(X_t - P_t\right) - 1 \le \left(X_t - P_t\right)^2 (e - 2) \le \left(X_t - P_t\right)^2$. Taking expectation conditional on $\mathcal{F}_{t-1}$ and noticing that $P_t - X_t$ is a Martingale difference sequence w.r.t. the filtration $\mathcal{F}_t$, we have

$$\mathbb{E}\left[\exp\left(X_t - P_t\right) \mid \mathcal{F}_{t-1}\right] \le 1 + \mathbb{E}\left[\left(X_t - P_t\right)^2 \mid \mathcal{F}_{t-1}\right] \le \exp\left[\mathbb{E}\left[\left(X_t - P_t\right)^2 \mid \mathcal{F}_{t-1}\right]\right] \le \exp\left(P_t\right),$$

where the last inequality comes from the fact that conditional on $\mathcal{F}_{t-1}$, $X_t$ is a Bernoulli random variable. We define $M_n := \exp\left[\sum_{t=1}^{n}\left(X_t - 2P_t\right)\right]$, which is a supermartingale from our derivation above. We define now the stopping time $\tau = \min\left\{t \in \mathbb{N} : M_t > e^W\right\}$ and the sequence $\tau_n = \min\left\{t \in \mathbb{N} : M_t > e^W \vee t \ge n\right\}$. Applying the convergence theorem for nonnegative supermartingales (Theorem 4.2.12 in [Durrett, 2019]), we get that $\lim_{t\to\infty} M_t$ is well-defined almost surely. Therefore, $M_\tau$ is well-defined even when $\tau = \infty$. By the optional stopping theorem for nonnegative supermartingales (Theorem 4.8.4 in [Durrett, 2019], we have $\mathbb{E}\left[M_{\tau_n}\right] \le \mathbb{E}\left[M_0\right] \le 1$ for all $n$ and applying Fatou's lemma, we obtain $\mathbb{E}\left[M_\tau\right] = \mathbb{E}\left[\lim_{n\to\infty} M_{\tau_n}\right] \le \liminf_{n\to\infty}\mathbb{E}\left[M_{\tau_n}\right] \le 1$. Using Markov's inequality, we can finally bound

$$\mathbb{P}\left(\exists n : \sum_{t=1}^{n} X_t > 2\sum_{t=1}^{n} P_t + W\right) > \mathbb{P}(\tau < \infty) \le \mathbb{P}\left(M_\tau > e^W\right) \le e^{-W}\mathbb{E}\left[M_\tau\right] \le e^{-W}.$$

$\square$

**Lemma D.3** (Empirical Bernstein Inequality, Theorem 11 in [Maurer and Pontil, 2009]). *Let $n \ge 2$ and $x_1, \cdots, x_n$ be i.i.d random variables such that $|x_i| \le A$ with probability 1. Let $\bar{x} = \frac{1}{n}\sum_{i=1}^{n} x_i$, and $\widehat{V}_n = \frac{1}{n}\sum_{i=1}^{n}\left(x_i - \bar{x}\right)^2$, then with probability $1 - \delta$ we have*

$$\left|\frac{1}{n}\sum_{i=1}^{n} x_i - \mathbb{E}[x]\right| \le \sqrt{\frac{2\widehat{V}_n \log(2/\delta)}{n}} + \frac{14A}{3n}\log(2/\delta)$$

**Lemma D.4** (Concentration for KL Divergence, Proposition 1 in [Jonsson et al., 2020]). *Let $X_1, X_2, \ldots, X_n, \ldots$ be i.i.d. samples from a distribution supported over $\{1, \ldots, m\}$, of probabilities given by $\mathbb{P} \in \Sigma_m$, where $\Sigma_m$ is the probability simplex of dimension $m - 1$. We denote by $\widehat{\mathbb{P}}_n$ the empirical vector of probabilities. Then, for any $\delta \in [0, 1]$, with probability at least $1 - \delta$, it holds that*

$$\forall n \in \mathbb{N}, \mathrm{KL}\left(\widehat{\mathbb{P}}_n, \mathbb{P}\right) \le \frac{1}{n}\log\left(\frac{1}{\delta}\right) + \frac{m}{n}\log\left(e\left(1 + \frac{n}{m}\right)\right).$$

*We remark that there is a slight difference between it and the original version. In [Jonsson et al., 2020], they use $m - 1$ instead of $m$. But since the second term of the right hand side above is increasing with $m$, our version also holds.*

**Lemma D.5** (Bernstein Transportation, Lemma 11 in [Talebi and Maillard, 2018]). *For any function $f$ and any two probability measure $\mathbb{Q}, \mathbb{P}$ which satisfy $\mathbb{Q} \ll \mathbb{P}$, we denote $\mathbb{V}_P[f] := \mathrm{Var}_{X \sim \mathbb{P}}(f(X))$ and $\mathbb{S}(f) := \sup_x f(x) - \inf_x f(x)$. We assume $\mathbb{V}_P[f]$ and $\mathbb{S}(f)$ are finite, then we have*

$$\mathbb{E}_Q[f] - \mathbb{E}_P[f] \leqslant \sqrt{2\mathbb{V}_P[f]\mathrm{KL}(Q,P)} + \frac{2}{3}\mathbb{S}(f)\mathrm{KL}(Q,P),$$

$$\mathbb{E}_P[f] - \mathbb{E}_Q[f] \leqslant \sqrt{2\mathbb{V}_P[f]\mathrm{KL}(Q,P)}.$$

**Lemma D.6** (Difference of Variance, Lemma 12 in [Ménard et al., 2021]). *Let $\mathbb{P}, \mathbb{Q}$ be two probability measure on a discrete sample space of cardinality $\mathcal{S}$. Let $f, g$ be two functions defined on $\mathcal{S}$ such that $0 \leq g(s), f(s) \leq b$ for all $s \in \mathcal{S}$, we have that*

$$\mathrm{Var}_{\mathbb{P}}(f) \leq 2\,\mathrm{Var}_{\mathbb{P}}(g) + 2b\mathbb{E}_{\mathbb{P}}|f - g| \quad and$$

$$\mathrm{Var}_{\mathbb{Q}}(f) \leq \mathrm{Var}_{\mathbb{Q}}(f) + 3b^2\|\mathbb{P} - \mathbb{Q}\|_1,$$

*Further, if $\mathrm{KL}\,(\mathbb{P};\mathbb{Q}) \leq \alpha$, it holds that*

$$\mathrm{Var}_{\mathbb{Q}}(f) \leq 2\,\mathrm{Var}_{\mathbb{P}}(f) + 4b^2\alpha.$$

**Lemma D.7.** *For any sequence of numbers $z_1, \ldots, z_n$ with $0 \leq z_k \leq 1$, we have*

$$\sum_{k=1}^n \frac{z_k}{\max\left[1; \sum_{i=1}^{k-1} z_i\right]} \leq 4\log\left(\sum_{i=1}^n z_i + 1\right)$$

*Proof.*

$$\sum_{k=1}^n \frac{z_k}{\max\left[1; \sum_{i=1}^{k-1} z_i\right]} \leq 4\sum_{k=1}^n \frac{\sum_{i=1}^k z_i - \sum_{i=1}^{k-1} z_i}{2 + 2\sum_{i=1}^{k-1} z_i}$$

$$\leq 4\sum_{k=1}^n \frac{\sum_{i=1}^k z_i - \sum_{i=1}^{k-1} z_i}{1 + \sum_{i=1}^k z_i}$$

$$\leq 4\sum_{k=1}^n \int_{\sum_{i=1}^{k-1} z_i}^{\sum_{i=1}^k z_i} \frac{1}{1+x}\mathrm{d}x \leq 4\log\left(\sum_{i=1}^n z_i + 1\right).$$

$\square$

**Lemma D.8.** *For $B \geq 16$ and $x \geq 3$, there exists a universal constant $C_1 \geq 4$, such that when*
$$x \geq C_1 B \log(B)^2,$$
*it holds that*
$$B\log(1 + x)\left(1 + \log(1 + x)\right) \leq x.$$

*Proof.* We have

$$B\log(1 + x)\left(1 + \log(1 + x)\right) \leq B\left(1 + \log(1 + x)\right)^2 \leq B\left(1 + \log(2x)\right)^2 \leq B\left[\log(6x)\right]^2.$$

We define $f(x) := x - B\left[\log(6x)\right]^2$, then we have $f'(x) = 1 - \frac{2B\log(6x)}{x}$. Since $x \geq 2C_1 B\log(B)$, we have

$$f'(x) \geq 1 - \frac{\log(12C_1 B\log(B))}{C_1\log(B)}.$$

We can take $C_1 \geq 1$ such that $C_1\log(B) - \log(12C_1 B\log(B)) \geq 0$ whenever $B \geq 16$. Therefore, we know $f(x)$ is increasing when $x \geq C_1 B\log(B)^2$. Then, it suffices to prove

$$\left[\log\left(6C_1 B\log(B)^2\right)\right]^2 \leq C_1\log(B)^2.$$

Since $\left[\log\left(6C_1 B\log(B)^2\right)\right]^2 \leq 2\log(B)^2 + 2\left[\log\left(6C_1\log(B)^2\right)\right]^2$, it suffices to prove

$$\log\left(6C_1\log(B)^2\right) \leq \sqrt{\frac{C_1 - 2}{2}}\log(B).$$

When $C_1 \geq 4$, the difference of right hand side and left hand side s always increasing w.r.t. $B$ for fixed $C_1$. Therefore, it suffices to prove the case when $B = 16$. Noticing that we can always take a sufficiently large uniform constant $C_1$ such that the inequality above holds when $B = 16$, we conclude. $\square$

**Lemma D.9** (Chain rule of Kullback-Leibler divergence, Exercise 3.2 in [Wainwright, 2019]). *Given two $n$-variate distributions $\mathbb{Q}$ and $\mathbb{P}$, show that the Kullback-Leibler divergence can be decomposed as*

$$D(\mathbb{Q}; \mathbb{P}) = D(\mathbb{Q}_1; \mathbb{P}_1) + \sum_{j=2}^{n} \mathbb{E}_{\mathbb{Q}_1^{j-1}} \left[ D\left( \mathbb{Q}_j \left( \cdot \mid X_1^{j-1} \right); \mathbb{P}_j \left( \cdot \mid X_1^{j-1} \right) \right) \right],$$

*where $\mathbb{Q}_j \left( \cdot \mid X_1^{j-1} \right)$ denotes the conditional distribution of $X_j$ given $(X_1, \ldots, X_{j-1})$ under $\mathbb{Q}$, with a similar definition for $\mathbb{P}_j \left( \cdot \mid X_1^{j-1} \right)$.*

**Lemma D.10** (Bretagnolle-Huber Inequality, Theorem 14.1 in [Lattimore and Szepesvári, 2020]). *Let $\mathbb{P}$ and $\mathbb{Q}$ be probability measures on the same measurable space $(\Omega, \mathcal{F})$, and let $A \in \mathcal{F}$ be an arbitrary event. Then,*

$$\mathbb{P}(A) + \mathbb{Q}(A^c) \geq \frac{1}{2} \exp(-D(\mathbb{P}; \mathbb{Q}))$$

*where $A^c = \Omega \backslash A$ is the complement of A.*

**Lemma D.11** (Value Difference Lemma, Lemma E.15 in [Dann et al., 2017]). *For any two MDPs $\mathcal{M}'$ and $\mathcal{M}''$ with rewards $r'$ and $r''$ and transition probabilities $\mathbb{P}'$ and $\mathbb{P}''$, the difference in values with respect to the same policy $\pi$ can be written as*

$$V_i'(s) - V_i''(s) = \mathbb{E}'' \left[ \sum_{t=i}^{H} (r'(s_t, a_t, t) - r''(s_t, a_t, t)) \mid s_i = s \right]$$

$$+ \mathbb{E}'' \left[ \sum_{t=i}^{H} (\mathbb{P}'(s_t, a_t, t) - \mathbb{P}''(s_t, a_t, t))^\top V_{t+1}' \mid s_i = s \right]$$

*where $V_{H+1}'(s) = V_{H+1}''(s) = 0$ for any state $s$ and the expectation $\mathbb{E}'$ is taken with respect to $\mathbb{P}'$ and $\pi$ and $\mathbb{E}''$ with respect to $\mathbb{P}''$ and $\pi$.*

# E  Details of the planning phase

In this section, we provide some details of the planning phase in algorithm 3. In the planning phase, we are given a reward function $r : \mathcal{S} \times \mathcal{A} \to [0, 1]$ and we compute an estimate of sparsified transition dynamics $\widetilde{\mathbb{P}}^\dagger$, which is formally defined appendix A.1.1. The goal of the planning phase is to compute the optimal policy $\pi_{final}$ on the MDP specified by the transition dynamics $\widetilde{\mathbb{P}}^\dagger$ and reward function $r^\dagger$, where $r^\dagger$ is the sparsified version of $r$ : $r^\dagger(s, a) = r(s, a)$ and $r^\dagger(s^\dagger, a) = 0$ for any $a \in \mathcal{A}$. To compute the optimal policy, we iteratively apply the Bellman optimality equation. First, we define $\widetilde{Q}_H(s, a) = r^\dagger(s, a)$ for any $(s, a)$ and solve

$$\pi_{final, H}(s) = \arg\max_{a \in \mathcal{A}} r^\dagger(s, a).$$

Then, for $h = H - 1, H - 2, ..., 2, 1$, we iteratively define

$$\widetilde{Q}_h(s, a) := r^\dagger(s, a) + \sum_{s'} \widetilde{\mathbb{P}}^\dagger(s' \mid s, a) \widetilde{Q}_{h+1}(s', \pi_{final, h+1}(s'))$$

for any $(s, a)$, and then solve

$$\pi_{final, h}(s) = \arg\max_{a \in \mathcal{A}} \widetilde{Q}_h(s, a)$$

for any $s \in \mathcal{S}$. For $s^\dagger$ and any $h \in [H]$, $\pi_{final, h}(s^\dagger)$ can be arbitrary action. Then, from the property of Bellman optimality equation, we know $\pi_{final}$ is the optimal policy on $\widetilde{\mathbb{P}}^\dagger$ and $r^\dagger$.

# F  More related works

**Other low-switching algorithms** Low-switching learning algorithms were initially studied in the context of bandits, with the UCB2 algorithm [Auer et al., 2002] achieving an $O(\mathcal{A} \log K)$ switching

cost. Gao et al. [2019] demonstrated a sufficient and necessary $O(\mathcal{A} \log \log K)$ switching cost for attaining the minimax optimal regret in multi-armed bandits. In both adversarial and stochastic online learning, [Cesa-Bianchi et al., 2013] designed an algorithm that achieves an $O(\log \log K)$ switching cost.

**Reward-free reinforcement learning** In reward-free reinforcement learning (RFRL) the goal is to find a near-optimal policy for any given reward function. [Jin et al., 2020a] proposed an algorithm based on EULER [Zanette and Brunskill, 2019] that can find a $\varepsilon$ policy with $\widetilde{O}(H^5 |\mathcal{S}|^2 |\mathcal{A}| /\varepsilon^2)$ trajectories. Subsequently, [Kaufmann et al., 2021] reduces the sample complexity by a factor $H$ by using uncertainty functions to upper bound the value estimation error. The sample complexity was further improved by another $H$ factor by [Ménard et al., 2021].

A lower bound of $\widetilde{O}(H^2 |\mathcal{S}|^2 |\mathcal{A}| /\varepsilon^2)$ was established for homogeneous MDPs by [Jin et al., 2020a], while an additional $H$ factor is conjectured for non-homogeneous cases. [Zhang et al., 2021] proposed the first algorithm with matching sample complexity in the homogeneous case. Similar results are available with linear [Wang et al., 2020a, Wagenmaker et al., 2022, Zanette et al., 2020] and general function approximation [Chen et al., 2022, Qiu et al., 2021].

**Offline reinforcement learning** In offline reinforcement learning the goal is to learn a near-optimal policy from an existing dataset which is generated from a (possibly very different) logging policy. Offline RL in tabular domains has been investigated extensively [Yin and Wang, 2020, Jin et al., 2020c, Nachum et al., 2019, Rashidinejad et al., 2021, Kallus and Uehara, 2022, Xie and Jiang, 2020a]. Similar results were shown using linear [Yin et al., Xiong et al., 2022, Nguyen-Tang et al., 2022, Zanette et al., 2021b] and general function approximation[Xie et al., 2021a, Long et al., 2021, Zhang et al., 2022, Duan et al., 2021, Jiang and Huang, 2020, Uehara and Sun, 2021, Zanette and Wainwright, 2022, Rashidinejad et al., 2022, Yin et al., 2022]. Offline RL is effective when the dataset 'covers' a near optimal policy, as measured by a certain concentrabiluty factor. In the function approximation setting additional conditions, such as Bellman completeness, may need to be approximately satisfied [Munos and Szepesvári, 2008, Chen and Jiang, 2019, Zanette, 2023, Wang et al., 2020b, Foster et al., 2021, Zhan et al., 2022].

**Task-agnostic reinforcement learning** Another related line of work is task-agnostic RL, where $N$ tasks are considered during the planning phase, and the reward functions is collected from the environment instead of being provided directly. [Zhang et al., 2020a] presented the first task-agnostic algorithm, UBEZero, with a sample complexity of $\widetilde{O}(H^5 |\mathcal{S}| |\mathcal{A}| \log(N)/\varepsilon^2)$. Recently, [Li et al., 2023a] proposed an algorithm that leverages offline RL techniques to estimate a well-behaved behavior policy in the reward-agnostic phase, achieving minimax sample complexity. Other works exploring effective exploration schemes in RL include [Hazan et al., 2019, Du et al., 2019, Misra et al., 2020]. [Li et al., 2023b] also considered an offline-to-online reinforcement learning algorithm which explores the environment using two mixed policies in a reward-free mode.

