\leq 1$, we have $\exp(X_t - P_t) - (X_t - P_t) - 1 \leq (X_t - P_t)^2 (e - 2) \leq (X_t - P_t)^2$. Taking expectation conditional on $\mathcal{F}_{t-1}$ and noticing that $P_t - X_t$ is a Martingale difference sequence w.r.t. the filtration $\mathcal{F}_t$, we have

$$\mathbb{E}\left[\exp(X_t - P_t) \mid \mathcal{F}_{t-1}\right] \leq 1 + \mathbb{E}\left[(X_t - P_t)^2 \mid \mathcal{F}_{t-1}\right] \leq \exp\left[\mathbb{E}\left[(X_t - P_t)^2 \mid \mathcal{F}_{t-1}\right]\right] \leq \exp(P_t),$$

where the last inequality comes from the fact that conditional on $\mathcal{F}_{t-1}$, $X_t$ is a Bernoulli random variable. We define $M_n := \exp\left[\sum_{t=1}^{n}(X_t - 2P_t)\right]$, which is a supermartingale from our derivation above. We define now the stopping time $\tau = \min\left\{t \in \mathbb{N} : M_t > e^W\right\}$ and the sequence $\tau_n = \min\left\{t \in \mathbb{N} : M_t > e^W \vee t \geq n\right\}$. Applying the convergence theorem for nonnegative supermartingales (Theorem 4.2.12 in [Durrett, 2019]), we get that $\lim_{t \to \infty} M_t$ is well-defined almost surely. Therefore, $M_\tau$ is well-defined even when $\tau = \infty$. By the optional stopping theorem for nonnegative supermartingales (Theorem 4.8.4 in [Durrett, 2019], we have $\mathbb{E}[M_{\tau_n}] \leq \mathbb{E}[M_0] \leq 1$ for all $n$ and applying Fatou's lemma, we obtain $\mathbb{E}[M_\tau] = \mathbb{E}[\lim_{n \to \infty} M_{\tau_n}] \leq \liminf_{n \to \infty} \mathbb{E}[M_{\tau_n}] \leq 1$.