# OpenReview forum: "Policy Finetuning in Reinforcement Learning via Design of Experiments using Offline Data"
_NeurIPS.cc/2023/Conference — NeurIPS 2023 poster_

### Official Review · Reviewer_s51D · 2023-07-02

**Soundness:** 3 good
**Presentation:** 4 excellent
**Contribution:** 3 good
**Rating:** 7
**Confidence:** 4

**Summary:**

The authors introduced the concept of sparsified MDP. Based on this concept, they proposed a new algorithm that takes as input a dataset, uses it to design and deploy a non-reactive exploratory policy, and then outputs a locally near-optimal policy. A nearly minimax-optimal upper bound for the sample complexity is also established to learn a local ε-optimal policy using this algorithm.

**Strengths:**

- This paper proposed a new setting where the agent is access to an offline dataset and can further explore the environment online with a non-reactive policy. In this way, the agent can utilize the offline dataset while avoid the engineering costs associated with switching policies.
- Sparsified MDP provides an perspective on combining both optimism and pessimism principle, which is interesting.
- The authors obtain the sample complexity bound of the new approach, and the bound is tighter than previous method.
- The paper is well-written and easy to follow.


**Weaknesses:**

- The algorithm is based on tabular setting, which limits its application.
- The exploratory policy only explore in the sparsified MDP. In this way, the final policy is strictly limited by the offline dataset even if the budget for online interaction is large.
- The transitions that cannot achieve the threshold is dropped directly, even though they contain some information of the environment.

**Questions:**

- Is it possible for the exploratory policy to explore out of the coverage of offline dataset? I know it is impossible if there is no data at all, but   it seems that using the information of the dropped transitions may help.

**Limitations:**

Limitations are not explicitly discussed by the paper.

---

> ### Author Rebuttal · Authors · 2023-08-10
>
> Thank you for your comments and suggestions on our paper! We will try to answer your questions below.
>
> 1. Q: The algorithm is based on tabular setting, which limits its application.
>
> A: Indeed our paper is a first step for exploration with a non-reactive policy which is computed with the help of offline data.
>
> Although we could have tackled the setting of function approximation, doing so would have involved a number of technical considerations that might have made the result more cluttered and less clear for a first paper on such topics. We do agree with the reviewer that the extensions to the function approximation setting are important future directions, and we will mention this in the conclusion.
>
> 2. Q: The exploratory policy only explores in the sparsified MDP. In this way, the final policy is strictly limited by the offline dataset even if the budget for online interaction is large.
>
> A: The reviewer’s observation is correct, and this is a limitation of exploration with non-reactive policies. In the worst case, one cannot explore the regions not covered by the offline dataset because they may contain hard-to-explore combinatorial structures which command an exponential sample complexity[1]. More practical problems often do not have such worst-case structure; however what can be achieved in those cases is problem dependent and can be an interesting direction for future research.
>
> 3. Q: The transitions that cannot achieve the threshold are dropped directly, even though they contain some information about the environment. Is it possible for the exploratory policy to explore out of the coverage of offline dataset? I know it is impossible if there is no data at all, but it seems that using the information of the dropped transitions may help.
>
> [1] Xiao et al, The curse of passive data collection in batch reinforcement learning.
>
> A: The reviewer is correct that some transitions are dropped even if in certain cases they may contain enough data about the environment. For example, some environments are nearly deterministic, and so even a few samples (below the value of our threshold) would suffice to explore. However, how to explore the regions that do not have enough data, and whether it is even possible to do so, depends on the particular structure of the problem. Including these considerations would have made the algorithm more complex and the analysis more cluttered and so we settled for a ‘hard’ threshold which is simpler but still very effective.
>
> 4.  Limitations are not explicitly discussed by the paper.
>
> A: Thanks for your advice on the writing! We will definitely add a conclusion with emphasis on the significance as well as the limitations of our works to the next version of paper.
>
> To summarize, in this paper we leverage offline data to conduct exploration using non-reactive policies. The key contributions lie in the originality of the setup and in the mathematical work to describe the conditions that need to be met for sub-optimality guarantees. Algorithmically, this is achieved by a novel blending of the principle of optimism and pessimism to design the exploration policy in a way that is probably efficient. Extending these algorithmic and theoretical insights to derive a practical reinforcement learning algorithm with function approximation is an important next step.

---

> > ### Comment · Reviewer_s51D · 2023-08-22
> >
> > Thanks for your response. I think this paper is meaningful and am looking forward to your future work to mitigate its shortness.

---

### Official Review · Reviewer_wsu5 · 2023-07-06

**Soundness:** 4 excellent
**Presentation:** 3 good
**Contribution:** 3 good
**Rating:** 7
**Confidence:** 3

**Summary:**

The paper explores reinforcement learning applications where a pre-existing dataset of collected experience is available, and suggests the possibility of obtaining additional online data to enhance policy quality. To avoid the costs associated with switching policies, the authors propose utilizing a single non-reactive exploration policy for gathering supplementary data. They present an algorithm with provable guarantees that leverages an offline dataset to design such a policy. The algorithm is analyzed theoretically, and the authors evaluate the final policy's quality based on the local coverage of the original dataset and the amount of additional data collected. Overall, the research contributes to improving reinforcement learning by optimizing data acquisition and policy design.

**Strengths:**

The advantages of this paper are as follows:
1. Although I haven't carefully derived each equation, the proofs in this paper are expected to be accurate, with complete steps and rigorous derivations.
2. This paper represents the pioneering effort in terms of theoretical rigor, addressing the challenge of designing an experiment in reinforcement learning for online passive exploration, leveraging a dataset comprising pre-collected experiences. The setting is novel and has some practical significance.
3. The proof method presented in this paper is innovative and the sample complexity mentioned in the conclusion is also tight. It will introduce new approaches to the theoretical research of RL. Additionally, the conclusions of this paper also reveal some interesting insights that will enrich the existing theoretical achievements in the offline-to-online field.


**Weaknesses:**

1. The basic assumption of this article is that the offline-trained policy cannot be switched during the online phase, and it allows for the collection of an unlimited number of samples using this policy. I have two concerns. Firstly, from a theoretical research perspective, this assumption narrows down the problem to a very specific setting, so even with rigorous mathematical proofs, the generalizability of the conclusions may be compromised. Secondly, from a practical application standpoint, if the offline-trained policy itself is poor but cannot be switched during the online phase, and a large number of online samples need to be collected using this policy, there will be even greater security issues. As a result, the advocated security considerations in this paper will no longer exist. I appreciate the mathematical methods used in this paper, but the lack of persuasive motivation will impact the significance of this paper.
2. The proof process in this paper is too lengthy and difficult to understand. Although the main text provides some introduction to the overall proof logic, there are too many specific terms involved in the proof without providing intuitive explanations for their generation, which increases the difficulty of understanding. I suggest that the author can provide a more concise version, even if the resulting sample complexity is not optimal, but it can be used to understand the overall proof framework.
3. Another limitation is that it currently only applies to smaller S and A. When S and A are larger, the application will become difficult and the sample complexity will be significant. So I'm curious to know if this approach can be extended to scenarios involving function approximation. If it is possible, what additional considerations or processing steps would be required? If it is not feasible, what challenges exist?


**Questions:**

See weakness.

**Limitations:**

The authors didn't discuss the limitation of their work.

---

> ### Author Rebuttal · Authors · 2023-08-10
>
> Thank you for your comments and suggestions on our paper! We will try to answer your questions below.
>
> 1. Q: Firstly, from a theoretical research perspective, this assumption narrows down the problem to a very specific setting, so even with rigorous mathematical proofs, the generalizability of the conclusions may be compromised.
>
> A: Thanks for your question! Our study on non-reactive exploration arise from the need for low deployment and switching costs in the real-world applications across various domains such as recommendation systems and healthcare. By necessity, we formalized such general research questions in a specific setting where we could obtain concrete results, but we believe that some of the insights and algorithmic design principles extend more broadly.
>
> 2. Q: Secondly, from a practical application standpoint, if the offline-trained policy itself is poor but cannot be switched during the online phase, and a large number of online samples need to be collected using this policy, there will be even greater security issues. As a result, the advocated security considerations in this paper will no longer exist. I appreciate the mathematical methods used in this paper, but the lack of persuasive motivation will impact the significance of this paper.
>
> A: Although safety is an important concern in the application of reinforcement learning,  it is largely orthogonal to the issues investigated in our paper. As the reviewer noticed, the safety of the online phase can partially depend on the pre-collected dataset, so if we want to take more safety factors into consideration, we can of course apply some additional techniques.
>
> Generally speaking, there are two ways to incorporate safety. The first is to incorporate some techniques from safe RL [1,2,3] into the design of the algorithm, and also to the procedure of collecting the offline dataset. The second is before deployment: non-reactive exploration produces a single policy, and so safety can be checked quite easily before deployment.
>
> 3. Q: The proof process in this paper is too lengthy and difficult to understand. Although the main text provides some introduction to the overall proof logic, there are too many specific terms involved in the proof without providing intuitive explanations for their generation, which increases the difficulty of understanding. I suggest that the author can provide a more concise version, even if the resulting sample complexity is not optimal, but it can be used to understand the overall proof framework.
>
> A: We understand that the proof can be long, but this is often a necessary compromise for the sake of rigor, and we will try our best to make it as clear as we can by adding some descriptions about the overall proof strategy.
>
> 4. Q: Another limitation is that it currently only applies to smaller S and A. When S and A are larger, the application will become difficult and the sample complexity will be significant. So I'm curious to know if this approach can be extended to scenarios involving function approximation. If it is possible, what additional considerations or processing steps would be required? If it is not feasible, what challenges exist?
>
> A: Thanks for your question! As the first paper working on the non-reactive exploring policy in offline-to-online RL, we considered the tabular case.
>
> It is an interesting next step to extend these insights to the function approximation setting. In the function approximation setting, one probably needs to overcome additional challenges, starting from defining of sparsified MDP in a way that takes into account function approximation. The method of adding positive bonus in the offline simulation phase can also be applied with slight change to the concrete form of bonus functions in the function approximation setting.
>
> 5. The authors didn't discuss the limitations of their work.
>
> A: Thanks for your advice on the writing! We will definitely add a conclusion with emphasis on the significance as well as the limitations of our works.
>
> To summarize, in this paper we leverage offline data to conduct exploration using non-reactive policies. The key contributions lie in the originality of the setup and in the mathematical work to describe the conditions that need to be met for sub-optimality guarantees. Algorithmically, this is achieved by a novel blending of the principle of optimism and pessimism to design the exploration policy in a way that is probably efficient. Extending these algorithmic and theoretical insights to derive a practical reinforcement learning algorithm with function approximation is an important next step.
>
> [1]. Gu et al. A review of safe reinforcement learning: Methods, theory and applications.
> [2]. Ding et al. Provably efficient safe exploration via primal-dual policy optimization.
> [3]. Cheng. End-to-end safe reinforcement learning through barrier functions for safety-critical continuous control tasks.

---

> > ### Comment · Reviewer_wsu5 · 2023-08-15
> >
> > Thanks for your detailed response and I have raised my score to 7. I would like to see the proofs with  more clear description in the final paper. Wishing you all the best with your publication.

---

### Official Review · Reviewer_TPkj · 2023-07-07

**Soundness:** 3 good
**Presentation:** 3 good
**Contribution:** 3 good
**Rating:** 5
**Confidence:** 2

**Summary:**

This paper proposes an algorithm for policy fine-tuning in reinforcement learning using a dataset of pre-collected experience. The algorithm leverages the dataset to design a non-reactive exploratory policy and outputs a locally near-optimal policy. The paper makes theoretical contributions in analyzing the quality of the policy and establishing sample complexity bounds. The work is motivated by the practical need for non-reactive exploration in domains where switching policies is costly and impractical. The paper presents innovative ideas and provides a novel solution to the problem.

**Strengths:**

- The paper introduces an algorithm that addresses the problem of non-reactive policy design in reinforcement learning and provides provable guarantees for the quality of the resulting policy.
- The concept of sparsiﬁed MDP is introduced and effectively used in the algorithm and theoretical analysis.
- The paper rigorously establishes a nearly minimax-optimal upper bound for the sample complexity needed to learn a local ε-optimal policy using the proposed algorithm.
- The paper addresses a practical need for non-reactive exploration in domains where policy switches are costly and provides a solution that can be valuable in such scenarios.


**Weaknesses:**

- The paper only considers discrete state and action spaces.
- The paper provides no empirical evaluations or demonstrations of the proposed algorithm. Neither does it shed light on the design of practical algorithms.


**Questions:**

- Can you give more details on the non-reactive property of a policy? It seems that most RL policies in an MDP will be non-reactive, as long as they take only the current state $s_t$ as input. How is the example in line 30-32 related to the non-reactive property?
- Online exploration may lead to safety violations. How safe will the online phase be in the original MDP?
- It seems contradictory to be both optimistic and pessimistic at the same time. How to determine the region that the agent knows how to navigate?
- Can you provide some empirical evaluations of the proposed algorithm, compared with fully offline or fully online algorithms?


**Limitations:**

Yes

---

> ### Author Rebuttal · Authors · 2023-08-10
>
> Thank you for your comments and suggestions on our paper! We will try to answer your questions below.
>
> 1. Q: Can you give more details on the non-reactive property of a policy? It seems that most RL policies in an MDP will be non-reactive, as long as they take only the current state s_t as input. How is the example in line 30-32 related to the non-reactive property?
>
> A: Thank you for your question and we will try to make it clearer in the next version. **A non-reactive policy is a policy that is not updated during the whole interaction process.**
>
> As the reviewer notices, RL algorithms typically employ a sequence of non-reactive policies as they explore the environment. **These algorithms are reactive to the data they acquire, because such data is used to update the currently deployed policy.** The sequence of non-reactive policies that they deploy could be interpreted as a **single reactive** policy, namely one that changes during the interaction.
>
> In contrast our work examines the setting where exploration must be done with a **single non-reactive** policy: in our case, no updates are possible during the entire exploration phase.
>
> While deploying a sequence of policies is the better approach from a theoretical point of view, **doing so is not always practically feasible**. In line 30-32 we give an example with a human in the loop. The human may need significant time to validate each policy that a reactive algorithm produces, and this may make real-time deployment of a reactive algorithm impractical. However, such a problem disappears if the algorithm only deploys a static policy for exploration, because this policy only needs to be checked once before deployment.
>
> 2. Q: Online exploration may lead to safety violations. How safe will the online phase be in the original MDP?
>
> A: Although safety is an important concern in the application of reinforcement learning,  it is not the main motivation of our paper. Nonetheless, there are two ways to incorporate safety.
>
> The first is to incorporate some techniques from safe RL [4,5,6] into the design of the algorithm, and also to the procedure of collecting the offline dataset. The second is before deployment: non-reactive exploration produces a single policy, and so safety can be checked quite easily before deployment.
>
> 3. Q: It seems contradictory to be both optimistic and pessimistic at the same time. How to determine the region that the agent knows how to navigate?
>
> A: One of our contributions is that our algorithm combines the principle of optimism and pessimism in a subtle way: they are applied to different regions of the MDP and so there is no contradiction between these two principles. We will clarify this in the introduction.
>
> At a high level, pessimism excludes the region where we have little to no knowledge about the transition dynamics, and it identifies a sub-MDP where approximate planning is possible. Within the sub-MDP, we explore using the principle of optimism. In summary, pessimism defines the MDP sub-region where we can leverage optimism to conduct exploration.
>
> 4. Q: Can you provide some empirical evaluations of the proposed algorithm, compared with fully offline or fully online algorithms?
>
> A: We thank reviewers for the suggestion that the paper could be made stronger with the addition of numerical experiments. Although we agree with the reviewers’ suggestions, a practically useful RL algorithm for this setting would need to leverage function approximation such as neural networks. Designing an effective algorithm for such a setting requires making several critical design choices that are specific to the function approximation setting, and overcoming the challenges that likely arise. This is beyond the scope of the paper, which is a first step towards understanding how offline data can be used for non-reactive exploration. We will highlight in the conclusion of the paper that the creation of a practical algorithm with function approximation is an important future direction.
>
> 5. Q: The paper only considers discrete state and action spaces.
>
> Indeed our paper is a first step for exploration with a non-reactive policy which is computed with the help of offline data.
>
> Although we could have tackled the setting of function approximation, doing so would have involved a number of technical considerations that might have made the result more cluttered and less clear for a first paper on such topics. We do agree with the reviewer that the extensions to the function approximation setting are important future directions, and we will mention this in the conclusion.
>
> [1]. Jin et al. Is q-learning provably effi394 cient?
> [2]. Kaufmann et al. Adaptive reward-free exploration.
> [3]. Ménard et al. Fast active learning for pure exploration in reinforcement learning.
> [4]. Gu et al. A review of safe reinforcement learning: Methods, theory and applications.
> [5]. Ding et al. Provably efficient safe exploration via primal-dual policy optimization.
> [6]. Cheng et al. End-to-end safe reinforcement learning through barrier functions for safety-critical continuous control tasks.

---

> > ### Comment · Reviewer_TPkj · 2023-08-19
> > **Response**
> >
> > I acknowledge the authors' rebuttal and I remain my rating towards acceptance.

---

### Official Review · Reviewer_rHsz · 2023-07-07

**Soundness:** 3 good
**Presentation:** 4 excellent
**Contribution:** 3 good
**Rating:** 7
**Confidence:** 3

**Summary:**

The paper proposes an algorithm that, given a previously collected dataset of transitions from an MDP, produces a non-reactive policy that can effectively collect additional data that enables a near-optimal policy to be obtained for any possible reward function. The algorithm is model-based and combines elements of optimism (exploration bonuses) and pessimism (early termination at OOD states/actions). Suboptimality is bounded relative to the optimal policy that stays within the “sparsified MDP” which is the subset of the original MDP that is sufficiently covered by the data.

**Strengths:**

* The paper presents positive, novel (to my knowledge) theoretical results in a well-motivated setting of practical interest.
* The interaction protocol is new AFAIK, but closely related to existing areas such as reward-free RL. It may inspire more work in the future.
* The paper is clear and not hard to follow, despite its technicality.

**Weaknesses:**

* No experiments, despite having “Design of Experiments” in the title :) (But this is a theory paper so I think it is okay)
* IMO the Related Work should cite the MOReL paper [1], which uses a pessimistic MDP construction similar to your sparsified MDP, in which low-density states/actions lead to a special absorbing state. Of course, they are tackling a different setting (offline RL with a particular reward function) and use a somewhat different termination criterion, but the idea is the same.

**Questions:**

I’m curious if it is known that better sample complexity can be obtained if you allow the policy to be adaptive? (Although I understand the engineering-related reasons for not doing so.) If so, it could be useful context to briefly comment on in the paper.

**Limitations:**

No, limitations are not addressed.

---

> ### Author Rebuttal · Authors · 2023-08-10
>
> Thank you for your comments and suggestions on our paper! We will try to answer your questions below.
>
> 1. Q: I’m curious if it is known that better sample complexity can be obtained if you allow the policy to be adaptive? (Although I understand the engineering-related reasons for not doing so.) If so, it could be useful context to briefly comment on in the paper.
>
> A: This is a very good question and we will stress this point in the next version of our paper. Yes, one can do better if you use reactive exploration policy. By using reactive policies, you can achieve eps-optimal policy on the entire MDP with O(H^2 S^2 A / \eps^2) trajectories[1,2], while the non-reactive exploring policies can only yield an eps-optimal policy on the sparsified MDP. This difference is natural since we can only effectively explore the region where we have some knowledge about (i.e., the sparsified MDP).
>
>
> 2. Q: No experiments, despite having “Design of Experiments” in the title :) (But this is a theory paper so I think it is okay)
>
> A: We thank reviewers for the suggestion that the paper could be made stronger with the addition of numerical experiments. Although we agree with the reviewers’ suggestions, a practically useful RL algorithm for this setting would need to leverage function approximation such as neural networks. Designing an effective algorithm for such a setting requires making several critical design choices that are specific to the function approximation setting, and overcoming the challenges that likely arise. This is beyond the scope of the paper, which is a first step towards understanding how offline data can be used for non-reactive exploration. We will highlight in the conclusion of the paper that the creation of a practical algorithm with function approximation is an important future direction.
>
> 3. IMO the Related Work should cite the MOReL paper, which uses a pessimistic MDP construction similar to your sparsified MDP, in which low-density states/actions lead to a special absorbing state. Of course, they are tackling a different setting (offline RL with a particular reward function) and use a somewhat different termination criterion, but the idea is the same.
>
> A: We thank the reviewer for pointing out a connection with MOReL; the paper is indeed relevant and we will cite it. As the reviewer notices, MOReL also constructs a sub-MDP using the principle of pessimism; a key algorithmic difference is that we combine such pessimistic construction with the principle of optimism to explore within the sub-MDP instead of directly using the dataset to output a policy.
>
> [1]. Ménard et al. Fast active learning for pure exploration in reinforcement learning.
> [2]. Jin et al. Reward-free exploration for reinforcement learning.

---

> > ### Comment · Reviewer_rHsz · 2023-08-21
> >
> > Thank you for your answers! I would say that my concerns are addressed and my score stands as-is. While experiments would of course strengthen the paper further (as the authors agree), I think the theoretical results are already a useful contribution.

---

### Official Review · Reviewer_gUtm · 2023-07-07

**Soundness:** 4 excellent
**Presentation:** 3 good
**Contribution:** 3 good
**Rating:** 7
**Confidence:** 2

**Summary:**

The work proposes a method to create a non-reactive exploratory policy from an initial input dataset. Then, leveraging the new data, the algorithm generates a locally near-optimal policy.

The relevance of the algorithm is in the low-switching algorithms, where it is assumed there is a cost to changing a deployed policy.

The proposed algorithm uses the initial data to build a sparsified MDP, an approximation of the original MDP that keeps only the transitions (s, a, s') for which there are at least Φ transitions.

The next part of the algorithm is building the exploratory policy using the sparsified MPD using a value iteration iterative strategy with an exploration bonus instead of actual rewards. The exploration bonus balances optimism and pessimism, boosting exploration towards less explored states while avoiding spending too much time in unknown parts of the MDP.

Finally, additional data is collected from the environment using this exploring policy, and a value iteration algorithm is employed on the combined datasets of experiences to build the final policy.

The authors provide optimality bounds, describing the conditions for the algorithm to discover an epsilon-suboptimal policy.

**Strengths:**

The strengths of this paper lie in the originality of the setup and in the mathematical work to describe the conditions that need to be met for sub-optimality guarantees.

**Weaknesses:**

The weakness of the paper is the need for empirical. For example, it would be helpful to see how well the algorithm performs given initial data sets of different sizes and coverage.

**Questions:**

Could some experiments be added to see the performance of the algorithm in practice, maybe compared to other offline RL methods?

---

> ### Author Rebuttal · Authors · 2023-08-10
>
> Thank you for your comments and suggestions on the paper! We will try to answer your questions below.
>
> 1. Q: The weakness of the paper is the need for empirical. For example, it would be helpful to see how well the algorithm performs given initial data sets of different sizes and coverage.
>
> A: We thank reviewers for the suggestion that the paper could be made stronger with the addition of numerical experiments. Although we agree with the reviewers’ suggestions, a practically useful RL algorithm for this setting would need to leverage function approximation such as neural networks. Designing an effective algorithm for such a setting requires making several critical design choices that are specific to the function approximation setting, and overcoming the challenges that likely arise. This is beyond the scope of the paper, which is a first step towards understanding how offline data can be used for non-reactive exploration. We will highlight in the conclusion of the paper that the creation of a practical algorithm with function approximation is an important future direction.

---

### Official Review · Reviewer_CnHc · 2023-07-24

**Soundness:** 4 excellent
**Presentation:** 3 good
**Contribution:** 3 good
**Rating:** 5
**Confidence:** 3

**Summary:**

The paper considers the setting where it is possible to leverage a dataset of transitions, together with the possibility of deploying a policy to collect additional information. The question then lies into what kind of policy should be deployed and what kind of data should be gathered. The authors argue that deploying an exploratory policy that switches, e.g. a policy that learns and adapts from its experience, leads a great engineering costs. As such, they propose to follow the principle of pessimism together with a non reactive policy that would be constrained to a sub-region of the MDP with enough transitions. They also make use of the principle of optimism to derive the exploratory actions taken within the subMDP. The authors provide a near optimal minimax-optimal upper bound for learning an epsilon-optimal policy.

**Strengths:**

The contributions and assumptions are made clear in the introduction (however, the concept of reactive policy is only clearly explained in plain a words a bit late). An intuition section is also provided which helps the reader follow the paper.

The authors provide theoretical guarantees both for the sparsified MDP together with the full MDP. In particular for the results on the full MDP it seems like a reduction in the epsilon coefficient is an important contribution.

The paper emphasizes a hybrid approach, exploring a setting that combines elements of both offline and online methods, hence offering a more practical and adaptable framework for various real-world applications.

**Weaknesses:**

Although the setting of interest is important (mix of online and offline) the paper doesn’t stress enough the importance of a non-reactive policy. Why is learning from the generated experience such a bad idea in practice? It seems like a slowly changing deployed policy (where changes perhaps happen through a trust region) would be a better choice.

I understand the paper is essentially a theoretical one, however it would be interesting to present some empirical evidence as to the practicability of the proposed algorithm. Indeed, it is not clear if the current dependencies on state, action, epsilon and such quantities in the bounds would provide a meaningful difference.

The paper misses an opportunity to provide a comprehensive conclusion, one that effectively synthesizes the results, giving the reader a clear understanding of the study's overall implications and potential future directions.

**Questions:**

How good of a policy (trained from scratch) could be obtained with the number of additional samples required in Corollary 5.2?

Although the dependence on the desired accuracy is reduced in Corollary 5.2, how much of a difference can this make given the concentrability coefficient?

**Limitations:**

The limitations are not clearly stated throughout the work. Some of the questions above try to probe into this.

---

> ### Author Rebuttal · Authors · 2023-08-10
>
> Thank you for your comments and suggestions on our paper! We will try to answer your questions below.
>
> 1. Q: Why is learning from the generated experience such a bad idea in practice? It seems like a slowly changing deployed policy (where changes perhaps happen through a trust region) would be a better choice.
>
> A: We agree with the reviewer that a slowly changing policy would be the preferred choice (and the approach that we would recommend) if it can be implemented easily.
>
> However, an interactive system that adaptively changes interventions in response to outcomes may be impractical in some settings due to the expertise and infrastructure needed.  An example is (1) when multiple agents collect data asynchronously and real-time communication to update their policy is difficult (such as in a large organization that is selling ads) and (2) whenever there is a significant overhead in the engineering infrastructure to implement an adaptive policy (such as in an organization with a complex production code).
>
> Running an experiment with a fixed decision policy is much simpler logistically, (since such organizations may already design decision policies by hand) and appealing, since many areas (education, healthcare, social sciences) commonly deploy experiments to find the best approach.  In these settings there is a key opportunity to design a sample efficient non-reactive policy that can be used to gather additional data to later identify better decision rules.
>
> Although some of these considerations are mentioned in Lines 30-40, we will expand them with additional discussion.
>
>
> 2. Q: It would be interesting to present some empirical evidence as to the practicability of the proposed algorithm. Indeed, it is not clear if the current dependencies on state, action, epsilon and such quantities in the bounds would provide a meaningful difference.
>
> A: We thank reviewers for the suggestion that the paper could be made stronger with the addition of numerical experiments.
>
> Although we agree with the reviewers’ suggestions, a practically useful RL algorithm for this setting would need to leverage function approximation such as neural networks. Designing an effective algorithm for such a setting requires making several critical design choices that are specific to the function approximation setting, and overcoming the challenges that likely arise. This is beyond the scope of the paper, which is a first step towards understanding how offline data can be used for non-reactive exploration. We will highlight in the conclusion of the paper that the creation of a practical algorithm with function approximation is an important future direction.
>
> **We improved the sample complexity compared to a purely offline approach.** The best tabular offline algorithm requires O(H^3 S^2 C^* / \eps^2 + H^{5.5} S^2 C^* / \eps) samples to find an eps-optimal policy [4] (when ‘translated’ to the time-homogeneous and reward-free setting). Our sample complexity is O(H^3 S^2 A / \eps^2 + H^4 S^2 A C^* / \eps). We shave off the concentrability coefficient C^* in the leading term and also shave off H^1.5 in the O(1/\eps) term. Since the concentrability coefficient can be extremely large, our approach offers a significant  improvement over purely offline algorithms.
>
> More precisely, the current dependencies on the S,A,H and \eps are minimax optimal up to the log factors in the reward-free setting [3]. This means that in the worst case one can not obtain an eps-optimal policy using less samples than our bounds.
>
>
> 3. Q: How good of a policy (trained from scratch) could be obtained with the number of additional samples required in Corollary 5.2?
>
> A: Generally it is not possible to obtain meaningful guarantees when training *from scratch* with a single non-reactive policy. In fact, in the absence of any information about the MDP the best one can do is just to deploy a uniformly random policy, which commands an exponential O(exp(H)) sample complexity in hard-to-explore settings [1]. When exploring from scratch it is essential to use *reactive* policies; a well designed algorithm can find \eps-optimal policies with O(H^3 S^2 A / \eps^2) samples in the reward-free setting [3].
>
>
> 4. Q: Although the dependence on the desired accuracy is reduced in Corollary 5.2, how much of a difference can this make given the concentrability coefficient?
>
> A: In practice, this concentrability coefficient is unknown and can be large. Since we remove C^* from the main term (the one with O(1/\eps^2) dependence), the sample complexity now only depends on the concentrability  C^* through a lower-order term ( the one with O(1/\eps) dependence ). That is, the smaller the target accuracy \eps, the smaller the effect of a large concentrability C^*.
>
>
> 5. Q: The paper misses an opportunity to provide a comprehensive conclusion, and the limitations are not clearly stated throughout the work.
>
> A: Thanks for your suggestion and we will add an additional conclusion section in the future version.
>
> To summarize, in this paper we leverage offline data to conduct exploration using non-reactive policies. The key contributions lie in the originality of the setup and in the mathematical work to describe the conditions that need to be met for sub-optimality guarantees. Algorithmically, this is achieved by a novel blending of the principle of optimism and pessimism to design the exploration policy in a way that is probably efficient. Extending these algorithmic and theoretical insights to derive a practical reinforcement learning algorithm with function approximation is an important next step.
>
> [1]. Xiao et al, The curse of passive data collection in batch reinforcement learning.
> [2]. Qiao et al. Sample-efficient reinforcement learning with loglog (t) switching cost.
> [3]. Jin et al. Reward-free exploration for reinforcement learning.
> [4]. Xie et al. Policy finetuning: Bridging sample-efficient offline and online reinforcement learning.

---

> > ### Comment · Reviewer_CnHc · 2023-08-14
> >
> > Dear authors,
> >
> > thank you for the very detailed rebuttal, it certainly helps in understanding the paper and its results.
> >
> > It is true that a practically use deep RL algorithm would require neural networks, however this does necessarily mean that neural networks are required for an illustrative experiment to be added to the paper. This would complement well the application example provided by the authors where complex production code could inhibit a reactive policy: one would imagine that in such an example the non-reactive policy would be a simpler and more predictable model (e.g. linear function approximation). Is it likely that finding a minimal example of where the proposed strategy makes a difference in practice (even in the tabular setting)? If this is the case, it would be important to at least state it and encourage more research on the matter.
> >
> > I appreciate the authors clearly stating the connections to previous works and bounds, it is a more convenient way to assess the contributions of the paper. It is also good to know that the authors will rework some of the presentation to include these together with a conclusion. For these reasons I am raising my score.

---

### Decision · Program_Chairs · 2023-09-21

**Decision:**

Accept (poster)

**Comment:**

This paper rigorously characterizes the problem of learning from offline data without reward, followed by a fine-tuning phase using a non-reactive policy. All the reviewers voted for acceptance, some strongly so, and the authors responded well to the reviews. Reviewer CnHc was confident the work was correct and well executed but questioned if the results are particularly useful or will have any impact on practice. The AC had similar concerns and in consultation with the SAC determined some of this stems from missing justifications and explanations in the text, that would go a long way to improve the paper. In particular, we strongly recommend the authors address the following:

1. More justification for reward-free offline data
2. Main result Theorem 5.1 shows the performance bound for the sparsified MDP, not the original MDP. Do we care about the performance in the sparsified MDP? It might be possible to get some results like the performance loss compared to the best covered policy used in [1]. It is well worth extra text detailing why this bound is relevant
3. Explain why, in line 3 of algorithm 1, we receive the entire reward information.
4. "We rigorously establish a nearly minimax-optimal upper bound”. There is only an upper bound. Where is the minimax optimality from?
5. The paper should motivate why considering this particular separation of offline and online phase is useful. Corollary 5.2 shows the bound for the original MDP by assuming the standard concentrability coefficient. If we make the assumption that data is sampled i.i.d and the standard concentrability coefficient, offline data is sufficient to get a nearly optimal policy like Eq (5.4).
6. Line 49 is a bit misleading. The paper cites Xiao et. al., 2020 to show that using only data from a non-reactive policy cannot be sample efficient. However, the paper considered the setting where the offline data is sampled in an i.i.d. fashion from some distribution, only in the online phase the data is collected from a policy.

This work is correct and well executed (ignoring the writing issues highlighted above). Even if limited, there is a community of researchers that will find this work interesting and build on it, therefore this paper should be accepted.

[1] Provably Good Batch Reinforcement Learning Without Great Exploration